# Amortizing Bayesian Posterior Inference in Tractable Likelihood Models

## Abstract

Bayesian inference provides a natural way of incorporating prior beliefs and assigning a probability measure to the space of hypotheses. However, it is often infeasible in practice as it requires expensive iterative routines like MCMC to approximate the posterior distribution. Not only are these methods computationally expensive, but they must also be re-run whenever new observations are available, making them impractical or of limited use. To alleviate such difficulties, we amortize the posterior parameter inference for probabilistic models through permutation invariant architectures. While this paradigm is briefly explored in Simulation Based Inference (SBI), Neural Processes (NPs) and Gaussian Process (GP) kernel estimation, a more general treatment of amortized Bayesian inference in known likelihood models has been largely unexplored. We rely on the reverse-KL based amortized Variational Inference (VI) approach to train inference systems and compare them with forward-KL based SBI approaches across different architectural setups. We additionally introduce a simple but strong approach to further amortize on the number of features in each observation, allowing a single system to infer variable dimensional parameters. Our thorough experiments demonstrate the effectiveness of our proposed approach, especially in real-world and model misspecification settings.

## 1 Introduction

Bayesian analysis of data has become increasingly popular and is widely used in numerous scientific disciplines. In politics, predictive models based on public polling and other factors play a crucial role in the discourse around the state of a campaign. Throughout the COVID-19 pandemic, models that estimate the infectiousness of the virus, the efficacy of public health measures, and the future course of the pandemic became critical to government planning and the public's understanding of the pandemic (Cooper et al., 2020). In cryogenic electron microscopy (cryo-EM), the posterior over an unknown 3D atomic-resolution molecular structure is explored given the 2D image observations (Glaeser et al., 2021).

While recent years have made such methods more accessible (Bingham et al., 2019; Carpenter et al., 2017; Štrumbelj et al., 2023), they still remain computationally burdensome. Further, in practical contexts where new observations are continuously available, the analysis must be re-run every time new data becomes available, e.g., when new case counts become available, previous measurements are corrected, or when applied to different geographic regions. As a result practitioners adopt approximations (Welling & Teh, 2011; Gelfand, 2000; Brooks, 1998), simplify their models (Hoffman et al., 2013; Blei et al., 2017) or reduce the frequency with which they perform their analyses.

A common thread is that the probabilistic model defining the relationship between its parameters and the observations is fixed. Poll aggregation models use hierarchical time series models (Athanasopoulos et al., 2023; Chen et al., 2023), infectious diseases are studied using variants on compartment models (Tang et al., 2020), and cryo-EM uses a linear image formation model (Glaeser et al., 2021). This makes these applications ideal candidates for amortized inference (Morris, 2013; Paige & Wood, 2016; Kingma & Welling, 2013; Rezende et al., 2014; Stuhlmüller et al., 2013).

We propose using neural networks to learn a function that maps an observed *dataset* directly to the corresponding posterior distribution without the need for iterative procedures, e.g., Markov chain Monte

Carlo (MCMC) sampling (Gelfand, 2000; Hoffman et al., 2014). To efficiently handle permutation invariance stemming from the *iid* nature of observations, we rely on efficient set-based architectures like Transformers and DeepSets (Zaheer et al., 2017; Vaswani et al., 2017; Lee et al., 2019). If learned properly, this mapping would allow generalization to different datasets zero-shot. We also leverage a simple masking procedure to amortize posterior estimation over datasets with a variable number of features, thus inching closer towards a generalist Bayesian learner − as evidenced by its generalization capabilities on real-world tasks zero-shot through only pre-training on synthetic data.

Generally, real-world datasets do not exactly follow standard models, e.g., while practitioners often rely on linear models, data rarely follows them exactly. Amortization methods like simulation-based inference (SBI), in particular neural posterior estimation (SBI-NPE), (Radev et al., 2020; Geffner et al., 2023; Cranmer et al., 2020) suffer from a gap in the training (simulation) and evaluation (real-world) protocol and thus can be suboptimal. Even further, they require access to the underlying model parameters generating the datasets during training and are thus not applicable in many practical settings where the true parameters are unknown.

Our primary motivation is posterior inference as the parametric values themselves are often of interest in applied statistical practice, e.g., for assessing the success of a pandemic intervention or the impact of a factor on public opinion polling. However, evaluating approximate posteriors is extremely difficult since the true posterior is highly intractable to either evaluate or sample from with the exception of MCMC sampling which only provides asymptotic guarantees. Despite their intractability, Bayesian posteriors have been claimed to provide better generalization capabilities as they encompass multiple different hypotheses in making predictions. Following this line of reasoning, we demonstrate the utility of our proposed approach in the closely related problem of posterior prediction, where the goal is to model future predictions given some observations.

Since SBI-NPE approaches are generally modeled through minimization of forward-KL objective, they suffer from over-dispersion during Bayesian posterior inference and consequently worse predictive performance. Instead, we propose a new training objective that can operate solely on datasets without access to the underlying parameters, allowing for a wider diversity of data to be incorporated during training for better generalization to real world settings. Our detailed experiments and benchmarking establish the superiority of our proposed approach as well as their qualitative differences. Our contributions include

- Proposing a general framework for amortizing Bayesian posterior estimation in probabilistic models and highlighting its superior performance to SBI-NPE.
- Extending amortization in both SBI-NPE and the proposed framework over a variable number of features in each observation, and not just the number of observations.
- Quantifying benefits of our proposed approach when the true underlying model is unknown (model misspecification), as well as providing its advantages in generalization to real-world tabular datasets.
- Benchmarking various design choices like architectural backbones through extensive ablations.

## 2 Background

We first cover some of the important preliminaries as well as approaches already existing in the literature.

**Bayesian Inference**. Let $\boldsymbol{x} \in \mathbb{R}^d$ denote the outcome of an experiment observed through a set of independent and identically (*iid*) distributed samples $\mathcal{D} = \{\boldsymbol{x}_1, ..., \boldsymbol{x}_N\}$. Given these observations, we are often interested in either quantifying the certainty of or generating potential future observations $\boldsymbol{x}_*$. Bayesian Inference provides a natural methodology of quantifying $p(\boldsymbol{x}_*|\mathcal{D})$ by prescribing a space of hypotheses $\boldsymbol{\theta} \in \mathbb{R}^k$ and a *prior* belief $p(\boldsymbol{\theta})$ over it. These hypotheses $\boldsymbol{\theta}$ define the *likelihood* of observing a particular outcome, i.e., $p(\boldsymbol{x}|\boldsymbol{\theta})$. The likelihood and prior are then combined through Bayes rule to define the *posterior* $p(\boldsymbol{\theta}|\mathcal{D})$, through which the quantity of interest can then be easily expressed as

$$p(\boldsymbol{x}_*|\mathcal{D}) = \int_{\boldsymbol{\theta}} p(\boldsymbol{x}_*|\boldsymbol{\theta})p(\boldsymbol{\theta}|\mathcal{D})d\boldsymbol{\theta} \tag{1}$$

However, the above expression poses two challenges: (a) the *posterior* $p(\boldsymbol{\theta}|\mathcal{D})$, which is often a quantity of interest in itself, is not known, and (b) even if known, the integration might be intractable. The intractability

of the integration is often resolved through Monte Carlo estimation

$$p(\boldsymbol{x}_*|\mathcal{D}) = \mathbb{E}_{\boldsymbol{\theta}|\mathcal{D}}\left[p(\boldsymbol{x}_*|\boldsymbol{\theta})\right] \tag{2}$$

$$\approx \frac{1}{M}\sum_{m=1}^{M}p(\boldsymbol{x}_*|\boldsymbol{\theta}^{(m)}) \tag{3}$$

where $\boldsymbol{\theta}^{(m)} \sim p(\boldsymbol{\theta}|\mathcal{D})$. The quantity $p(\boldsymbol{\theta}|\mathcal{D})$ can be obtained through an application of Bayes rule

$$p(\boldsymbol{\theta}|\mathcal{D}) = \frac{p(\mathcal{D}|\boldsymbol{\theta})\,p(\boldsymbol{\theta})}{p(\mathcal{D})} \tag{4}$$

$$= \frac{p(\boldsymbol{\theta})}{p(\mathcal{D})}\prod_{n=1}^{N}p(\boldsymbol{x}_n|\boldsymbol{\theta}) \tag{5}$$

Given the form of the *likelihood* and the *prior*, the above distribution is often difficult to sample from, especially with the added complexity of the marginal $p(\mathcal{D}) = \int_{\boldsymbol{\theta}} p(\mathcal{D}|\boldsymbol{\theta})\,p(\boldsymbol{\theta})$ being intractable. Additionally, the posterior itself is often of interest on its own, especially in cases where $\boldsymbol{\theta}$ is interpretable, for example, if we model the bias of a coin based on multiple tosses. We refer the readers to Bishop & Nasrabadi (2006) for applications of Bayesian Inference to supervised learning, etc.

**Variational Inference**. To bypass the intractability of the posterior distribution, or at least the difficulty to sample from it, VI methods approximate the true posterior $p(\boldsymbol{\theta}|\mathcal{D})$ with a variational distribution $q_\varphi(\boldsymbol{\theta})$ and convert the estimation problem into the following optimization problem

$$\varphi^* = \arg\min_{\varphi} \mathbb{KL}[q_\varphi(\cdot)||p(\cdot|\mathcal{D})] \tag{6}$$

which is equivalent to optimizing the Evidence Lower-Bound (ELBO) (Gelman et al., 2013)

$$\varphi^* = \arg\max_{\varphi} \mathbb{E}_{\boldsymbol{\theta}\sim q_\varphi(\cdot)}\left[\log\frac{p(\mathcal{D},\boldsymbol{\theta})}{q_\varphi(\boldsymbol{\theta})}\right] \tag{7}$$

The above optimization procedure finds a member in the family of variational distributions $\{q_\varphi\}_\varphi$ that is closest to the true posterior under the *reverse-KL* divergence. Once the optimal parameters $\varphi^*$ are obtained, the *posterior predictive* distribution $p(\boldsymbol{x}_*|\mathcal{D})$ can be approximated as

$$p(\boldsymbol{x}_*|\mathcal{D}) \approx \mathbb{E}_{q_{\varphi^*}(\boldsymbol{\theta})}\left[p(\boldsymbol{x}_*|\boldsymbol{\theta})\right] \tag{8}$$

The family of distributions $q_\varphi$ is chosen such that it is easy to sample from. Typical choices include independent multivariate Gaussian distribution (mean-field approximation) or normalizing flows (Rezende & Mohamed, 2015; Papamakarios et al., 2021; Ardizzone et al., 2018-2022).

**Amortization**. One of the most powerful capabilities of neural networks is their ability to learn and generalize to a wide variety of domains and settings provided sufficient variability during training. For example, Variational Autoencoders (VAEs) define a latent-variable model $p(\boldsymbol{x},\boldsymbol{z})$ where $\boldsymbol{x}$ represents the observation and $\boldsymbol{z}$ the latent variable. VI typically relies on solving a separate optimization problem $q_{\varphi_i^*}(\boldsymbol{z}_i)$ for each posterior $p(\boldsymbol{z}_i|\boldsymbol{x}_i)$. The cost of learning separate variational approximations can be amortized through training of a joint network $q_\varphi(\boldsymbol{z}|\boldsymbol{x})$, where $\varphi$ now represents the parameters of a neural network which takes $\boldsymbol{x}$ explicitly as input. The VI procedure then reduces to optimizing $\varphi$, which is shared across all observations, as opposed to optimizing for separate $\varphi_i's$, in the hope that $q_\varphi(\boldsymbol{z}_i|\boldsymbol{x}_i) \approx q_{\varphi_i}(\boldsymbol{z}_i)$ for any $\boldsymbol{x}_i$. When modeling using Gaussian distributions, this distinction can be seen as $q_\varphi(\boldsymbol{z}_i|\boldsymbol{x}_i) := \mathcal{N}(\,\cdot\,;\boldsymbol{\mu}_\varphi(\boldsymbol{x}_i),\boldsymbol{\Sigma}_\varphi(\boldsymbol{x}_i))$ while $q_{\varphi_i}(\boldsymbol{z}_i) := \mathcal{N}(\,\cdot\,;\boldsymbol{\mu}_{\varphi_i},\boldsymbol{\Sigma}_{\varphi_i})$ (note the functional dependencies). In a similar fashion, Garnelo et al. (2018b) amortize on datasets as explicit inputs, while score-based generative models (Song et al., 2020) amortize on timesteps. Such models are largely successful owing to the generalization capabilities of neural networks to new unseen observations as long as the encoder $q_\varphi(\boldsymbol{z}|\boldsymbol{x}_i)$ is trained on diverse enough observations $\boldsymbol{x}_i's$.

**Simulation-Based Inference**. SBI considers the problem of inferring the parameters of the simulator from observations. This is often tackled via neural posterior estimation methods (SBI-NPE) where a deep-learning

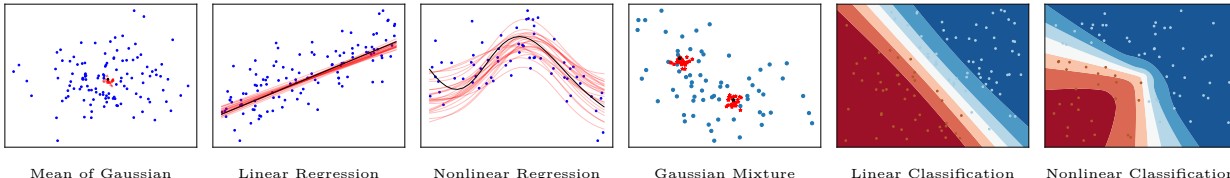

Mean of Gaussian    Linear Regression    Nonlinear Regression    Gaussian Mixture    Linear Classification    Nonlinear Classification

Figure 1: **Amortized Bayesian Posterior Estimation:** Illustration of predictions from proposed approach when trained on a fixed dimensional observation space. Model predictions, true predictions and sample points are shown in red, black and blue respectively. Additionally for classification, we label sample points with their ground-truth class, and draw the decision boundary according to the model.

based model is trained to infer the posterior by explicitly conditioning an approximate distribution $q_\varphi(\boldsymbol{\theta}|\mathcal{D})$ on the dataset, and modeling the gap between the distributions through a *Forward*-KL based optimization

$$\arg\min_{\varphi} \mathbb{E}_{\mathcal{D}} \mathbb{KL}\left[p(\boldsymbol{\theta}|\mathcal{D})||q_\varphi(\boldsymbol{\theta}|\mathcal{D})\right] \tag{9}$$

which often leads to mode averaging behavior that can be problematic in high dimensions. While the above objective often enjoys applications to tasks without tractable likelihood functions, it can only be used for training when the dataset $\mathcal{D}$ is sampled according to the probabilistic model, and thus cannot utilize off-policy non-simulated data, hindering generalization to real-world scenarios. Precisely because of this, SBI-NPE has been leveraged in controlled scenarios like modeling inverse problems with low-dimensional non-differentiable simulators where the likelihood of an observation is not tractable; but has seen limited applicability in more general high dimensional estimation problems like the distribution over weights of a Bayesian Neural Networks.

**Neural Processes**. NPs also leverage amortized VI in training a latent-variable system for modeling predictive problems. However, unlike our setup, they define an approximate posterior distribution only over an arbitrary latent space and learn how this latent variable impacts the likelihood through point estimation of likelihood parameters. In particular, NPs rely on the Variational-EM setup, where they perform point estimation for the parameters of the likelihood and VI for the latent variable. In contrast, we focus on a similar setting where we instead do a full VI treatment of the parameters of the likelihood function; which in some sense means that our *latent variables* are now parameters of the likelihood model. Thus, our approach can be seen as a fully Bayesian Inference procedure for likelihood models, whereas NPs can be seen as latent-variable models which only provide point estimates for the parameters.

**Gaussian Process Kernel Estimation**. A specific application of our framework is the estimation of the kernel function for Gaussian Process likelihood models (Liu et al., 2020; Simpson et al., 2021; Bitzer et al., 2023), which leverages amortized inference for tractable likelihoods defined by GP regression setups. In contrast, we provide a more general framework for conducting amortized inference, which we test across a wide variety of domains ranging from supervised to unsupervised learning and from regression to classification tasks. Thus, our proposed approach provides a framework for parameter estimation through amortized variational inference and GP kernel estimation along these lines is a specific application of this approach.

**In-Context Learning**. ICL refers to large pre-trained models having the ability of solving new tasks at inference based solely on some context examples being provided as prompt. A natural parallel to conditioning on context examples via prompt in natural language is explicit amortization on observations for general-purpose tasks (Von Oswald et al., 2023; Müller et al., 2021). In this sense, our approach can be seen as training a general-purpose in-context learner to perform Bayesian inference, as opposed to direct predictions.

**Sampling from Energy**. Another line of similar works (Zhang & Chen, 2021; Berner et al., 2022; Vargas et al., 2023; Akhound-Sadegh et al., 2024; Bengio et al., 2021) look at sampling proportional to a pre-specified energy function, which in our case can be seen as $p(\boldsymbol{\theta}|\mathcal{D}) \propto e^{-\mathcal{E}(\boldsymbol{\theta})}$, with $\mathcal{E}(\boldsymbol{\theta}) = -\log p(\mathcal{D}, \boldsymbol{\theta})$. However, such works currently focus on non-amortized setups and we defer exploring amortized diffusion-styled posterior models as relevant future work.

We refer the readers to Appendix A for a detailed discussion about prior work, as well as its connections and differences with our proposed mechanism.

| Objective | $q_\varphi$ | Model | $L_2$ Loss ($\downarrow$) | | | | | Accuracy ($\uparrow$) | | |
|---|---|---|---|---|---|---|---|---|---|---|
| | | | **Gaussian** | **GMM** | **LR** | **NLR** | | **LC** | **NLC** | |
| | | | *100D* | *5D 2 cl* | *100D* | *1D* | *25D* | *100D* | *2D* | *25D* |
| Baseline | - | Random | $301.06_{\pm0.35}$ | $5.00_{\pm0.04}$ | $202.6_{\pm0.3}$ | $65.94_{\pm0.91}$ | $831.6_{\pm8.7}$ | $50.0_{\pm0.0}$ | $50.3_{\pm0.6}$ | $50.0_{\pm0.3}$ |
| | - | Optimization | $101.24_{\pm0.00}$ | $0.42_{\pm0.00}$ | $25.1_{\pm0.0}$ | $0.36_{\pm0.00}$ | $104.0_{\pm0.1}$ | $70.3_{\pm0.0}$ | $96.9_{\pm0.0}$ | $77.9_{\pm0.0}$ |
| | - | Langevin | $102.35_{\pm0.03}$ | $0.45_{\pm0.01}$ | $23.3_{\pm0.7}$ | $0.31_{\pm0.00}$ | $132.4_{\pm1.0}$ | $65.1_{\pm0.4}$ | $96.0_{\pm0.3}$ | $73.2_{\pm0.3}$ |
| | - | HMC | $102.41_{\pm0.03}$ | $0.48_{\pm0.01}$ | $18.7_{\pm0.2}$ | $0.37_{\pm0.00}$ | $98.1_{\pm0.7}$ | $62.1_{\pm0.2}$ | $91.8_{\pm0.2}$ | $70.4_{\pm0.1}$ |
| Fwd-KL | Gaussian | GRU | $102.64_{\pm0.01}$ | $2.43_{\pm0.03}$ | $124.8_{\pm0.1}$ | $49.33_{\pm0.95}$ | $671.6_{\pm10.5}$ | $59.7_{\pm0.1}$ | $59.5_{\pm0.4}$ | $56.9_{\pm0.3}$ |
| | | DeepSets | $103.22_{\pm0.05}$ | $2.44_{\pm0.04}$ | $123.1_{\pm1.1}$ | $49.86_{\pm0.98}$ | $684.9_{\pm2.6}$ | $50.0_{\pm0.1}$ | $59.4_{\pm0.2}$ | $56.8_{\pm0.2}$ |
| | | Transformer | $102.78_{\pm0.00}$ | $2.50_{\pm0.03}$ | $45.9_{\pm1.3}$ | $49.68_{\pm0.94}$ | $680.9_{\pm5.8}$ | $63.0_{\pm0.1}$ | $59.6_{\pm0.4}$ | $57.1_{\pm0.4}$ |
| Fwd-KL | | GRU | $102.51_{\pm0.01}$ | $0.47_{\pm0.01}$ | $60.2_{\pm0.9}$ | $0.43_{\pm0.00}$ | $106.0_{\pm0.6}$ | $63.5_{\pm0.3}$ | $92.4_{\pm0.2}$ | $72.5_{\pm0.0}$ |
| | | DeepSets | $102.60_{\pm0.04}$ | $0.50_{\pm0.02}$ | $62.8_{\pm0.6}$ | $0.43_{\pm0.00}$ | $125.9_{\pm0.8}$ | $60.9_{\pm0.3}$ | $92.5_{\pm0.1}$ | $59.8_{\pm0.3}$ |
| | | Transformer | $102.54_{\pm0.03}$ | $0.49_{\pm0.02}$ | $28.7_{\pm0.3}$ | $0.42_{\pm0.01}$ | $102.3_{\pm1.8}$ | $68.2_{\pm0.0}$ | $92.6_{\pm0.4}$ | $75.2_{\pm0.1}$ |
| Fwd-KL | Flow | GRU | $102.66_{\pm0.02}$ | $0.67_{\pm0.09}$ | $119.1_{\pm0.2}$ | $15.78_{\pm0.21}$ | $539.0_{\pm4.3}$ | $59.9_{\pm0.2}$ | $76.9_{\pm0.3}$ | $58.3_{\pm0.0}$ |
| | | DeepSets | $103.34_{\pm0.03}$ | $0.65_{\pm0.08}$ | $125.7_{\pm3.7}$ | $15.05_{\pm0.12}$ | $548.5_{\pm3.3}$ | $50.1_{\pm0.0}$ | $72.3_{\pm1.8}$ | $58.1_{\pm0.1}$ |
| | | Transformer | $102.77_{\pm0.02}$ | $0.62_{\pm0.07}$ | $43.3_{\pm2.7}$ | $16.11_{\pm0.31}$ | $539.3_{\pm4.3}$ | $64.3_{\pm0.1}$ | $77.3_{\pm0.2}$ | $58.3_{\pm0.1}$ |
| Rev-KL | | GRU | $102.49_{\pm0.01}$ | $0.47_{\pm0.00}$ | $61.3_{\pm1.0}$ | $0.41_{\pm0.01}$ | $106.0_{\pm0.4}$ | $64.7_{\pm0.2}$ | $93.4_{\pm0.1}$ | $72.0_{\pm0.5}$ |
| | | DeepSets | $102.67_{\pm0.05}$ | $0.52_{\pm0.01}$ | $76.4_{\pm2.0}$ | $0.40_{\pm0.00}$ | $128.2_{\pm1.5}$ | $58.4_{\pm0.8}$ | $93.3_{\pm0.2}$ | $60.9_{\pm0.2}$ |
| | | Transformer | $102.53_{\pm0.05}$ | $0.47_{\pm0.01}$ | $29.4_{\pm1.6}$ | $0.39_{\pm0.00}$ | $102.6_{\pm0.9}$ | $68.7_{\pm0.1}$ | $93.6_{\pm0.1}$ | $75.0_{\pm0.5}$ |

Table 1: **Amortized Bayesian Posterior Estimation**. Results for estimating the mean of a Gaussian (Gaussian), means of a Gaussian mixture model (GMM), (non-)linear regression (NLR/LR) and (non-)linear binary classification (NLC/LC). We ablate over different architectures and density parameterizations. Our baselines include the prior (Random), dataset-specific maximum likelihood training (Optimization), Langevin and Hamiltonian MCMC and Forward-KL based SBI. We use the expected $L_2$ loss and accuracy according to the posterior predictive as metrics.

## 3 Method

Given any modeling assumption, we are interested in estimating the posterior distribution over its parameters conditioned explicitly over observations to allow for a fast and scalable approximation. That is, given a probabilistic model $p(\cdot|\boldsymbol{\theta})$ with parameters $\boldsymbol{\theta}$, we are interested in approximating the full Bayesian inference solution over $\boldsymbol{\theta}$. Our goal is to train a system that approximates the posterior distribution $p(\boldsymbol{\theta}|\mathcal{D})$ given a dataset $\mathcal{D} := \{\boldsymbol{x}_1, \boldsymbol{x}_2, \cdots, \boldsymbol{x}_n\} \subseteq \mathbb{R}^d$ where $\boldsymbol{x}_i \sim p(\boldsymbol{x}|\boldsymbol{\theta})$. We achieve this by learning an amortized variational distribution $q_\varphi(\boldsymbol{\theta}|\mathcal{D})$ conditioned explicitly on the dataset $\mathcal{D}$. Similar to standard VI approaches, we can train $q_\varphi$ through the following optimization problem

$$\arg\min_{\varphi} \mathbb{KL}\left[q_\varphi(\cdot|\mathcal{D})||p(\cdot|\mathcal{D})\right] \tag{10}$$

While this is the case for VI on a single dataset, we are interested in generalizing to a family of datasets $\chi$. Similar to how a VAE encoder efficiently generalizes to approximate posteriors for new images when trained across multiple image observations, we amortize over datasets by training over multiple $\mathcal{D} \sim \chi$ in the hope of generalizing to posterior approximations on new datasets. Thus, given a probability measure over the space of datasets $\chi$, we posit learning of shared parameters $\varphi$ of the variational distribution across different datasets as

$$\arg\min_{\varphi} \mathbb{E}_{\mathcal{D}\sim\chi} \mathbb{KL}\left[q_\varphi(\boldsymbol{\theta}|\mathcal{D})||p(\boldsymbol{\theta}|\mathcal{D})\right] \tag{11}$$

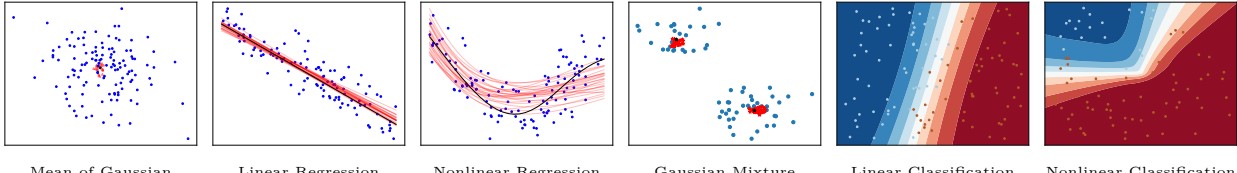

Mean of Gaussian     Linear Regression     Nonlinear Regression     Gaussian Mixture     Linear Classification     Nonlinear Classification

Figure 2: **Variable-Dimension Visualization:** Illustration of proposed approach when trained on a variable observational space but visualized on low-dimensional tasks, with model predictions, ground truth prediction and data points in red, black and blue respectively. For classification, data points are colored by their ground-truth labels, and decision boundary corresponds to the model prediction.

which equivalently reduces to maximizing the ELBO:

$$\arg\max_{\varphi} \mathbb{E}_{\mathcal{D}\sim\chi}\mathbb{E}_{\boldsymbol{\theta}\sim q_{\varphi}(\cdot|\mathcal{D})}\left[\log\frac{p(\mathcal{D},\boldsymbol{\theta})}{q_{\varphi}(\boldsymbol{\theta}|\mathcal{D})}\right] \tag{12}$$

To recap, while VI strategies approach this problem by learning a separate $q_{\varphi}$ for each $\mathcal{D}$, we instead train a shared $q_{\varphi}$ that explicitly takes $\mathcal{D}$ as input and, in doing so, possesses the ability to generalize to new datasets without additional training.

This training paradigm naturally introduces a dependency on the dataset generating distribution $\chi$. Since we are working with a known probabilistic model, an obvious choice of $\chi$ is to treat this probabilistic model as a black-box simulator, akin to $p(n)p(\boldsymbol{\theta})\prod_{i=1}^{n}p(\boldsymbol{x}_i|\boldsymbol{\theta})$, samples from which can be obtained using ancestral sampling. Here, $n$ is the dataset cardinality and $p(n)$ is a distribution over positive integers. Thus, obtaining at least one dataset-generating distribution is easy given any probabilistic model. However, $\chi$ can also be obtained from other sources, for example, a stream of real-world data, through interventions in the data-generating process, or through bagging on a large real-world dataset.

Another design choice in the above setup is the parameterization for $q_{\varphi}$: we explore a Gaussian distribution and a normalizing flow parameterization with either a Transformer, GRU or DeepSets architectural backbone to process $\mathcal{D}$. It is important to note that not all deep learning architectures (eg. GRU) are amenable in this setting since, given the *iid* nature of the samples, the posterior is permutation invariant

$$p(\boldsymbol{\theta}|\mathcal{D}) = p(\boldsymbol{\theta}|\Pi\mathcal{D}) \tag{13}$$

for some permutation matrix $\Pi$. This constraint should thus be reflected when modeling the approximation. To satisfy it, for example for a Gaussian distribution, we can define $q_{\varphi}$ as

$$q_{\varphi}(\cdot) = \mathcal{N}\left(\cdot|\boldsymbol{\mu}_{\varphi}(\mathcal{D}), \boldsymbol{\Sigma}_{\varphi}(\mathcal{D})\right) \tag{14}$$

where $\boldsymbol{\mu}_{\varphi}$ and $\boldsymbol{\Sigma}_{\varphi}$ are modeled using permutation invariant architectures, thus satisfying the desired constraint

$$q_{\varphi}(\cdot|\mathcal{D}) = q_{\varphi}(\cdot|\Pi\mathcal{D}) \tag{15}$$

Finally, another important design choice in this setup is the prior $p(\boldsymbol{\theta})$. In this work, we assume it to be known and fixed to $\mathcal{N}(\cdot; 0, \mathbf{I})$, which is a common assumption across various problems. Our proposed amortized VI approach, in contrast to SBI-NPE which relies on *Forward KL* optimization, relies on the *Reverse KL* objective.[1] Finally, we defer details regarding the architectural choices to Appendix B.

## 3.1 Amortizing Variable Feature Dimensions

So far, we have only considered amortization over different datasets for the same underlying likelihood model. For example, for a 2-dimensional Bayesian linear regression model, the amortized posterior $q_{\varphi}(\boldsymbol{\theta}|\mathcal{D})$ approximates the true posterior distribution $p(\boldsymbol{\theta}|\mathcal{D})$ for arbitrary sets of 2-dimensional observations. It is important to note that a deep learning-based approach leaves hopes of generalizing to new datasets since the underlying functional form of the solution remains constant across different datasets, irrespective of the number of features, and is given by the solution obtained from Equation 4.

---

[1]Given the equivalences, we will use amortized VI and Reverse KL interchangeably.

| Objective | $q_\varphi$ | Model | $L_2$ Loss ($\downarrow$) | | | | | Accuracy ($\uparrow$) | | |
|---|---|---|---|---|---|---|---|---|---|---|
| | | | Gaussian | GMM | LR | NLR | | LC | NLC | |
| | | | *100D* | *5D 2 cl* | *100D* | *1D* | *50D* | *100D* | *2D* | *50D* |
| Baseline | - | Random | $298.24_{\pm0.23}$ | $4.66_{\pm0.03}$ | $200.8_{\pm0.6}$ | $73.01_{\pm0.17}$ | $1704.3_{\pm9.3}$ | $50.0_{\pm0.1}$ | $50.0_{\pm0.3}$ | $49.9_{\pm0.3}$ |
| | - | Optimization | $100.88_{\pm0.00}$ | $0.43_{\pm0.00}$ | $20.1_{\pm0.0}$ | $0.36_{\pm0.00}$ | $309.2_{\pm0.2}$ | $71.2_{\pm0.0}$ | $96.8_{\pm0.0}$ | $76.1_{\pm0.0}$ |
| | - | Langevin | $101.92_{\pm0.04}$ | $0.44_{\pm0.00}$ | $21.8_{\pm1.0}$ | $0.31_{\pm0.00}$ | N/A | $65.5_{\pm0.5}$ | $96.1_{\pm0.0}$ | $70.1_{\pm0.2}$ |
| | - | HMC | $102.01_{\pm0.01}$ | $0.46_{\pm0.01}$ | $17.8_{\pm0.1}$ | $0.38_{\pm0.01}$ | $303.9_{\pm2.5}$ | $62.6_{\pm0.2}$ | $91.7_{\pm0.2}$ | $68.0_{\pm0.4}$ |
| Fwd-KL | Gaussian | GRU | $133.22_{\pm0.58}$ | $2.36_{\pm0.02}$ | $139.4_{\pm1.0}$ | $51.45_{\pm0.03}$ | $1346.5_{\pm6.8}$ | $57.9_{\pm0.2}$ | $59.6_{\pm0.2}$ | $58.6_{\pm0.2}$ |
| | | DeepSets | $129.69_{\pm0.74}$ | $2.35_{\pm0.02}$ | $149.8_{\pm0.8}$ | $51.90_{\pm1.54}$ | $1357.5_{\pm5.3}$ | $50.8_{\pm0.1}$ | $49.9_{\pm0.3}$ | $49.9_{\pm0.3}$ |
| | | Transformer | $108.98_{\pm0.10}$ | $2.40_{\pm0.02}$ | $64.3_{\pm3.7}$ | $50.81_{\pm0.53}$ | $1319.9_{\pm12.2}$ | $62.4_{\pm0.0}$ | $59.9_{\pm0.2}$ | $58.8_{\pm0.2}$ |
| Rev-KL | | GRU | $105.14_{\pm0.10}$ | $0.46_{\pm0.01}$ | $62.6_{\pm0.1}$ | $2.31_{\pm0.13}$ | $316.3_{\pm6.2}$ | $63.4_{\pm0.2}$ | $88.8_{\pm0.5}$ | $68.4_{\pm0.3}$ |
| | | DeepSets | $105.06_{\pm0.21}$ | $0.48_{\pm0.02}$ | $64.1_{\pm0.2}$ | $0.98_{\pm0.12}$ | $451.9_{\pm2.8}$ | $61.3_{\pm0.1}$ | $91.0_{\pm0.5}$ | $61.7_{\pm0.1}$ |
| | | Transformer | $104.71_{\pm0.12}$ | $0.47_{\pm0.01}$ | $32.0_{\pm0.5}$ | $0.81_{\pm0.02}$ | $278.3_{\pm1.1}$ | $67.7_{\pm0.1}$ | $90.0_{\pm0.2}$ | $73.7_{\pm0.3}$ |
| Fwd-KL | Flow | GRU | $125.84_{\pm1.98}$ | $0.60_{\pm0.07}$ | $138.3_{\pm1.0}$ | $38.41_{\pm0.36}$ | $1097.4_{\pm9.5}$ | $58.0_{\pm0.1}$ | $61.2_{\pm0.8}$ | $60.2_{\pm0.1}$ |
| | | DeepSets | $133.23_{\pm1.93}$ | $0.58_{\pm0.03}$ | $153.2_{\pm0.8}$ | $43.31_{\pm2.06}$ | $1120.0_{\pm5.5}$ | $50.5_{\pm0.1}$ | $49.6_{\pm0.2}$ | $50.1_{\pm0.1}$ |
| | | Transformer | $108.48_{\pm0.16}$ | $0.59_{\pm0.08}$ | $63.1_{\pm2.0}$ | $39.70_{\pm0.52}$ | $1073.3_{\pm1.5}$ | $63.6_{\pm0.1}$ | $60.9_{\pm0.3}$ | $60.3_{\pm0.1}$ |
| Rev-KL | | GRU | $105.19_{\pm0.03}$ | $0.47_{\pm0.01}$ | $71.3_{\pm1.3}$ | $2.31_{\pm0.41}$ | $302.9_{\pm5.6}$ | $63.4_{\pm0.1}$ | $90.4_{\pm0.7}$ | $66.2_{\pm0.1}$ |
| | | DeepSets | $105.09_{\pm0.06}$ | $0.49_{\pm0.01}$ | $76.8_{\pm1.8}$ | $0.83_{\pm0.02}$ | $454.1_{\pm10.2}$ | $59.1_{\pm0.5}$ | $89.1_{\pm0.3}$ | $62.9_{\pm0.1}$ |
| | | Transformer | $104.91_{\pm0.11}$ | $0.46_{\pm0.00}$ | $33.1_{\pm0.3}$ | $0.99_{\pm0.07}$ | $274.0_{\pm1.3}$ | $68.1_{\pm0.2}$ | $91.1_{\pm0.2}$ | $72.6_{\pm0.1}$ |

Table 2: **Variable-Dimension Posterior Prediction**. Results for estimating the mean of a Gaussian (Gaussian), means of a Gaussian mixture model (GMM), (non-)linear regression (NLR/LR) and (non-)linear binary classification (NLC/LC). In this setting, for each task only a single model was trained to solve for different dimensional problems; e.g. the same model is used to evaluate *1D* and *50D* nonlinear regression.

We note that the form of the solution remains the same irrespective of whether the underlying problem is a 2- or 100-dimensional Bayesian linear regression problem. Additionally, we can see that a low-dimensional problem can just be embedded into a high-dimensional space, with the extra features and parameters set to 0, akin to the procedure of masking unnecessary dimensions. This simple but strong insight allows us to amortize $q_\varphi$ over datasets with varying dimensionalities by embedding them in a higher dimensional space.

## 3.2 Handling Model Misspecification

Knowledge about the true likelihood model underlying the data-generating scheme is often unknown. Practitioners thus rely on making a modeling assumption by prescribing a particular likelihood model for the data and then fitting its parameters to best explain the data. For example, the underlying model of whether an email is spam or not may be unknown, but one can still assume a linear model to solve this classification problem. This defines the basis of model misspecification, where there is a mismatch between the underlying true model and the assumption used.

In contrast to SBI-NPE approaches, our method can efficiently handle model misspecification. This can be seen from the contrast between Equations 9 and 12, where the former can only be trained when $\chi$ is defined according to the modeling choice while the latter doesn't put any constraints on $\chi$. Thus, in the presence of a stream of real-world data, SBI-NPE cannot leverage this data during training because the corresponding parameters in the assumed likelihood model are not known, while our method can leverage it during training and can thus lead to more robust predictions on data that is of actual practical significance.

A simple example of this setup is when the true data might be coming from a Gaussian Process, but we might not know this. To still be able to efficiently make predictions, we can model it with a linear relation. However, SBI-NPE approaches can only train the amortized posterior for the linear model using linear data,

| $\chi_{real}$ ($\rightarrow$) | $q_\varphi$ | *Data* | Linear | MLP Nonlinear | GP Nonlinear |
|---|---|---|---|---|---|
| $\chi_{sim}$ ($\rightarrow$) | | *Model* | NLR | LR | NLR |
| Fwd-KL | Gaussian | | $15.454_{\pm 0.246}$ | $2.216_{\pm 0.097}$ | $14.733_{\pm 0.513}$ |
| Rev-KL | | | $0.382_{\pm 0.003}$ | $1.892_{\pm 0.113}$ | $0.155_{\pm 0.006}$ |
| *+ switched data* | | | $0.367_{\pm 0.006}$ | $1.226_{\pm 0.001}$ | $0.066_{\pm 0.004}$ |
| Fwd-KL | Flow | | $7.949_{\pm 0.419}$ | $1.632_{\pm 0.070}$ | $8.557_{\pm 0.561}$ |
| Rev-KL | | | $0.347_{\pm 0.001}$ | $1.471_{\pm 0.016}$ | $0.120_{\pm 0.005}$ |
| *+ switched data* | | | $0.346_{\pm 0.002}$ | $1.226_{\pm 0.004}$ | $0.055_{\pm 0.002}$ |

Table 3: **Model Misspecification:** We see benefits of our proposed approach when the true underlying data generating process is not known. $\chi_{real}$ denotes the real data-generating distribution, $\chi_{sim}$ the simulated one according to the (wrongly) assumed model. We train different models on $\chi_{sim}$, with switched data denoting training on $\chi_{real}$. Evaluation is done solely on $\chi_{real}$.

while our method can actually leverage the GP data that we observe. This leads to better generalization to linear modeling of GP data and can be further leveraged in other model misspecification setups.

## 4  Experiments

To showcase the wide applicability of our approach, we perform experiments on various well-known probabilistic models encompassing supervised and unsupervised scenarios. In particular, we look at the problem of estimating the Bayesian posterior over the (a) mean of a Gaussian distribution (GM), (b) means of a Gaussian mixture model (GMM), (c) parameters of a (non-)linear regression model (NLR/LR), and (d) parameters of a (non-)linear classification model (NLC/LC). We refer the readers to Appendix C for particulars about the probabilistic models, including their likelihoods and priors considered. Throughout our experiments, we observe superior performance of our proposed approach in terms of predictive performance metrics, especially in problems with high-dimensional and multi-modal posteriors.

We generally consider two types of baselines: dataset-specific and amortized. For dataset-specific baselines, we use the prior (Random), perform maximum likelihood estimation using gradient-based optimization (Optimization) as well as an approximate Bayesian inference procedure through Langevin and Hamiltonian based MCMC sampling, which also uses the gradient information. Such baselines rely on iterative procedures and must be run independently for different datasets. For amortized baselines, we consider the SBI-NPE based forward-KL baseline as outlined in Equation 9. We refer the readers to Appendices F, D, and H for details about the experiments, metrics, and additional results, respectively.

**Zero-Shot Posterior Approximation**.

Given a known probabilistic model $p(\mathcal{D}, \boldsymbol{\theta})$, we train an amortized inference system $q_\varphi(\cdot|\mathcal{D})$ to approximate the often intractable posterior over the parameter $\boldsymbol{\theta}$ for different datasets with varying cardinality. Figure 1 visualizes the performance of the amortized model on different probabilistic models spanning supervised and unsupervised learning for low dimensional problems. We see that the proposed approach provides reliable samples of the parameters for previously unseen datasets, showing that it can indeed generalize.

Next, we empirically evaluate the performance of our proposed approach in more complex, high-dimensional setups for the same set of probabilistic models. We set the data-generating distribution $\chi$ to be according to the probabilistic model in order to allow comparisons with SBI-NPE, which can only be trained in this particular setting. Table 1 highlights the performance of our proposed approach in high-dimensional frameworks, showcasing its superior performance to SBI-NPE benchmarks, as well as comparative to dataset-specific non-amortized baselines. For multiple high-dimensional problems, we see that the proposed amortized posterior performs similarly to Optimization and MCMC without being slow and iterative at inference.

**Generalizing to Variable Feature Dimensions**

| Objective | $q_\varphi$ | Model | $L_2$ Loss ($\downarrow$) | | Accuracy ($\uparrow$) | |
| | | | LR | NLR | LC | NLC |
|---|---|---|---|---|---|---|
| Baseline | - | Random | $23.52_{\pm0.42}$ | $209.35_{\pm9.92}$ | $50.10_{\pm0.17}$ | $50.84_{\pm1.00}$ |
| Fwd-KL | Gaussian | GRU | $8.95_{\pm0.47}$ | $84.63_{\pm3.96}$ | $76.34_{\pm1.58}$ | $60.00_{\pm0.98}$ |
| | | DeepSets | $10.81_{\pm0.08}$ | $97.51_{\pm2.96}$ | $68.26_{\pm0.31}$ | $50.84_{\pm0.99}$ |
| | | Transformer | $9.35_{\pm0.99}$ | $111.07_{\pm6.55}$ | $63.97_{\pm3.21}$ | $60.39_{\pm0.53}$ |
| Rev-KL | | GRU | $9.78_{\pm2.75}$ | $17.01_{\pm5.16}$ | $79.71_{\pm1.12}$ | $77.19_{\pm0.21}$ |
| | | DeepSets | $9.26_{\pm0.10}$ | $8.03_{\pm0.23}$ | $77.40_{\pm0.14}$ | $72.63_{\pm0.14}$ |
| | | Transformer | $7.30_{\pm0.21}$ | $8.23_{\pm1.26}$ | $76.96_{\pm1.58}$ | $71.33_{\pm5.44}$ |
| Fwd-KL | Flow | GRU | $8.28_{\pm0.33}$ | $52.04_{\pm3.39}$ | $76.45_{\pm1.43}$ | $61.10_{\pm0.61}$ |
| | | DeepSets | $12.94_{\pm0.41}$ | $71.69_{\pm2.38}$ | $67.99_{\pm2.15}$ | $49.74_{\pm0.76}$ |
| | | Transformer | $9.64_{\pm0.50}$ | $84.45_{\pm7.88}$ | $68.26_{\pm3.23}$ | $61.65_{\pm1.22}$ |
| Rev-KL | | GRU | $8.17_{\pm0.25}$ | $17.35_{\pm8.02}$ | $66.30_{\pm2.81}$ | $78.55_{\pm1.27}$ |
| | | DeepSets | $11.05_{\pm0.35}$ | $8.54_{\pm0.49}$ | $78.18_{\pm0.13}$ | $71.69_{\pm0.18}$ |
| | | Transformer | $7.48_{\pm0.26}$ | $8.80_{\pm1.80}$ | $72.75_{\pm5.73}$ | $78.23_{\pm0.89}$ |

Table 4: **Tabular Experiments**: Zero-shot performance of the amortized variable-dimensional models across (non-)linear regression and classification real-world tabular tasks. All the models are trained solely on simulated data, and evaluated zero-shot on real-world data with varying number of both features and training (observations that are fed as in-context amortization) observations.

Next, we turn our attention to performing amortization additionally over different datasets *with a variable number of feature dimensions*, as highlighted in Section 3.1. To do this, we embed all low-dimensional problems in a 100-dimensional space by masking the unnecessary dimensions, and then train an amortized VI system. A key benefit of this approach is that instead of having to train different models for different dimensional problems, we can rely on a single model that can generalize across all. We visualize samples from the approximate posterior for low-dimensional problems in Figure 2 and further provide quantitative predictive metrics for high-dimensional problems in Table 2. Our results highlight that the proposed approach can generalize quite well to variable dimensional setups without requiring an iterative procedure at inference.

**Handling Model Misspecification**

As discussed in Section 3 and prior work (Müller et al., 2021; Hollmann et al., 2022), the performance and use-case of such amortized systems relies heavily on the choice of the data-generating distribution $\chi$. So far, we only look at cases where $\chi$ defines simulations according to the probabilistic model to allow for a fair comparison with SBI-NPE. This setup, however, implicitly assumes that we know the ground-truth data-generating process, which is often not the case. A more common scenario is where we do not know the data-generating process but have access to samples from it defined through $\chi_{real}$. To make predictions for new query points, we, as statisticians, assume a probabilistic model $p(\boldsymbol{x}|\boldsymbol{\theta})$ to explain the data, which may be quite far from the true underlying model.

We first note that $\chi_{real}$ cannot be used to train a forward KL model (SBI-NPE) and is thus unusable in this case. To resolve this issue, let $\chi_{sim}$ denote the data generating distribution obtained from simulations via the assumed model, which can then be used to train the forward KL model. We compare this setup with our proposed approach, which can be trained on either $\chi_{sim}$ or $\chi_{real}$. However, for this setup, we always perform evaluation on the real data of interest, i.e. $\chi_{real}$.

For our experiments, we pick $\chi_{real}$ and $\chi_{sim}$ by simulation through probabilistic models corresponding to linear, MLP, and Gaussian Process regression, such that $\chi_{real}$ and $\chi_{sim}$ are never the same. Table 3 showcases experimental results of model misspecification, where we see that the forward KL models suffer a lot due to mismatch between the training and evaluation data, compounded by the potential overestimation of variance (mode-averaging) while our proposed method significantly outperforms it. Even further, we see that the proposed method can be directly trained on $\chi_{real}$ (switched data), leading to even better performance — for eg., when $\chi_{real}$ denotes data generated from a Gaussian Process and the probabilistic model assumed

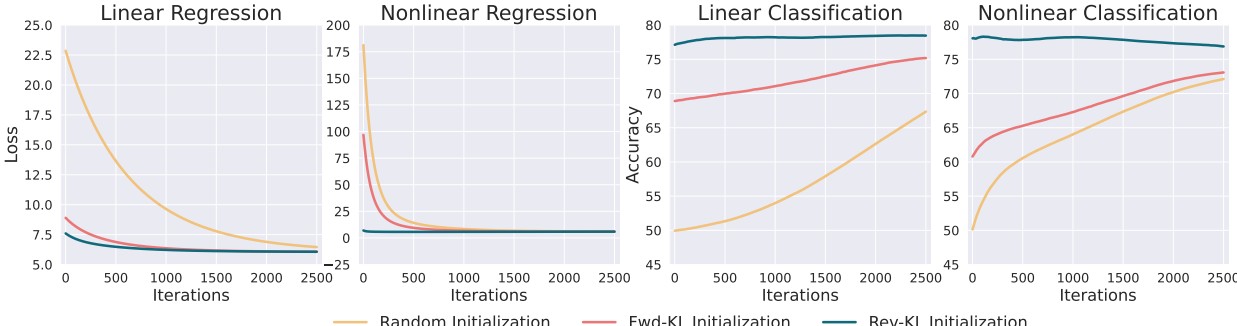

Figure 3: **Tabular Experiments:** Initializing parameters from the proposed amortized model leads to good zero-shot performance and often optimal initialization across (non-)linear regression and classification tasks. is MLP-based (last column), our proposed approach outperforms forward KL when trained on the same simulated data from the MLP, and does even better when trained on the GP-based data. We refer the readers to Appendix F.3 and H.3 for experimental details and additional results on model misspecification.

**Applications to Tabular Domains**

While we see clear benefits in simulated settings, we next turn our attention to real-world scenarios and consider a suite of tasks from the OpenML platform for both regression and binary classification by filtering them out from the *OpenML-CTR23 - A curated tabular regression benchmarking suite* (Fischer et al., 2023) for regression and *OpenML-CC18 Curated Classification benchmark* (Bischl et al., 2019) for classification problems (details in Appendix G). We end up with 9 regression and 13 classification datasets with varying number of features and use our amortized inference systems trained on simulated data with variable feature dimensions to predict the parameters of interest.

After initializing from the inference model, we further train the parameters of the respective probabilistic models with maximum a-posteriori (*map*) objective and compare the performance with a corresponding model initialized from the prior. Even in this extreme case of domain shift, we see in Table 4 that the amortized model provides good initializations for different real-world tasks zero-shot after training only on simulated data. Figure 3 also shows that the amortized model surpasses the SBI-NPE baseline in terms of performance and convergence speed for both linear and nonlinear setups. While Figure 3 only provides a normalized aggregated performance over all datasets considered for a $q_\varphi$ parameterized as a diagonal Gaussian, we refer the readers to Appendix H.4 for results on individual datasets with different $q_\varphi$, as well as Appendix F.4 for implementation details.

**Evaluating Posterior Inference**

While comparing with the true posterior is hard due to its intractability, it is still available for the problem of estimating the mean of a Gaussian distribution as well as for Bayesian Linear Regression. We plot the kernel density estimate of the samples obtained from the true posterior, the amortized forward, and the reverse KL model in Figure 4 (**Right**), which shows that both the amortization setups efficiently capture the true posterior distribution. Further, we quantify the quality of the approximate posteriors through a symmetric KL Divergence metric in Table 5, showing that the amortized model obtains good approximations but does worsen with increasing dimensionality, with reverse KL modeling outperforming SBI-NPE on high dimensional linear regression.

| $q_\varphi$ | | Model | Symmetric KL Divergence ($\downarrow$) | | | |
| --- | --- | --- | --- | --- | --- | --- |
| | | | **Gaussian Mean** | | **LR** | |
| | | | *2D* | *100D* | *1D* | *100D* |
| Baseline | - | Random | 44.32 | 46.78 | 179.2 | 186.8 |
| Fwd-KL | Gaussian | GRU | $0.018_{\pm0.007}$ | $0.07_{\pm0.00}$ | $0.04_{\pm0.01}$ | $81.8_{\pm0.2}$ |
| | | DeepSets | $0.037_{\pm0.015}$ | $0.22_{\pm0.01}$ | $0.06_{\pm0.00}$ | $82.0_{\pm0.4}$ |
| | | Transformer | $0.030_{\pm0.008}$ | $0.06_{\pm0.00}$ | $0.04_{\pm0.01}$ | $20.6_{\pm1.0}$ |
| Rev-KL | | GRU | $0.017_{\pm0.005}$ | $0.08_{\pm0.00}$ | $0.03_{\pm0.00}$ | $78.7_{\pm1.5}$ |
| | | DeepSets | $0.029_{\pm0.002}$ | $0.19_{\pm0.01}$ | $0.05_{\pm0.00}$ | $104.5_{\pm8.8}$ |
| | | Transformer | $0.035_{\pm0.013}$ | $0.05_{\pm0.00}$ | $0.03_{\pm0.00}$ | $32.7_{\pm1.0}$ |

Table 5: **Normalized Symmetric KL Divergence**. Amortized models approximate the true posterior well in tasks with tractable posteriors, when compared to the prior .

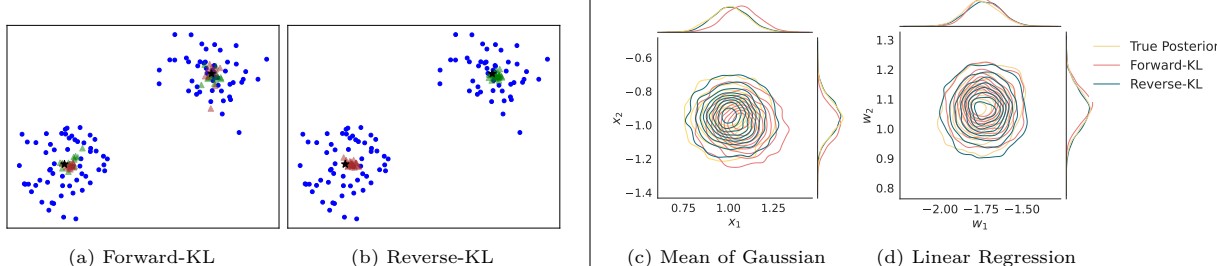

| (a) Forward-KL | (b) Reverse-KL | (c) Mean of Gaussian | (d) Linear Regression |

Figure 4: **Left**: Estimation of the means of a GMM, where red and green samples denote the first and second mean vectors. Unlike in reverse KL, the cluster labels switch in forward KL, highlighting its ability to capture underlying multi-modality. **Right**: Kernel density estimation of the true posterior, overlaid with estimates from forward and reverse KL systems, for different probabilistic models.

For more complex problems, we additionally compute the squared wasserstein metric $\mathcal{W}_2^2$ between samples from the amortized approximate posterior and from multiple chains of Langevin MCMC. Table 6 shows that reverse KL approaches do slightly better in high-dimensional setups, while for low-dimensional multi-modal scenarios (eg. GMM), forward KL approaches fare better; highlighting that they suffer less from mode collapse. However, it is imperative to know that this metric only provides a crude proxy to the quality of the posterior, since MCMC methods only provide guarantees of true samples asymptotically.

## 5 Discussion

Having presented a method to amortize posterior estimation for different probabilistic models and the corresponding empirical evidence, we now highlight some of our key findings.

**Posterior Inference**. Our results indicate that the proposed amortized Bayesian posterior estimate generalizes quite well zero-shot to novel tasks without the need for fine-tuning. While we see that in low-dimensional multi-modal settings, forward KL performs better at capturing multiple modes, it fails to capture complex posteriors in high dimensions. Our posterior predictive results additionally indicate that forward KL often performs comparable to random chance (prior) in high-dimensional, multi-modal setups. In contrast, reverse KL provides reasonable estimates, alluding to its ability to better capture at least a mode of the posterior and highlights its superiority and usability when modeling predictions.

**Architecture Choices**. To better understand the architectural inductive biases that lead to better posterior estimates, we compare permutation invariant architectures like DeepSets and Transformers with non invariant architecture like GRU. We first note that GRU outperforms DeepSets even though the latter is permutation invariant, which can be explained by the fact that the former can still learn invariant structures after seeing a lot of datasets during training while the latter is restrictive in its invariance due to the choice of aggregation operator, which we keep as mean in our setting. On the other hand, we see that Transformer outperforms both DeepSets and GRUs as it does not rely on a fixed aggregation scheme (e.g. sum or mean pooling) while still respecting the invariant structure of the posterior and learning it in a context-aware fashion. However, we do see that in some rare cases of OoD generalization, DeepSets and GRUs can still outperform Transformers (Appendix H.3).

**Capacity of $q_\varphi$**. Next, we see that increasing the capacity of $q_\varphi$ with normalizing flows only helps marginally for the reverse KL objective but substantially for the forward KL setup. We hypothesize that given the mode-seeking tendency of reverse KL, even with the capacity to model different modes, the algorithm seeks and latches to only a single mode and capturing multiple modes in this setup is challenging, whereas in forward KL setup without additional capacity the model overestimates the variance a lot.

**Forward vs Reverse KL**. Our experiments on GMM show that forward KL objective leads to the learning of a multi-modal distribution while reverse KL only captures one mode (Figure 4, **Left**). However, in high-dimensional multi-modal settings like BNNs, the former does not lead to learning of a reasonable distribution as it attempts to cover all the modes while the latter does not cover multiple modes but better models an individual mode (Tables 1 and 2; Appendix H.1 and H.2). Furthermore, unlike forward KL, the reverse KL

| Objective | $q_\varphi$ | Model | $\mathcal{W}_2^2$ $(\downarrow)$ | | | | | | | |
|---|---|---|---|---|---|---|---|---|---|---|
| | | | Gaussian | GMM | LR | NLR | | LC | NLC | |
| | | | *100D* | *5D 2 cl* | *100D* | *1D* | *25D* | *100D* | *2D* | *25D* |
| Baseline | - | Random | $13.96_{\pm0.00}$ | $4.01_{\pm0.00}$ | $13.53_{\pm0.00}$ | $11.21_{\pm0.00}$ | $36.46_{\pm0.00}$ | $16.72_{\pm0.00}$ | $14.71_{\pm0.00}$ | $36.67_{\pm0.00}$ |
| Fwd-KL | Gaussian | GRU | $1.37_{\pm0.00}$ | $2.35_{\pm0.01}$ | $10.42_{\pm0.02}$ | $11.10_{\pm0.00}$ | $36.33_{\pm0.01}$ | $15.25_{\pm0.01}$ | $14.67_{\pm0.00}$ | $36.66_{\pm0.00}$ |
| | | DeepSets | $1.55_{\pm0.01}$ | $2.35_{\pm0.01}$ | $10.39_{\pm0.03}$ | $11.11_{\pm0.01}$ | $36.41_{\pm0.01}$ | $16.72_{\pm0.00}$ | $14.68_{\pm0.00}$ | $36.66_{\pm0.00}$ |
| | | Transformer | $1.41_{\pm0.00}$ | $2.40_{\pm0.01}$ | $5.83_{\pm0.09}$ | $11.09_{\pm0.01}$ | $36.32_{\pm0.01}$ | $14.70_{\pm0.01}$ | $14.67_{\pm0.00}$ | $36.66_{\pm0.00}$ |
| Rev-KL | | GRU | $1.34_{\pm0.00}$ | $2.98_{\pm0.01}$ | $7.39_{\pm0.03}$ | $11.31_{\pm0.01}$ | $35.92_{\pm0.32}$ | $12.31_{\pm0.01}$ | $14.14_{\pm0.01}$ | $35.16_{\pm0.00}$ |
| | | DeepSets | $1.38_{\pm0.02}$ | $2.98_{\pm0.02}$ | $7.58_{\pm0.05}$ | $11.31_{\pm0.02}$ | $35.58_{\pm0.20}$ | $12.93_{\pm0.06}$ | $14.10_{\pm0.01}$ | $35.05_{\pm0.00}$ |
| | | Transformer | $1.34_{\pm0.01}$ | $2.98_{\pm0.03}$ | $4.84_{\pm0.03}$ | $11.38_{\pm0.01}$ | $35.86_{\pm0.08}$ | $12.84_{\pm0.04}$ | $14.17_{\pm0.02}$ | $35.36_{\pm0.00}$ |
| Fwd-KL | Flow | GRU | $1.37_{\pm0.00}$ | $1.71_{\pm0.16}$ | $10.18_{\pm0.04}$ | $11.09_{\pm0.02}$ | $36.35_{\pm0.01}$ | $15.29_{\pm0.02}$ | $14.66_{\pm0.01}$ | $36.66_{\pm0.00}$ |
| | | DeepSets | $1.58_{\pm0.01}$ | $1.81_{\pm0.08}$ | $10.48_{\pm0.14}$ | $11.08_{\pm0.01}$ | $36.41_{\pm0.00}$ | $16.72_{\pm0.00}$ | $14.67_{\pm0.01}$ | $36.66_{\pm0.00}$ |
| | | Transformer | $1.40_{\pm0.01}$ | $1.20_{\pm0.26}$ | $5.64_{\pm0.23}$ | $11.08_{\pm0.00}$ | $36.32_{\pm0.02}$ | $14.66_{\pm0.01}$ | $14.65_{\pm0.01}$ | $36.66_{\pm0.00}$ |
| Rev-KL | | GRU | $1.33_{\pm0.00}$ | $2.99_{\pm0.02}$ | $7.48_{\pm0.06}$ | $11.15_{\pm0.04}$ | $35.97_{\pm0.04}$ | $13.51_{\pm0.02}$ | $14.33_{\pm0.01}$ | $35.79_{\pm0.01}$ |
| | | DeepSets | $1.41_{\pm0.02}$ | $2.96_{\pm0.01}$ | $8.40_{\pm0.09}$ | $11.15_{\pm0.04}$ | $36.02_{\pm0.09}$ | $13.61_{\pm0.05}$ | $14.32_{\pm0.01}$ | $35.69_{\pm0.01}$ |
| | | Transformer | $1.33_{\pm0.01}$ | $3.00_{\pm0.04}$ | $4.90_{\pm0.15}$ | $11.21_{\pm0.02}$ | $36.00_{\pm0.17}$ | $13.63_{\pm0.02}$ | $14.37_{\pm0.01}$ | $35.88_{\pm0.02}$ |

Table 6: **Sample Based Metrics**. We compute the squared $2-$Wasserstein metric between samples from the approximate posterior and samples obtained from Langevin MCMC algorithm, over a range of different tasks.

paradigm can be trained without observing $\boldsymbol{\theta}$ but does require a known differentiable likelihood. We show an application of this setting where we don't have access to the ground-truth model but only to samples from it.

## 6 Conclusion

We show that Bayesian posterior inference can be amortized for a broad class of probabilistic models and explore a variety of design decisions associated with it. In particular, we show that reverse KL is effective for learning the amortization network and has significant benefits when modeling the predictive, especially in the presence of model misspecification and generalization to out-of-domain real-world setups. It provides an exciting direction of research which could reduce the load of real-world, complex, and iterative approximations through quick and cheap inference over a trained amortized network. Even further, we believe that our approach provides a direction into learning of generalist in-context Bayesian estimator capable of estimating the posterior for novel datasets zero-shot.

We believe that scaling our approach to more complex probabilistic models as well as leveraging existing real-world data in the training paradigm can lead to even better amortized posterior estimators. Another important future work is to combine our approach with diffusion-based (Zhang & Chen, 2021; Vargas et al., 2023) and other off-policy variational systems (Bengio et al., 2021) for modeling more complex high dimensional distributions.

## Impact Statement

This work studies amortizing variational inference for Bayesian posterior estimation which is a widespread strategy for performing inference in statistics. It provides a natural way of quantifying uncertainty and potentially leading to more robust predictions. While we do not foresee any negative impacts of progress in this area, we encourage caution when applying the methodologies in practice.

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

# Appendix

## A    Related Work

In this section, we draw parallels of our work to various approaches that have been proposed to tackle the problem of either providing a good initialization for different tasks, performing implicit optimization to model predictive distributions for new tasks, or estimating the posterior through a different objective.

### A.1    Variational Autoencoders

VAEs (Kingma & Welling, 2013; Rezende et al., 2014; Rezende & Mohamed, 2015; Kingma et al., 2019) are latent variable models which model observations $\boldsymbol{x}$ conditioned on latent variables $\boldsymbol{z}$ through the joint distribution $p_\theta(\boldsymbol{x}, \boldsymbol{z}) = p_\theta(\boldsymbol{x}|\boldsymbol{z})p(\boldsymbol{z})$ where $p(\boldsymbol{z})$ is generally chosen as $\mathcal{N}(\boldsymbol{0}, \mathbf{I})$. Training the model is done through VI where $q_\varphi(\boldsymbol{z})$ is obtained by explicit amortization over the data point, that is, $q_\varphi(\boldsymbol{z}|\boldsymbol{x}) = \mathcal{N}(\boldsymbol{\mu}_\varphi(\boldsymbol{x}), \boldsymbol{\Sigma}_\varphi(\boldsymbol{x}))$. Training this system on a dataset $\mathcal{D}$ is done by similarly optimizing the Evidence Lower-Bound, which boils down to the following optimization problem

$$\arg\max_{\theta,\varphi} \mathbb{E}_{\boldsymbol{x}\sim\mathcal{D}} \mathbb{E}_{\boldsymbol{z}\sim q(\cdot|\boldsymbol{x})} \left[ \log \frac{p_\theta(\boldsymbol{x}, \boldsymbol{z})}{q_\varphi(\boldsymbol{z}|\boldsymbol{x})} \right] \tag{16}$$

This objective can easily be optimized using gradient-based learning and the reparameterization trick. While typically, a diagonal Gaussian distribution is considered for $q_\varphi$, more complex distributions utilizing normalizing flows can also be used.

### A.2    Hypernetworks

Hypernetworks are neural networks that generate weights for another neural network, used in tasks such as uncertainty quantification, zero-shot learning, etc. We refer for a comprehensive overview to Chauhan et al. (2023). Based on experiments on predicting the weights of a compact MLP (section 4), our work shows similarities with studies in this area but also has significant differences. Regarding uncertainty quantification, hypernetworks are instrumental in creating an ensemble of models by generating multiple weight vectors for the primary network. Each model within this ensemble possesses distinct parameter configurations, enabling robust estimation of uncertainty in model predictions. This feature is precious in safety-critical domains like healthcare, where confidence in predictions is essential. Multiple weight sets can be generated through techniques like dropout within hypernetworks or sampling from a noise distribution. The latter (Krueger et al., 2017) is based on a Bayesian framework where weights can be sampled using invertible network architecture, such as normalizing flows. However, while we amortize posterior inference, the weights sampled from the hypernetwork are not conditioned on information from the currently observed input data during inference time but indirectly solely on the dataset available during training, and retraining would need to be done given a new dataset. Departing from the Bayesian framework, Sun et al. (2017) have shown data-specific discriminative weight prediction, which aligns well with their specific objective of defending a convolutional neural network against adversarial attacks. Combining the ability to sample a new set of weights dataset-specifically but also handling dataset exchangeability, even in the more realistic case of missing information, our work has a distinctly different focus but also can be seen as an extension to hypernetwork research.

### A.3    In-Context Learning

Amortized inference has close links to in-context learning (ICL), which has been gaining popularity, especially in natural language modeling. Various works show how in-context learning can be seen as performing implicit optimization based on the context examples, with some constructions showing exact equivalence with gradient descent in linear regression (Von Oswald et al., 2023; von Oswald et al., 2023). Other works have shown how such systems can be seen as implicitly modeling the Bayesian posterior predictive distribution (Müller et al., 2021). In a similar vein, there have been additional works aimed at directly modeling the posterior predictive distribution by providing the training data as "context" to a Transformer model and training it based on the maximum log-likelihood principle (Hollmann et al., 2022). While such approaches have been

| Objective | $q_\varphi$ | Model | $L_2$ Loss ($\downarrow$) | | | | | |
| | | | Gaussian | | GMM | | | |
| | | | 2D | 100D | 2D-2cl | 2D-5cl | 5D-2cl | 5D-5cl |
|---|---|---|---|---|---|---|---|---|
| Baseline | - | Random | $5.839_{\pm0.015}$ | $301.065_{\pm0.346}$ | $1.887_{\pm0.031}$ | $0.730_{\pm0.004}$ | $5.001_{\pm0.037}$ | $1.670_{\pm0.008}$ |
| | - | Optimization | $1.989_{\pm0.000}$ | $101.243_{\pm0.000}$ | $0.169_{\pm0.000}$ | $0.119_{\pm0.001}$ | $0.425_{\pm0.000}$ | $0.308_{\pm0.000}$ |
| | - | Langevin | $2.013_{\pm0.004}$ | $102.346_{\pm0.031}$ | $0.173_{\pm0.001}$ | $0.125_{\pm0.001}$ | $0.448_{\pm0.009}$ | $0.352_{\pm0.005}$ |
| | - | HMC | $2.018_{\pm0.008}$ | $102.413_{\pm0.028}$ | $0.174_{\pm0.001}$ | $0.135_{\pm0.001}$ | $0.479_{\pm0.007}$ | $0.449_{\pm0.002}$ |
| Fwd-KL | Gaussian | GRU | $2.014_{\pm0.001}$ | $102.641_{\pm0.011}$ | $0.921_{\pm0.013}$ | $0.522_{\pm0.001}$ | $2.430_{\pm0.034}$ | $1.235_{\pm0.011}$ |
| | | DeepSets | $2.012_{\pm0.002}$ | $103.215_{\pm0.054}$ | $0.920_{\pm0.019}$ | $0.522_{\pm0.001}$ | $2.436_{\pm0.037}$ | $1.238_{\pm0.009}$ |
| | | Transformer | $2.013_{\pm0.002}$ | $102.783_{\pm0.005}$ | $0.931_{\pm0.017}$ | $0.522_{\pm0.001}$ | $2.498_{\pm0.026}$ | $1.230_{\pm0.009}$ |
| Rev-KL | | GRU | $2.012_{\pm0.001}$ | $102.509_{\pm0.008}$ | $0.183_{\pm0.002}$ | $0.132_{\pm0.002}$ | $0.471_{\pm0.010}$ | $0.413_{\pm0.019}$ |
| | | DeepSets | $2.011_{\pm0.001}$ | $102.599_{\pm0.042}$ | $0.186_{\pm0.001}$ | $0.127_{\pm0.002}$ | $0.495_{\pm0.018}$ | $0.409_{\pm0.005}$ |
| | | Transformer | $2.013_{\pm0.002}$ | $102.540_{\pm0.025}$ | $0.185_{\pm0.004}$ | $0.122_{\pm0.001}$ | $0.489_{\pm0.019}$ | $0.328_{\pm0.002}$ |
| Fwd-KL | Flow | GRU | $2.014_{\pm0.001}$ | $102.656_{\pm0.019}$ | $0.186_{\pm0.006}$ | $0.242_{\pm0.005}$ | $0.670_{\pm0.094}$ | $0.563_{\pm0.018}$ |
| | | DeepSets | $2.014_{\pm0.001}$ | $103.340_{\pm0.029}$ | $0.185_{\pm0.006}$ | $0.237_{\pm0.008}$ | $0.648_{\pm0.082}$ | $0.583_{\pm0.028}$ |
| | | Transformer | $2.016_{\pm0.002}$ | $102.774_{\pm0.024}$ | $0.188_{\pm0.012}$ | $0.252_{\pm0.001}$ | $0.621_{\pm0.070}$ | $0.592_{\pm0.019}$ |
| Rev-KL | | GRU | $2.013_{\pm0.001}$ | $102.490_{\pm0.012}$ | $0.184_{\pm0.006}$ | $0.130_{\pm0.002}$ | $0.467_{\pm0.003}$ | $0.384_{\pm0.005}$ |
| | | DeepSets | $2.011_{\pm0.001}$ | $102.674_{\pm0.046}$ | $0.188_{\pm0.005}$ | $0.131_{\pm0.002}$ | $0.519_{\pm0.008}$ | $0.405_{\pm0.005}$ |
| | | Transformer | $2.013_{\pm0.001}$ | $102.525_{\pm0.050}$ | $0.187_{\pm0.004}$ | $0.123_{\pm0.001}$ | $0.468_{\pm0.007}$ | $0.326_{\pm0.008}$ |

Table 7: **Fixed-Dimensional**. Results for estimating the mean of a Gaussian (Gaussian) and means of a Gaussian mixture model (GMM) with the expected $L_2$ loss according to the posterior predictive as metric.

seeing tremendous success, they cannot be directly applied to cases where we care about and want to analyze the solution space as the solution space is only modeled implicitly, and thus, recovering it is not possible. For example, if our goal is to learn a linear regression model, an ICL model could end up learning a nonlinear model and would provide no information about the actual parameters used for prediction. As opposed to this, we obtain parameters explicitly. We thus can answer questions like the relevance of a particular feature (which corresponds to its weight in the output, and we know the weight vector explicitly). Even further, many systems grounded in physics and economics only admit a constrained solution space; for example, the movement of a human arm lies on a particular manifold, or the configuration of molecules and proteins cannot be arbitrary. Thus, performing predictions through an implicit solution space, which may violate several constraints, is not ideal. Furthermore, explicitly modeling the solution space and encoding the constraints present can be done through the prior and the parametric distribution used for modeling.

## A.4    Meta Learning

Meta-learning (Hospedales et al., 2022) aims to equip models with the ability to quickly learn from different tasks or data sets to generalize to new tasks in resource-constrained domains. This attribute is precious in practical scenarios where obtaining large amounts of task-specific data is impractical or costly. A simple way of obtaining this is through nonparametric or similarity-based models like k-Nearest Neighbours, where no training is involved. Thus, new tasks can be solved quickly based on a few examples by computing a similarity metric with these examples (Koch et al., 2015; Vinyals et al., 2016; Sung et al., 2018). Another way of achieving this is through optimization-based setups, which use a nested optimization procedure. An inner step learns individual tasks from a shared initialization, whereas the outer loop computes the gradient of the whole inner process and moves the initialization in a way that allows for better generalization. Here, by relying on only a few iterations in the inner loop, the outer loop has the incentive to move the initialization to a point from which solutions to multiple tasks are reachable (Finn et al., 2017). Given the similarities between

| Objective | $q_\varphi$ | Model | $L_2$ Loss ($\downarrow$) | | Accuracy ($\uparrow$) | | | |
| | | | Linear Regression | | Linear Classification | | | |
| | | | 2D | 100D | 2D-2cl | 2D-5cl | 100D-2cl | 100D-5cl |
|---|---|---|---|---|---|---|---|---|
| Baseline | - | Random | $4.178_{\pm 0.018}$ | $202.601_{\pm 0.321}$ | $50.498_{\pm 0.357}$ | $19.891_{\pm 0.028}$ | $50.046_{\pm 0.047}$ | $20.054_{\pm 0.053}$ |
| | - | Optimization | $0.257_{\pm 0.000}$ | $25.083_{\pm 0.006}$ | $96.982_{\pm 0.000}$ | $93.449_{\pm 0.002}$ | $70.258_{\pm 0.012}$ | $41.338_{\pm 0.012}$ |
| | - | Langevin | $0.263_{\pm 0.002}$ | $23.340_{\pm 0.689}$ | $95.034_{\pm 0.412}$ | $88.277_{\pm 0.290}$ | $65.123_{\pm 0.370}$ | $32.544_{\pm 0.422}$ |
| | - | HMC | $0.263_{\pm 0.001}$ | $18.659_{\pm 0.189}$ | $92.659_{\pm 0.344}$ | $82.169_{\pm 0.518}$ | $62.145_{\pm 0.245}$ | $29.582_{\pm 0.371}$ |
| Fwd-KL | Gaussian | GRU | $0.264_{\pm 0.001}$ | $124.823_{\pm 0.135}$ | $81.170_{\pm 0.389}$ | $71.170_{\pm 0.275}$ | $59.740_{\pm 0.102}$ | $23.042_{\pm 0.246}$ |
| | | DeepSets | $0.264_{\pm 0.000}$ | $123.133_{\pm 1.080}$ | $81.281_{\pm 0.278}$ | $70.993_{\pm 0.191}$ | $50.047_{\pm 0.051}$ | $20.053_{\pm 0.045}$ |
| | | Transformer | $0.264_{\pm 0.000}$ | $45.856_{\pm 1.331}$ | $80.960_{\pm 0.285}$ | $71.484_{\pm 0.437}$ | $62.954_{\pm 0.062}$ | $26.789_{\pm 0.110}$ |
| Rev-KL | | GRU | $0.263_{\pm 0.000}$ | $60.215_{\pm 0.866}$ | $94.258_{\pm 0.034}$ | $87.339_{\pm 0.023}$ | $63.465_{\pm 0.307}$ | $28.270_{\pm 0.462}$ |
| | | DeepSets | $0.263_{\pm 0.000}$ | $62.837_{\pm 0.617}$ | $94.285_{\pm 0.116}$ | $87.342_{\pm 0.021}$ | $60.867_{\pm 0.265}$ | $21.339_{\pm 0.085}$ |
| | | Transformer | $0.264_{\pm 0.001}$ | $28.735_{\pm 0.252}$ | $94.302_{\pm 0.054}$ | $87.540_{\pm 0.117}$ | $68.185_{\pm 0.007}$ | $32.950_{\pm 0.284}$ |
| Fwd-KL | Flow | GRU | $0.264_{\pm 0.001}$ | $119.119_{\pm 0.233}$ | $96.305_{\pm 0.008}$ | $88.927_{\pm 0.200}$ | $59.920_{\pm 0.221}$ | $23.025_{\pm 0.077}$ |
| | | DeepSets | $0.264_{\pm 0.001}$ | $125.677_{\pm 3.731}$ | $96.191_{\pm 0.021}$ | $88.643_{\pm 0.102}$ | $50.061_{\pm 0.021}$ | $20.021_{\pm 0.094}$ |
| | | Transformer | $0.264_{\pm 0.000}$ | $43.272_{\pm 2.700}$ | $96.344_{\pm 0.059}$ | $89.624_{\pm 0.215}$ | $64.349_{\pm 0.147}$ | $26.952_{\pm 0.203}$ |
| Rev-KL | | GRU | $0.263_{\pm 0.000}$ | $61.295_{\pm 1.008}$ | $95.241_{\pm 0.012}$ | $88.429_{\pm 0.024}$ | $64.669_{\pm 0.207}$ | $28.409_{\pm 1.167}$ |
| | | DeepSets | $0.263_{\pm 0.001}$ | $76.412_{\pm 2.038}$ | $95.296_{\pm 0.021}$ | $88.464_{\pm 0.061}$ | $58.384_{\pm 0.812}$ | $21.569_{\pm 0.117}$ |
| | | Transformer | $0.263_{\pm 0.000}$ | $29.358_{\pm 1.569}$ | $95.339_{\pm 0.063}$ | $88.644_{\pm 0.047}$ | $68.721_{\pm 0.121}$ | $33.107_{\pm 0.333}$ |

Table 8: **Fixed-Dimensional**. Results for estimating the parameters of linear regression (LR) and classification (LC) models with the expected $L_2$ loss and accuracy according to the posterior predictive as metrics.

meta-learning and hierarchical Bayesian inference (Grant et al., 2018), our approach can be considered as a kind of meta-learning framework; however, the line between meta-learning and Bayesian posterior inference is quite blurry as any amortized approach for the latter can be seen as a case of the former.

## A.5 Neural Processes

A notable approach in meta-learning related to our research is neural processes (NP), which excel in learning scenarios with few examples. NPs (Garnelo et al., 2018a;b; Kim et al., 2019; Pakman et al., 2020; Gordon et al., 2019) can be seen as a more flexible and powerful extension of Gaussian processes that leverage a neural network-based encoder-decoder architecture for learning to model a distribution over functions that approximate a stochastic process. However, while we are interested in approximating the posterior distribution over the parameters, NPs are used to approximate the posterior predictive distribution to make predictions based on observed data. Similar to our setup, NPs rely on amortized VI for obtaining the predictive posterior. Still, instead of working with a known probabilistic model, they train the probabilistic model primarily for prediction-based tasks through approaches analogous to variational expectation maximization. Thus, they cannot provide an explicit posterior over the parameters, but they are suitable for tasks where only predictive posteriors are essential, such as those in supervised learning. NPs, in their most basic form, accomplish this by training for the objective:

$$\arg\max_{\boldsymbol{\theta},\varphi} \mathbb{E}_{\mathcal{D}\sim\chi} \mathbb{E}_{\boldsymbol{z}\sim q_\varphi(\cdot|\mathcal{D})} \left[ \log \frac{p_{\boldsymbol{\theta}}(\mathcal{D}, \boldsymbol{z})}{q_\varphi(\boldsymbol{z}|\mathcal{D})} \right] \quad (17)$$

where $\boldsymbol{z} \in \mathbb{R}^p$ is an arbitrary latent variable often uninterpretable, and the parameters of the probabilistic model $\boldsymbol{\theta}$ do not get a Bayesian treatment. In particular, NPs are more suited to modeling datasets of the form $\mathcal{D} = \{\boldsymbol{x}_i, \boldsymbol{y}_i\}_{i=1}^n$, where all probabilities in Equation 17 are conditioned on the input $\boldsymbol{x}$'s, and only the predictive over $\boldsymbol{y}$'s is modeled, and $p_{\boldsymbol{\theta}}$ is modeled as a Neural Network.

| Objective | $q_\varphi$ | Model | $L_2$ Loss $(\downarrow)$ | | | |
|---|---|---|---|---|---|---|
| | | | Nonlinear Regression \| ReLU | | | |
| | | | 1-layer | | 2-layers | |
| | | | 1D | 25D | 1D | 25D |
| Baseline | - | Random | $65.936_{\pm 0.913}$ | $831.595_{\pm 8.696}$ | $1029.407_{\pm 11.542}$ | $12067.691_{\pm 183.598}$ |
| | - | Optimization | $0.360_{\pm 0.001}$ | $103.967_{\pm 0.110}$ | $2.370_{\pm 0.015}$ | $1894.574_{\pm 4.266}$ |
| | - | Langevin | $0.308_{\pm 0.000}$ | $132.391_{\pm 0.992}$ | N/A | N/A |
| | - | HMC | $0.374_{\pm 0.002}$ | $98.061_{\pm 0.730}$ | $22.314_{\pm 0.814}$ | $3903.510_{\pm 5.377}$ |
| Fwd-KL | Gaussian | GRU | $49.332_{\pm 0.946}$ | $671.639_{\pm 10.494}$ | $774.045_{\pm 7.521}$ | $9905.246_{\pm 214.545}$ |
| | | DeepSets | $49.864_{\pm 0.979}$ | $684.853_{\pm 2.581}$ | $768.921_{\pm 8.278}$ | $9946.090_{\pm 109.933}$ |
| | | Transformer | $49.678_{\pm 0.940}$ | $680.853_{\pm 5.838}$ | $747.221_{\pm 12.189}$ | $9982.609_{\pm 85.596}$ |
| Rev-KL | | GRU | $0.426_{\pm 0.004}$ | $105.976_{\pm 0.586}$ | $1.066_{\pm 0.069}$ | $1796.512_{\pm 5.805}$ |
| | | DeepSets | $0.426_{\pm 0.004}$ | $125.853_{\pm 0.791}$ | $1.394_{\pm 0.108}$ | $1892.402_{\pm 2.793}$ |
| | | Transformer | $0.417_{\pm 0.005}$ | $102.295_{\pm 1.825}$ | $2.075_{\pm 0.147}$ | $1811.440_{\pm 115.435}$ |
| Fwd-KL | Flow | GRU | $15.781_{\pm 0.210}$ | $538.962_{\pm 4.269}$ | $614.925_{\pm 15.494}$ | $7564.076_{\pm 67.160}$ |
| | | DeepSets | $15.051_{\pm 0.120}$ | $548.535_{\pm 3.288}$ | $622.461_{\pm 7.043}$ | $7618.364_{\pm 115.946}$ |
| | | Transformer | $16.109_{\pm 0.307}$ | $539.338_{\pm 4.336}$ | $597.718_{\pm 8.358}$ | $7635.052_{\pm 109.037}$ |
| Rev-KL | | GRU | $0.405_{\pm 0.010}$ | $106.001_{\pm 0.420}$ | $0.988_{\pm 0.045}$ | $1814.649_{\pm 8.327}$ |
| | | DeepSets | $0.395_{\pm 0.004}$ | $128.169_{\pm 1.451}$ | $1.215_{\pm 0.028}$ | $1886.698_{\pm 7.294}$ |
| | | Transformer | $0.387_{\pm 0.004}$ | $102.610_{\pm 0.863}$ | $2.549_{\pm 0.058}$ | $1791.741_{\pm 49.585}$ |

Table 9: **Fixed-Dimensional**. Results for estimating the parameters of nonlinear regression models with ReLU activation function, with the expected $L_2$ loss according to the posterior predictive as metric.

These approaches can be seen as quite related to ICL, where the exchangeable architecture backbone is switched from DeepSets to Transformers. Similar to ICL, they do not provide control over the solution space as they aim to model either the posterior predictive or an arbitrary latent space. While this leads to good predictive performance on various tasks, they cannot be freely applied to problems that pose certain constraints on the underlying probabilistic model. In such cases, estimating the actual parameters is important to enforce constraints in the parameter space as well as for interpretability, which we already discussed in the ICL section.

## A.6 Simulation-Based Inference

In the case of simulation-based inference (Cranmer et al., 2020), when the likelihood $p(\boldsymbol{x}|\boldsymbol{\theta})$ is intractable, BayesFlow (Radev et al., 2020) and similar methods (Lorch et al., 2022) provide a solution framework to amortize Bayesian inference of parameters in complex models. Starting from the forward KL divergence between the true and approximate posteriors, the resulting objective is to optimize for parameters of the approximate posterior distribution that maximize the posterior probability of data-generating parameters $\boldsymbol{\theta}$ given observed data $\mathcal{D}$ for all $\boldsymbol{\theta}$ and $\mathcal{D}$. Density estimation of the approximate posterior can then be done using the change-of-variables formula and a conditional invertible neural network that parameterizes the approximate posterior distribution.

$$\arg\min_{\varphi} \mathbb{KL}[p(\boldsymbol{\theta}|\mathcal{D})||q_\varphi(\boldsymbol{\theta}|\mathcal{D})] = \arg\min_{\varphi=\{\nu,\psi\}} \mathbb{E}_{(\boldsymbol{\theta},\mathcal{D})\sim p(\boldsymbol{\theta},\mathcal{D})} \left[ -\log p_{\boldsymbol{z}}(f_\nu(\boldsymbol{\theta}; h_\psi(\mathcal{D}))) - \log |\det J_{f_\nu}| \right] \quad (18)$$

Since their goal is to learn a global estimator for the probabilistic mapping from $\mathcal{D}$ to data generating $\boldsymbol{\theta}$, the information about the observed dataset is encoded in the output of a summary network $h_\psi$. It is used as conditional input to the normalizing flow $f_\nu$. Although the likelihood function does not need to be

| Objective | $q_\varphi$ | Model | $L_2$ Loss $(\downarrow)$ | | | |
|---|---|---|---|---|---|---|
| | | | **Nonlinear Regression $\mid$ TanH** | | | |
| | | | *1-layer* | | *2-layers* | |
| | | | *1D* | *25D* | *1D* | *25D* |
| Baseline | - | Random | $31.448_{\pm 0.186}$ | $52.644_{\pm 0.173}$ | $52.735_{\pm 1.122}$ | $52.583_{\pm 0.132}$ |
| | - | Optimization | $0.366_{\pm 0.001}$ | $13.352_{\pm 0.005}$ | $0.651_{\pm 0.002}$ | $30.176_{\pm 0.056}$ |
| | - | Langevin | $0.296_{\pm 0.003}$ | $17.221_{\pm 0.130}$ | $0.363_{\pm 0.003}$ | $28.528_{\pm 0.115}$ |
| | - | HMC | $0.398_{\pm 0.003}$ | $12.607_{\pm 0.192}$ | $0.733_{\pm 0.021}$ | $23.571_{\pm 0.346}$ |
| Fwd-KL | Gaussian | GRU | $31.391_{\pm 0.161}$ | $52.008_{\pm 0.282}$ | $52.725_{\pm 1.149}$ | $51.989_{\pm 0.139}$ |
| | | DeepSets | $31.421_{\pm 0.074}$ | $52.137_{\pm 0.215}$ | $52.850_{\pm 1.192}$ | $51.904_{\pm 0.334}$ |
| | | Transformer | $31.350_{\pm 0.219}$ | $52.945_{\pm 0.430}$ | $52.693_{\pm 1.188}$ | $52.364_{\pm 0.164}$ |
| Rev-KL | | GRU | $0.415_{\pm 0.003}$ | $15.874_{\pm 6.958}$ | $0.951_{\pm 0.047}$ | $25.907_{\pm 0.012}$ |
| | | DeepSets | $0.405_{\pm 0.004}$ | $25.333_{\pm 0.010}$ | $0.912_{\pm 0.013}$ | $25.877_{\pm 0.002}$ |
| | | Transformer | $0.412_{\pm 0.013}$ | $11.784_{\pm 0.949}$ | $0.847_{\pm 0.010}$ | $20.405_{\pm 3.874}$ |
| Fwd-KL | Flow | GRU | $12.415_{\pm 0.800}$ | $52.039_{\pm 0.065}$ | $52.695_{\pm 0.611}$ | $52.576_{\pm 0.225}$ |
| | | DeepSets | $31.790_{\pm 0.163}$ | $51.933_{\pm 0.115}$ | $52.903_{\pm 0.625}$ | $52.643_{\pm 0.239}$ |
| | | Transformer | $10.392_{\pm 0.195}$ | $52.470_{\pm 0.364}$ | $52.385_{\pm 0.689}$ | $52.646_{\pm 0.622}$ |
| Rev-KL | | GRU | $0.386_{\pm 0.005}$ | $11.401_{\pm 0.041}$ | $0.736_{\pm 0.009}$ | $25.892_{\pm 0.010}$ |
| | | DeepSets | $0.374_{\pm 0.005}$ | $25.685_{\pm 0.004}$ | $0.686_{\pm 0.019}$ | $25.885_{\pm 0.007}$ |
| | | Transformer | $0.376_{\pm 0.002}$ | $10.486_{\pm 0.040}$ | $0.724_{\pm 0.026}$ | $25.885_{\pm 0.011}$ |

Table 10: **Fixed-Dimensional**. Results for estimating the parameters of nonlinear regression models with TanH activation function, with the expected $L_2$ loss according to the posterior predictive as metric.

known, the method requires access to paired observations $(\boldsymbol{x}, \boldsymbol{\theta})$ for training, which is sometimes unavailable. This approach is equivalent to the *Forward KL* setup in our experiments when trained with DeepSets and Normalizing Flows. Current research has also leveraged score-based generative models for SBI which can condition on a dataset by learning a score model conditional only on single observations (Geffner et al., 2023).

## A.7 Amortization in Gaussian Processes

Gaussian Processes (GPs) define a class of probabilistic models that do enjoy tractable likelihood. However, inference in such systems is slow and sensitive to the choice of kernel function that defines the covariance matrix. Similar to meta learning and neural processes, current research also focuses on estimating the kernel function in GPs by leveraging permutation invariant architectures like transformers (Liu et al., 2020; Simpson et al., 2021; Bitzer et al., 2023). Additionally, often these approaches amortize based on point estimates and are leveraged when considering GPs for regression problems, and it is not straightforward to extend them to classification or unsupervised learning. In contrast, our approach is more general and can work for all problems that define a differentiable likelihood function. Additionally, our approach also approximates the Bayesian posterior distribution over the parameters of interest, as opposed to point estimates.

## A.8 Mode Collapse in Variational Inference

Reverse KL based methods have been widely known to suffer from mode collapse due to the nature of the optimization objective (Bishop & Nasrabadi, 2006), which implies that even if the approximate distribution possesses the ability to represent multiple modes, optimization is often sub-optimal and the distribution ends up covering only a small handful of them. Improving normalizing flow based methods with repulsive terms or through the lens of natural gradient optimization procedure for a mixture approximate distribution (Arenz

| Objective | $q_\varphi$ | Model | *Accuracy* ($\uparrow$) | | | |
|---|---|---|---|---|---|---|
| | | | **Nonlinear Classification \| ReLU - 2 class** | | | |
| | | | *1-layer* | | *2-layers* | |
| | | | *2D* | *25D* | *2D* | *25D* |
| Baseline | - | Random | $50.306_{\pm 0.590}$ | $50.008_{\pm 0.326}$ | $50.394_{\pm 0.190}$ | $49.846_{\pm 0.635}$ |
| | - | Optimization | $96.879_{\pm 0.028}$ | $77.896_{\pm 0.023}$ | $96.770_{\pm 0.062}$ | $82.073_{\pm 0.200}$ |
| | - | Langevin | $95.971_{\pm 0.313}$ | $73.165_{\pm 0.282}$ | $96.645_{\pm 0.101}$ | $76.541_{\pm 0.139}$ |
| | - | HMC | $91.763_{\pm 0.163}$ | $70.395_{\pm 0.110}$ | $91.797_{\pm 0.048}$ | $76.445_{\pm 0.372}$ |
| Fwd-KL | Gaussian | GRU | $59.518_{\pm 0.355}$ | $56.858_{\pm 0.319}$ | $60.962_{\pm 0.599}$ | $60.063_{\pm 0.695}$ |
| | | DeepSets | $59.383_{\pm 0.244}$ | $56.806_{\pm 0.204}$ | $61.090_{\pm 0.690}$ | $59.933_{\pm 0.599}$ |
| | | Transformer | $59.588_{\pm 0.389}$ | $57.089_{\pm 0.376}$ | $61.151_{\pm 0.560}$ | $60.041_{\pm 0.680}$ |
| Rev-KL | | GRU | $92.384_{\pm 0.195}$ | $72.455_{\pm 0.032}$ | $86.157_{\pm 0.066}$ | $69.966_{\pm 0.333}$ |
| | | DeepSets | $92.488_{\pm 0.133}$ | $59.806_{\pm 0.315}$ | $86.275_{\pm 0.733}$ | $69.550_{\pm 0.371}$ |
| | | Transformer | $92.627_{\pm 0.377}$ | $75.178_{\pm 0.142}$ | $86.351_{\pm 0.217}$ | $69.812_{\pm 0.527}$ |
| Fwd-KL | Flow | GRU | $76.931_{\pm 0.266}$ | $58.338_{\pm 0.026}$ | $62.981_{\pm 0.452}$ | $63.199_{\pm 0.202}$ |
| | | DeepSets | $72.313_{\pm 1.829}$ | $58.113_{\pm 0.126}$ | $62.438_{\pm 0.277}$ | $62.884_{\pm 0.201}$ |
| | | Transformer | $77.296_{\pm 0.201}$ | $58.344_{\pm 0.148}$ | $63.753_{\pm 0.155}$ | $63.590_{\pm 0.390}$ |
| Rev-KL | | GRU | $93.392_{\pm 0.073}$ | $72.002_{\pm 0.506}$ | $85.712_{\pm 0.665}$ | $71.419_{\pm 0.200}$ |
| | | DeepSets | $93.312_{\pm 0.197}$ | $60.943_{\pm 0.184}$ | $85.960_{\pm 0.896}$ | $71.288_{\pm 0.211}$ |
| | | Transformer | $93.578_{\pm 0.093}$ | $74.956_{\pm 0.546}$ | $87.138_{\pm 0.438}$ | $71.525_{\pm 0.159}$ |

Table 11: **Fixed-Dimensional**. Results for estimating the parameters of nonlinear classification models with ReLU activation function and two classes, with the expected accuracy according to the posterior predictive as metric.

et al., 2022; Lin et al., 2020) is an important topic of research, and we believe it would be quite an important future work to experimentally validate if they help in learning multi-modality in amortized posterior inference problems that are studied in this work.

# B    Architectures respecting Exchangeability

In this section, we highlight how DeepSets and Transformer models satisfy the dataset exchangeability criteria, which is essential in modeling the posterior distribution over the parameters of any probabilistic model relying on *iid* data.

## B.1    DeepSets

DeepSets (Zaheer et al., 2017) operate on arbitrary sets $\mathcal{X} = \{x_1, ... x_N\} \subset \mathbb{R}^d$ of fixed dimensionality $d$ by first mapping each individual element $\boldsymbol{x}_i \in \mathcal{X}$ to some high-dimensional space using a nonlinear transform, which is parameterized as a multi-layered neural network with parameters $\varphi_1$

$$\boldsymbol{z}_i = f_{\varphi_1}(\boldsymbol{x}_i) \tag{19}$$

After having obtained this high-dimensional embedding of each element of the set, it applies an aggregation function $a(\cdot)$, which is a permutation invariant function that maps a set of elements $\mathcal{Z} = \{\boldsymbol{z}_1, ..., \boldsymbol{z}_N\} \in \mathbb{R}^z$ to an element $\boldsymbol{h} \in \mathbb{R}^z$,

$$\boldsymbol{h} = a(\mathcal{Z}) \tag{20}$$

| Objective | $q_\varphi$ | Model | Accuracy ($\uparrow$) | | | |
|---|---|---|---|---|---|---|
| | | | Nonlinear Classification \| ReLU - 5 class | | | |
| | | | 1-layer | | 2-layers | |
| | | | 2D | 25D | 2D | 25D |
| Baseline | - | Random | $19.952_{\pm 0.119}$ | $20.026_{\pm 0.062}$ | $19.832_{\pm 0.202}$ | $19.985_{\pm 0.091}$ |
| | - | Optimization | $94.369_{\pm 0.022}$ | $60.589_{\pm 0.064}$ | $93.664_{\pm 0.025}$ | $60.824_{\pm 0.023}$ |
| | - | Langevin | $91.286_{\pm 0.149}$ | $50.845_{\pm 0.366}$ | $92.449_{\pm 0.068}$ | $52.216_{\pm 0.170}$ |
| | - | HMC | $81.387_{\pm 0.742}$ | $47.408_{\pm 0.548}$ | $81.098_{\pm 0.246}$ | $52.854_{\pm 0.190}$ |
| Fwd-KL | Gaussian | GRU | $32.597_{\pm 0.169}$ | $29.932_{\pm 0.142}$ | $31.061_{\pm 0.089}$ | $30.187_{\pm 0.070}$ |
| | | DeepSets | $32.481_{\pm 0.090}$ | $29.909_{\pm 0.218}$ | $30.923_{\pm 0.205}$ | $19.983_{\pm 0.089}$ |
| | | Transformer | $32.977_{\pm 0.114}$ | $30.064_{\pm 0.134}$ | $31.478_{\pm 0.079}$ | $30.307_{\pm 0.121}$ |
| Rev-KL | | GRU | $83.460_{\pm 0.269}$ | $36.688_{\pm 0.189}$ | $71.710_{\pm 0.375}$ | $28.234_{\pm 0.148}$ |
| | | DeepSets | $83.734_{\pm 0.313}$ | $34.078_{\pm 0.066}$ | $67.519_{\pm 1.735}$ | $47.585_{\pm 0.143}$ |
| | | Transformer | $84.645_{\pm 0.515}$ | $36.850_{\pm 0.138}$ | $74.390_{\pm 0.280}$ | $27.808_{\pm 0.232}$ |
| Fwd-KL | Flow | GRU | $43.850_{\pm 0.412}$ | $31.926_{\pm 0.096}$ | $33.771_{\pm 0.400}$ | $31.871_{\pm 0.228}$ |
| | | DeepSets | $43.620_{\pm 0.250}$ | $31.829_{\pm 0.126}$ | $32.919_{\pm 0.126}$ | $19.950_{\pm 0.192}$ |
| | | Transformer | $44.057_{\pm 0.122}$ | $31.789_{\pm 0.094}$ | $34.079_{\pm 0.224}$ | $32.392_{\pm 0.212}$ |
| Rev-KL | | GRU | $84.459_{\pm 0.194}$ | $37.343_{\pm 0.134}$ | $64.309_{\pm 0.273}$ | $48.459_{\pm 0.542}$ |
| | | DeepSets | $84.775_{\pm 0.129}$ | $34.917_{\pm 0.069}$ | $67.937_{\pm 4.341}$ | $48.563_{\pm 0.199}$ |
| | | Transformer | $85.857_{\pm 0.096}$ | $46.968_{\pm 0.094}$ | $75.005_{\pm 0.605}$ | $48.889_{\pm 0.380}$ |

Table 12: **Fixed-Dimensional**. Results for estimating the parameters of nonlinear classification models with ReLU activation function and five classes, with the expected accuracy according to the posterior predictive as metric.

Thus, the outcome does not change under permutations of $\mathcal{Z}$. Finally, another nonlinear transform, parameterized by a multi-layered neural network with parameters $\varphi_2$, is applied to the outcome $\boldsymbol{h}$ to provide the final output.

$$\boldsymbol{o} = g_{\varphi_2}(\boldsymbol{h}) \tag{21}$$

For our experiments, we then use the vector $\boldsymbol{o}$ to predict the parameters of a parametric family of distributions (e.g., Gaussian or Flows) using an additional nonlinear neural network. As an example, for the Gaussian case, we consider the distribution $\mathcal{N}(\cdot|\boldsymbol{\mu}, \boldsymbol{\Sigma})$, where

$$\boldsymbol{\mu} := \boldsymbol{\mu}_{\varphi_3}(\boldsymbol{o}) \quad \text{and} \quad \boldsymbol{\Sigma} := \boldsymbol{\Sigma}_{\varphi_4}(\boldsymbol{o}) \tag{22}$$

which makes $\boldsymbol{\mu}$ implicitly a function of the original input set $\mathcal{X}$. To understand why the posterior distribution modeled in this fashion does not change when the inputs are permuted, let us assume that $\Pi$ is a permutation over the elements of $\mathcal{X}$. If we look at one of the parameters of the posterior distribution, e.g., $\boldsymbol{\mu}$, we can see that

$$\boldsymbol{\mu}(\Pi\mathcal{X}) = \boldsymbol{\mu}_{\varphi_3}\left(g_{\varphi_2}\left(a\left(\{f_{\varphi_1}(\boldsymbol{x}_{\Pi(i)})\}_{i=1}^N\right)\right)\right) \tag{23}$$

$$= \boldsymbol{\mu}_{\varphi_3}\left(g_{\varphi_2}\left(a\left(\{f_{\varphi_1}(\boldsymbol{x}_i)\}_{i=1}^N\right)\right)\right) \tag{24}$$

$$= \boldsymbol{\mu}(\mathcal{X}) \tag{25}$$

which simply follows from the fact that $a(\cdot)$ is a permutation invariant operation, e.g., sum or mean. We can also provide similar reasoning for the other parameters (e.g., $\boldsymbol{\Sigma}$). This shows that DeepSets can be used to model the posterior distribution over parameters of interest as it respects the exchangeability criteria (*iid* observations) assumptions in the data through its permutation invariant structure.

| Objective | $q_\varphi$ | Model | Accuracy ($\uparrow$) | | | |
|-----------|-----|-------|-----|-----|-----|-----|
| | | | **Nonlinear Classification \| TanH - 2 class** | | | |
| | | | *1-layer* | | *2-layers* | |
| | | | *2D* | *25D* | *2D* | *25D* |
| Baseline | - | Random | $50.147_{\pm 0.603}$ | $49.963_{\pm 0.083}$ | $49.942_{\pm 0.452}$ | $49.978_{\pm 0.113}$ |
| | - | Optimization | $96.552_{\pm 0.005}$ | $75.228_{\pm 0.029}$ | $94.130_{\pm 0.018}$ | $69.052_{\pm 0.029}$ |
| | - | Langevin | $94.778_{\pm 0.210}$ | $68.787_{\pm 0.258}$ | $92.417_{\pm 0.149}$ | $62.722_{\pm 0.180}$ |
| | - | HMC | $91.674_{\pm 0.200}$ | $67.415_{\pm 0.667}$ | $88.218_{\pm 0.202}$ | $62.323_{\pm 0.262}$ |
| Fwd-KL | Gaussian | GRU | $50.151_{\pm 0.616}$ | $49.959_{\pm 0.087}$ | $49.942_{\pm 0.448}$ | $49.987_{\pm 0.112}$ |
| | | DeepSets | $50.147_{\pm 0.601}$ | $49.961_{\pm 0.083}$ | $49.939_{\pm 0.451}$ | $49.977_{\pm 0.110}$ |
| | | Transformer | $50.146_{\pm 0.622}$ | $49.959_{\pm 0.082}$ | $49.948_{\pm 0.448}$ | $49.970_{\pm 0.104}$ |
| Rev-KL | | GRU | $89.813_{\pm 0.181}$ | $49.969_{\pm 0.085}$ | $49.956_{\pm 0.442}$ | $49.974_{\pm 0.088}$ |
| | | DeepSets | $89.558_{\pm 0.264}$ | $49.970_{\pm 0.100}$ | $49.961_{\pm 0.436}$ | $49.986_{\pm 0.094}$ |
| | | Transformer | $89.879_{\pm 0.387}$ | $49.978_{\pm 0.075}$ | $79.102_{\pm 0.155}$ | $49.987_{\pm 0.089}$ |
| Fwd-KL | Flow | GRU | $50.066_{\pm 0.488}$ | $49.835_{\pm 0.240}$ | $49.836_{\pm 0.328}$ | $49.911_{\pm 0.206}$ |
| | | DeepSets | $50.078_{\pm 0.468}$ | $49.848_{\pm 0.242}$ | $49.834_{\pm 0.333}$ | $49.910_{\pm 0.205}$ |
| | | Transformer | $50.067_{\pm 0.483}$ | $49.827_{\pm 0.235}$ | $49.813_{\pm 0.331}$ | $49.905_{\pm 0.210}$ |
| Rev-KL | | GRU | $90.396_{\pm 0.126}$ | $49.910_{\pm 0.169}$ | $50.018_{\pm 0.171}$ | $50.001_{\pm 0.076}$ |
| | | DeepSets | $90.247_{\pm 0.067}$ | $49.901_{\pm 0.124}$ | $50.132_{\pm 0.162}$ | $49.942_{\pm 0.110}$ |
| | | Transformer | $90.416_{\pm 0.311}$ | $49.904_{\pm 0.133}$ | $81.729_{\pm 0.070}$ | $49.945_{\pm 0.203}$ |

Table 13: **Fixed-Dimensional**. Results for estimating the parameters of nonlinear classification models with TanH activation function and two classes, with the expected accuracy according to the posterior predictive as metric.

### B.2 Transformers

Similarly, we can look at Transformers (Vaswani et al., 2017) as candidates for respecting the exchangeability conditions in the data. In particular, we consider transformer systems without positional encodings and consider an additional [CLS] token, denoted by $\boldsymbol{c} \in \mathbb{R}^d$, to drive the prediction. If we look at the application of a layer of transformer model, it can be broken down into two components.

**Multi-Head Attention**. Given a query vector obtained from $\boldsymbol{c}$ and keys and values coming from our input set $\mathcal{X} \subset \mathbb{R}^d$, we can model the update of the context $\boldsymbol{c}$ as

$$\hat{\boldsymbol{c}}(\mathcal{X}) = \text{Softmax}\left(\boldsymbol{c}^T \boldsymbol{W}_Q \boldsymbol{W}_K^T \boldsymbol{X}^T\right) \boldsymbol{X} \boldsymbol{W}_V \tag{26}$$

where $\boldsymbol{W}_Q \in \mathbb{R}^{d \times k}, \boldsymbol{W}_K \in \mathbb{R}^{d \times k}, \boldsymbol{W}_V \in \mathbb{R}^{d \times k}$ and $\boldsymbol{X} \in \mathbb{R}^{N \times d}$ denotes a certain ordering of the elements in $\mathcal{X}$. Further, $\hat{\boldsymbol{c}}$ is the updated vector after attention, and Softmax is over the rows of $\boldsymbol{X}$. Here, we see that if we were to apply a permutation to the elements in $\boldsymbol{X}$, the outcome would remain the same. In particular

$$\hat{\boldsymbol{c}}(\Pi \boldsymbol{X}) = \text{Softmax}\left(\boldsymbol{c}^T \boldsymbol{W}_Q \boldsymbol{W}_K^T \boldsymbol{X}^T \Pi^T\right) \Pi \boldsymbol{X} \boldsymbol{W}_V \tag{27}$$

$$= \text{Softmax}\left(\boldsymbol{c}^T \boldsymbol{W}_Q \boldsymbol{W}_K^T \boldsymbol{X}^T\right) \Pi^T \Pi \boldsymbol{X} \boldsymbol{W}_V \tag{28}$$

$$= \text{Softmax}\left(\boldsymbol{c}^T \boldsymbol{W}_Q \boldsymbol{W}_K^T \boldsymbol{X}^T\right) \boldsymbol{X} \boldsymbol{W}_V \tag{29}$$

$$= \hat{\boldsymbol{c}}(\boldsymbol{X}) \tag{30}$$

which follows because Softmax is an equivariant function, i.e., applying Softmax on a permutation of columns is equivalent to applying Softmax first and then permuting the columns correspondingly. Thus, we see that the update to the [CLS] token $\boldsymbol{c}$ is permutation invariant. This output is then used independently as input to

| Objective | $q_\varphi$ | Model | Accuracy ($\uparrow$) | | | |
| | | | Nonlinear Classification \| TanH - 5 class | | | |
| | | | 1-layer | | 2-layers | |
| | | | 2D | 25D | 2D | 25D |
| Baseline | - | Random | $19.745_{\pm 0.269}$ | $20.037_{\pm 0.075}$ | $20.214_{\pm 0.043}$ | $19.864_{\pm 0.089}$ |
| | - | Optimization | $92.919_{\pm 0.011}$ | $49.972_{\pm 0.070}$ | $88.156_{\pm 0.014}$ | $39.410_{\pm 0.043}$ |
| | - | Langevin | $88.635_{\pm 0.374}$ | $39.758_{\pm 0.212}$ | $84.050_{\pm 0.138}$ | $31.472_{\pm 0.010}$ |
| | - | HMC | $81.057_{\pm 0.277}$ | $35.378_{\pm 0.130}$ | $75.305_{\pm 0.134}$ | $29.730_{\pm 0.670}$ |
| Fwd-KL | Gaussian | GRU | $19.960_{\pm 0.271}$ | $20.235_{\pm 0.083}$ | $20.452_{\pm 0.059}$ | $20.028_{\pm 0.077}$ |
| | | DeepSets | $19.807_{\pm 0.209}$ | $20.040_{\pm 0.072}$ | $20.210_{\pm 0.043}$ | $19.861_{\pm 0.094}$ |
| | | Transformer | $19.977_{\pm 0.273}$ | $20.241_{\pm 0.068}$ | $20.453_{\pm 0.062}$ | $20.029_{\pm 0.082}$ |
| Rev-KL | | GRU | $77.711_{\pm 0.014}$ | $20.026_{\pm 0.079}$ | $20.213_{\pm 0.043}$ | $19.877_{\pm 0.092}$ |
| | | DeepSets | $76.414_{\pm 0.378}$ | $20.038_{\pm 0.060}$ | $20.216_{\pm 0.027}$ | $19.887_{\pm 0.089}$ |
| | | Transformer | $79.163_{\pm 0.183}$ | $20.026_{\pm 0.093}$ | $51.408_{\pm 0.484}$ | $19.872_{\pm 0.083}$ |
| Fwd-KL | Flow | GRU | $32.900_{\pm 0.115}$ | $20.209_{\pm 0.048}$ | $20.105_{\pm 0.385}$ | $20.040_{\pm 0.037}$ |
| | | DeepSets | $20.137_{\pm 0.070}$ | $20.000_{\pm 0.048}$ | $19.887_{\pm 0.392}$ | $19.895_{\pm 0.025}$ |
| | | Transformer | $30.135_{\pm 2.459}$ | $20.224_{\pm 0.048}$ | $20.104_{\pm 0.399}$ | $20.030_{\pm 0.039}$ |
| Rev-KL | | GRU | $79.329_{\pm 0.320}$ | $20.074_{\pm 0.033}$ | $19.904_{\pm 0.180}$ | $19.864_{\pm 0.079}$ |
| | | DeepSets | $20.071_{\pm 0.302}$ | $19.999_{\pm 0.045}$ | $19.872_{\pm 0.259}$ | $19.906_{\pm 0.051}$ |
| | | Transformer | $80.064_{\pm 0.159}$ | $20.002_{\pm 0.032}$ | $19.777_{\pm 0.210}$ | $19.911_{\pm 0.082}$ |

Table 14: **Fixed-Dimensional**. Results for estimating the parameters of nonlinear classification models with TanH activation function and five classes, with the expected accuracy according to the posterior predictive as metric.

a multi-layered neural network with residual connections, and the entire process is repeated multiple times without weight sharing to simulate multiple layers. Since all the individual parts are permutation invariant w.r.t permutations on $\mathcal{X}$, the entire setup ends up being permutation invariant. Obtaining the parameters of a parametric family of distribution for posterior estimation then follows the same recipe as DeepSets, with $\boldsymbol{o}$ replaced by $\boldsymbol{c}$.

## C  Probabilistic Models

This section details the various candidate probabilistic models used in our experiments for amortized computation of Bayesian posteriors over the parameters. Here, we explain the parameters associated with the probabilistic model over which we want to estimate the posterior and the likelihood and prior that we use for experimentation.

**Mean of Gaussian (GM):** As a proof of concept, we consider the simple setup of estimating the posterior distribution over the mean of a Gaussian distribution $p(\boldsymbol{\mu}|\mathcal{D})$ given some observed data. In this case, prior and likelihood defining the probabilistic model $p(\boldsymbol{x}, \boldsymbol{\theta})$ (with $\boldsymbol{\theta}$ being the mean $\boldsymbol{\mu}$) are given by:

$$p(\boldsymbol{\mu}) = \mathcal{N}\left(\boldsymbol{\mu}|\mathbf{0}, \mathbf{I}\right) \tag{31}$$
$$p(\boldsymbol{x}|\boldsymbol{\mu}) = \mathcal{N}\left(\boldsymbol{x}|\boldsymbol{\mu}, \boldsymbol{\Sigma}\right) \tag{32}$$

and $\boldsymbol{\Sigma}$ is known beforehand and defined as a unit variance matrix.

| Objective | $q_\varphi$ | Model | $L_2$ Loss ($\downarrow$) | | | | | |
|---|---|---|---|---|---|---|---|---|
| | | | **Gaussian** | | **GMM** | | | |
| | | | *2D* | *100D* | *2D-2cl* | *2D-5cl* | *5D-2cl* | *5D-5cl* |
| Baseline | - | Random | $6.297_{\pm 0.017}$ | $298.238_{\pm 0.228}$ | $2.078_{\pm 0.134}$ | $0.626_{\pm 0.037}$ | $4.659_{\pm 0.034}$ | $1.632_{\pm 0.004}$ |
| | - | Optimization | $2.020_{\pm 0.000}$ | $100.885_{\pm 0.000}$ | $0.175_{\pm 0.002}$ | $0.121_{\pm 0.002}$ | $0.427_{\pm 0.000}$ | $0.323_{\pm 0.002}$ |
| | - | Langevin | $2.036_{\pm 0.004}$ | $101.917_{\pm 0.042}$ | $0.178_{\pm 0.002}$ | $0.123_{\pm 0.002}$ | $0.440_{\pm 0.002}$ | $0.340_{\pm 0.006}$ |
| | - | HMC | $2.044_{\pm 0.008}$ | $102.015_{\pm 0.009}$ | $0.189_{\pm 0.013}$ | $0.132_{\pm 0.004}$ | $0.462_{\pm 0.013}$ | $0.423_{\pm 0.009}$ |
| Fwd-KL | Gaussian | GRU | $2.300_{\pm 0.105}$ | $133.224_{\pm 0.579}$ | $1.119_{\pm 0.150}$ | $0.473_{\pm 0.012}$ | $2.360_{\pm 0.017}$ | $1.208_{\pm 0.002}$ |
| | | DeepSets | $2.216_{\pm 0.017}$ | $129.695_{\pm 0.737}$ | $1.113_{\pm 0.150}$ | $0.477_{\pm 0.013}$ | $2.352_{\pm 0.018}$ | $1.210_{\pm 0.003}$ |
| | | Transformer | $2.352_{\pm 0.013}$ | $108.977_{\pm 0.100}$ | $1.134_{\pm 0.153}$ | $0.476_{\pm 0.013}$ | $2.399_{\pm 0.019}$ | $1.208_{\pm 0.004}$ |
| Rev-KL | | GRU | $2.047_{\pm 0.002}$ | $105.141_{\pm 0.102}$ | $0.187_{\pm 0.005}$ | $0.147_{\pm 0.006}$ | $0.462_{\pm 0.011}$ | $0.398_{\pm 0.019}$ |
| | | DeepSets | $2.049_{\pm 0.003}$ | $105.062_{\pm 0.212}$ | $0.194_{\pm 0.004}$ | $0.145_{\pm 0.003}$ | $0.481_{\pm 0.021}$ | $0.387_{\pm 0.002}$ |
| | | Transformer | $2.057_{\pm 0.004}$ | $104.709_{\pm 0.122}$ | $0.195_{\pm 0.004}$ | $0.140_{\pm 0.003}$ | $0.468_{\pm 0.006}$ | $0.335_{\pm 0.016}$ |
| Fwd-KL | Flow | GRU | $2.358_{\pm 0.005}$ | $125.835_{\pm 1.983}$ | $0.287_{\pm 0.032}$ | $0.212_{\pm 0.002}$ | $0.596_{\pm 0.068}$ | $0.513_{\pm 0.015}$ |
| | | DeepSets | $2.053_{\pm 0.003}$ | $133.229_{\pm 1.933}$ | $0.271_{\pm 0.050}$ | $0.202_{\pm 0.006}$ | $0.584_{\pm 0.030}$ | $0.517_{\pm 0.016}$ |
| | | Transformer | $2.060_{\pm 0.005}$ | $108.484_{\pm 0.164}$ | $0.344_{\pm 0.054}$ | $0.221_{\pm 0.006}$ | $0.591_{\pm 0.080}$ | $0.533_{\pm 0.010}$ |
| Rev-KL | | GRU | $2.050_{\pm 0.004}$ | $105.187_{\pm 0.030}$ | $0.199_{\pm 0.014}$ | $0.142_{\pm 0.004}$ | $0.466_{\pm 0.007}$ | $0.373_{\pm 0.004}$ |
| | | DeepSets | $2.054_{\pm 0.003}$ | $105.095_{\pm 0.064}$ | $0.202_{\pm 0.007}$ | $0.146_{\pm 0.003}$ | $0.494_{\pm 0.013}$ | $0.379_{\pm 0.003}$ |
| | | Transformer | $2.049_{\pm 0.003}$ | $104.914_{\pm 0.113}$ | $0.193_{\pm 0.004}$ | $0.138_{\pm 0.002}$ | $0.460_{\pm 0.003}$ | $0.327_{\pm 0.006}$ |

Table 15: **Variable-Dimensional**. Results for estimating the mean of a Gaussian (Gaussian) and means of a Gaussian mixture model (GMM) with the expected $L_2$ loss according to the posterior predictive as metric.

**Linear Regression (LR):** We then look at the problem of estimating the posterior over the weight vector for Bayesian linear regression given a dataset $p(\boldsymbol{w}, b|\mathcal{D})$, where the underlying model $p(\mathcal{D}, \boldsymbol{\theta})$ is given by:

$$p(\boldsymbol{w}) = \mathcal{N}(\boldsymbol{w}|\mathbf{0}, \mathbf{I}) \tag{33}$$

$$p(b) = \mathcal{N}(b|0, 1) \tag{34}$$

$$p(y|\boldsymbol{x}, \boldsymbol{w}, b) = \mathcal{N}\left(y|\boldsymbol{w}^T\boldsymbol{x} + b, \sigma^2\right) , \tag{35}$$

and with $\sigma^2 = 0.25$ known beforehand. Inputs $\boldsymbol{x}$ are generated from $p(\boldsymbol{x}) = \mathcal{N}(\mathbf{0}, I)$.

**Linear Classification (LC):** We now consider a setting where the true posterior cannot be obtained analytically as the likelihood and prior are not conjugate. In this case, we consider the underlying probabilistic model by:

$$p(\boldsymbol{W}) = \mathcal{N}\left(\boldsymbol{W}|\mathbf{0}, \mathbf{I}\right) \tag{36}$$

$$p(y|\boldsymbol{x}, \boldsymbol{W}) = \text{Categorical}\left(y \left| \frac{1}{\tau} \boldsymbol{W}\boldsymbol{x}\right.\right) , \tag{37}$$

where $\tau$ is the known temperature term which is kept as 0.1 to ensure peaky distributions, and $\boldsymbol{x}$ is being generated from $p(\boldsymbol{x}) = \mathcal{N}(\mathbf{0}, I)$.

**Nonlinear Regression (NLR):** Next, we tackle the more complex problem where the posterior distribution is multi-modal and obtaining multiple modes or even a single good one is challenging. For this, we consider the model as a Bayesian Neural Network (BNN) for regression with fixed hyper-parameters like the number of layers, dimensionality of the hidden layer, etc. Let the BNN denote the function $f_{\boldsymbol{\theta}}$ where $\boldsymbol{\theta}$ are the network parameters such that the estimation problem is to approximate $p(\boldsymbol{\theta}|\mathcal{D})$. Then, for regression, we specify the

| | | | $L_2$ Loss ($\downarrow$) | | Accuracy ($\uparrow$) | | | |
|---|---|---|---|---|---|---|---|---|
| | | | Linear Regression | | Linear Classification | | | |
| Objective | $q_\varphi$ | Model | 2D | 100D | 2D-2cl | 2D-5cl | 100D-2cl | 100D-5cl |
| Baseline | - | Random | $4.272_{\pm 0.068}$ | $200.836_{\pm 0.609}$ | $50.125_{\pm 0.264}$ | $20.078_{\pm 0.065}$ | $50.005_{\pm 0.061}$ | $20.033_{\pm 0.082}$ |
| | - | Optimization | $0.258_{\pm 0.000}$ | $20.127_{\pm 0.003}$ | $97.301_{\pm 0.000}$ | $91.752_{\pm 0.000}$ | $71.231_{\pm 0.010}$ | $42.345_{\pm 0.001}$ |
| | - | Langevin | $0.263_{\pm 0.002}$ | $21.781_{\pm 0.953}$ | $95.441_{\pm 0.209}$ | $86.445_{\pm 0.496}$ | $65.469_{\pm 0.513}$ | $32.668_{\pm 0.145}$ |
| | - | HMC | $0.263_{\pm 0.000}$ | $17.774_{\pm 0.120}$ | $92.961_{\pm 0.228}$ | $78.793_{\pm 0.314}$ | $62.602_{\pm 0.171}$ | $30.055_{\pm 0.506}$ |
| Fwd-KL | Gaussian | GRU | $0.271_{\pm 0.004}$ | $139.396_{\pm 1.012}$ | $79.467_{\pm 0.711}$ | $65.124_{\pm 0.861}$ | $57.872_{\pm 0.157}$ | $22.677_{\pm 0.081}$ |
| | | DeepSets | $0.269_{\pm 0.001}$ | $149.784_{\pm 0.766}$ | $80.323_{\pm 0.429}$ | $20.078_{\pm 0.059}$ | $50.767_{\pm 0.058}$ | $20.035_{\pm 0.081}$ |
| | | Transformer | $0.279_{\pm 0.001}$ | $64.282_{\pm 3.711}$ | $79.901_{\pm 0.271}$ | $60.984_{\pm 1.590}$ | $62.382_{\pm 0.029}$ | $26.997_{\pm 0.098}$ |
| Rev-KL | | GRU | $0.291_{\pm 0.013}$ | $62.624_{\pm 0.123}$ | $93.367_{\pm 0.289}$ | $82.020_{\pm 0.127}$ | $63.411_{\pm 0.248}$ | $28.655_{\pm 0.149}$ |
| | | DeepSets | $0.279_{\pm 0.004}$ | $64.064_{\pm 0.221}$ | $93.977_{\pm 0.093}$ | $83.832_{\pm 0.106}$ | $61.305_{\pm 0.114}$ | $27.877_{\pm 0.265}$ |
| | | Transformer | $0.271_{\pm 0.007}$ | $31.984_{\pm 0.482}$ | $94.336_{\pm 0.210}$ | $82.976_{\pm 0.074}$ | $67.676_{\pm 0.078}$ | $33.125_{\pm 0.051}$ |
| Fwd-KL | Flow | GRU | $0.273_{\pm 0.001}$ | $138.284_{\pm 1.030}$ | $92.078_{\pm 0.190}$ | $75.151_{\pm 0.274}$ | $57.982_{\pm 0.138}$ | $22.430_{\pm 0.208}$ |
| | | DeepSets | $0.270_{\pm 0.002}$ | $153.207_{\pm 0.814}$ | $85.950_{\pm 6.332}$ | $20.059_{\pm 0.233}$ | $50.494_{\pm 0.055}$ | $19.976_{\pm 0.095}$ |
| | | Transformer | $0.276_{\pm 0.004}$ | $63.102_{\pm 1.963}$ | $94.494_{\pm 0.368}$ | $74.876_{\pm 0.857}$ | $63.559_{\pm 0.072}$ | $27.098_{\pm 0.147}$ |
| Rev-KL | | GRU | $0.276_{\pm 0.008}$ | $71.260_{\pm 1.265}$ | $94.292_{\pm 0.175}$ | $83.622_{\pm 0.057}$ | $63.391_{\pm 0.133}$ | $27.340_{\pm 0.243}$ |
| | | DeepSets | $0.274_{\pm 0.003}$ | $76.772_{\pm 1.836}$ | $94.570_{\pm 0.178}$ | $85.059_{\pm 0.127}$ | $59.116_{\pm 0.491}$ | $22.810_{\pm 0.245}$ |
| | | Transformer | $0.279_{\pm 0.013}$ | $33.056_{\pm 0.321}$ | $94.793_{\pm 0.135}$ | $84.929_{\pm 0.027}$ | $68.124_{\pm 0.214}$ | $33.251_{\pm 0.130}$ |

Table 16: **Variable-Dimensional**. Results for estimating the parameters of linear regression (LR) and classification (LC) models with the expected $L_2$ loss and accuracy according to the posterior predictive as metrics.

probabilistic model using:

$$p(\boldsymbol{\theta}) = \mathcal{N}\left(\boldsymbol{\theta}|\mathbf{0}, \mathbf{I}\right) \tag{38}$$

$$p(y|\boldsymbol{x}, \boldsymbol{\theta}) = \mathcal{N}\left(y|f_{\boldsymbol{\theta}}(\boldsymbol{x}), \sigma^2\right) , \tag{39}$$

where $\sigma^2 = 0.25$ is a known quantity and $\boldsymbol{x}$ being generated from $p(\boldsymbol{x}) = \mathcal{N}(\mathbf{0}, I)$.

**Nonlinear Classification (NLC):** Like in Nonlinear Regression, we consider BNNs with fixed hyper-parameters for classification problems with the same estimation task of approximating $p(\boldsymbol{\theta}|\mathcal{D})$. In this formulation, we consider the probabilistic model as:

$$p(\boldsymbol{\theta}) = \mathcal{N}\left(\boldsymbol{\theta}|\mathbf{0}, \mathbf{I}\right) \tag{40}$$

$$p(y|\boldsymbol{x}, \boldsymbol{\theta}) = \text{Categorical}\left(y \left| \frac{1}{\tau} f_{\boldsymbol{\theta}}(\boldsymbol{x})\right.\right) \tag{41}$$

where $\tau$ is the known temperature term which is kept as 0.1 to ensure peaky distributions, and $\boldsymbol{x}$ is being generated from $p(\boldsymbol{x}) = \mathcal{N}(\mathbf{0}, I)$.

**Gaussian Mixture Model (GMM):** While we have mostly looked at predictive problems, where the task is to model some predictive variable $y$ conditioned on some input $\boldsymbol{x}$, we now look at a well-known probabilistic model for unsupervised learning, Gaussian Mixture Model (GMM), primarily used to cluster data. Consider a $K$-cluster GMM with:

$$p(\boldsymbol{\mu}_k) = \mathcal{N}\left(\boldsymbol{\mu}_k|\mathbf{0}, \mathbf{I}\right) \tag{42}$$

$$p(\boldsymbol{x}|\boldsymbol{\mu}_{1:K}) = \sum_{k=1}^{K} \pi_k \mathcal{N}\left(\boldsymbol{x}|\boldsymbol{\mu}_k, \boldsymbol{\Sigma}_k\right) . \tag{43}$$

| | | | $L_2$ Loss ($\downarrow$) | | | |
|---|---|---|---|---|---|---|
| | | | *1-layer* | | *2-layers* | |
| **Objective** | $q_\varphi$ | **Model** | *1D* | *50D* | *1D* | *50D* |
| Baseline | - | Random | $73.006_{\pm0.171}$ | $1704.335_{\pm9.330}$ | $998.890_{\pm16.317}$ | $27799.887_{\pm165.653}$ |
| | - | Optimization | $0.359_{\pm0.002}$ | $309.162_{\pm0.204}$ | $3.078_{\pm0.057}$ | $4894.141_{\pm4.290}$ |
| | - | Langevin | $0.308_{\pm0.002}$ | N/A | N/A | N/A |
| | - | HMC | $0.381_{\pm0.005}$ | $303.857_{\pm2.487}$ | $7.999_{\pm0.595}$ | $12905.846_{\pm9.903}$ |
| Fwd-KL | Gaussian | GRU | $51.448_{\pm0.029}$ | $1346.462_{\pm6.833}$ | $754.388_{\pm32.650}$ | $21121.735_{\pm112.458}$ |
| | | DeepSets | $51.901_{\pm1.542}$ | $1357.462_{\pm5.348}$ | $767.781_{\pm18.262}$ | $21110.264_{\pm72.613}$ |
| | | Transformer | $50.813_{\pm0.532}$ | $1319.868_{\pm12.165}$ | $753.177_{\pm23.023}$ | $21037.666_{\pm116.929}$ |
| Rev-KL | | GRU | $2.308_{\pm0.126}$ | $316.344_{\pm6.189}$ | $23.520_{\pm3.816}$ | $4673.633_{\pm98.443}$ |
| | | DeepSets | $0.977_{\pm0.118}$ | $451.942_{\pm2.785}$ | $8.034_{\pm0.375}$ | $6127.317_{\pm190.224}$ |
| | | Transformer | $0.815_{\pm0.024}$ | $278.282_{\pm1.073}$ | $8.300_{\pm1.673}$ | $4744.375_{\pm24.609}$ |
| Fwd-KL | Flow | GRU | $38.407_{\pm0.364}$ | $1097.410_{\pm9.498}$ | $664.409_{\pm9.590}$ | $18372.905_{\pm62.019}$ |
| | | DeepSets | $43.305_{\pm2.063}$ | $1119.990_{\pm5.461}$ | $745.412_{\pm27.599}$ | $20719.466_{\pm627.997}$ |
| | | Transformer | $39.701_{\pm0.519}$ | $1073.256_{\pm1.504}$ | $619.631_{\pm22.299}$ | $16700.463_{\pm350.596}$ |
| Rev-KL | | GRU | $2.305_{\pm0.406}$ | $302.918_{\pm5.644}$ | $13.729_{\pm1.445}$ | $4832.392_{\pm50.014}$ |
| | | DeepSets | $0.827_{\pm0.024}$ | $454.141_{\pm10.203}$ | $5.889_{\pm0.135}$ | $7589.795_{\pm373.293}$ |
| | | Transformer | $0.985_{\pm0.075}$ | $274.021_{\pm1.333}$ | $6.364_{\pm0.201}$ | $4801.964_{\pm59.175}$ |

Table 17: **Variable-Dimensional**. Results for estimating the parameters of nonlinear regression models with ReLU activation function, with the expected $L_2$ loss according to the posterior predictive as metric.

We assume $\mathbf{\Sigma}_k$ and $\pi_k$ to be known and set $\mathbf{\Sigma}_k$ to be an identity matrix and the mixing coefficients to be equal, $\pi_k = 1/K$, for all clusters $k$ in our experiments.

# D  Metrics

In this section, we provide details about the metrics considered for the different tasks. We generally look at two main metrics for benchmarking performance: $L_2$ loss and Accuracy. For estimating the mean of a Gaussian distribution, the $L_2$ loss is defined as

$$GM_{L_2} = \mathbb{E}_{\mathcal{D}\sim\chi}\mathbb{E}_{\boldsymbol{\mu}\sim q_\varphi(\cdot|\mathcal{D})}\left[\sum_{i=1}^{N_\mathcal{D}}(\boldsymbol{x}_i - \boldsymbol{\mu})^2\right] \tag{44}$$

where $\mathcal{D} = \{\boldsymbol{x}_i\}_{i=1}^{N_\mathcal{D}}$. Intuitively, this captures the quality of the estimation of the mean parameter by measuring how far the observations are from it. Lower value implies better estimation of the mean parameter. Similarly, for estimating the means of a Gaussian Mixture Model, we rely on a similar metric but we also find the cluster closest to the observation, which can be defined as

$$GMM_{L_2} = \mathbb{E}_{\mathcal{D}\sim\chi}\mathbb{E}_{\boldsymbol{\mu}_k\sim q_\varphi(\cdot|\mathcal{D})}\left[\sum_{i=1}^{N_\mathcal{D}}(\boldsymbol{x}_i - \boldsymbol{\mu}_{\text{Match}(\boldsymbol{x}_i,\{\boldsymbol{\mu}_1,...\boldsymbol{\mu}_K\})})^2\right] \tag{45}$$

$$\text{Match}(\boldsymbol{x}, \{\boldsymbol{\mu}_1, ..., \boldsymbol{\mu}_K\}) = \arg\min_k(\boldsymbol{x} - \boldsymbol{\mu}_k)^2 \tag{46}$$

| Objective | $q_\varphi$ | Model | $L_2$ Loss ($\downarrow$) | | | |
| --- | --- | --- | --- | --- | --- | --- |
| | | | *1-layer* | | *2-layers* | |
| | | | *1D* | *50D* | *1D* | *50D* |
| Baseline | - | Random | $33.972_{\pm0.273}$ | $56.891_{\pm0.213}$ | $50.843_{\pm0.270}$ | $55.383_{\pm0.124}$ |
| | - | Optimization | $0.330_{\pm0.000}$ | $23.257_{\pm0.029}$ | $0.672_{\pm0.007}$ | $34.900_{\pm0.120}$ |
| | - | Langevin | $0.296_{\pm0.003}$ | $26.292_{\pm0.266}$ | $0.356_{\pm0.006}$ | $36.598_{\pm0.354}$ |
| | - | HMC | $0.404_{\pm0.005}$ | $19.937_{\pm0.259}$ | $0.654_{\pm0.014}$ | $32.884_{\pm0.448}$ |
| Fwd-KL | Gaussian | GRU | $34.235_{\pm0.240}$ | $56.947_{\pm0.339}$ | $50.493_{\pm0.676}$ | $55.633_{\pm0.396}$ |
| | | DeepSets | $33.979_{\pm0.273}$ | $56.900_{\pm0.213}$ | $50.844_{\pm0.284}$ | $55.390_{\pm0.132}$ |
| | | Transformer | $33.998_{\pm0.499}$ | $56.365_{\pm0.129}$ | $49.940_{\pm0.317}$ | $55.708_{\pm0.149}$ |
| Rev-KL | | GRU | $0.813_{\pm0.036}$ | $18.042_{\pm0.084}$ | $17.506_{\pm10.364}$ | $26.729_{\pm2.167}$ |
| | | DeepSets | $0.604_{\pm0.015}$ | $22.582_{\pm2.064}$ | $24.819_{\pm0.009}$ | $28.247_{\pm0.006}$ |
| | | Transformer | $0.896_{\pm0.097}$ | $16.726_{\pm0.724}$ | $2.249_{\pm0.651}$ | $24.442_{\pm0.643}$ |
| Fwd-KL | Flow | GRU | $34.989_{\pm0.475}$ | $56.847_{\pm0.289}$ | $49.915_{\pm0.980}$ | $55.505_{\pm0.479}$ |
| | | DeepSets | $34.857_{\pm0.278}$ | $56.736_{\pm0.395}$ | $49.594_{\pm0.622}$ | $55.862_{\pm0.498}$ |
| | | Transformer | $34.878_{\pm0.829}$ | $55.751_{\pm0.409}$ | $49.309_{\pm0.651}$ | $55.475_{\pm0.090}$ |
| Rev-KL | | GRU | $0.969_{\pm0.038}$ | $17.454_{\pm0.041}$ | $24.796_{\pm0.028}$ | $28.258_{\pm0.008}$ |
| | | DeepSets | $0.729_{\pm0.013}$ | $22.888_{\pm2.610}$ | $24.794_{\pm0.021}$ | $28.253_{\pm0.003}$ |
| | | Transformer | $0.624_{\pm0.051}$ | $15.971_{\pm0.075}$ | $3.095_{\pm0.010}$ | $23.740_{\pm0.343}$ |

Table 18: **Variable-Dimensional**. Results for estimating the parameters of nonlinear regression models with TanH activation function, with the expected $L_2$ loss according to the posterior predictive as metric.

which intuitively captures the distance of observations from the cluster closest to them. Next, we define the metric for evaluating (non-)linear regression models as

$$(N-)LR_{L_2} = \mathbb{E}_{\mathcal{D}\sim\chi}\mathbb{E}_{\boldsymbol{\theta}\sim q_\varphi(\cdot|\mathcal{D})}\left[\sum_{i=1}^{N_\mathcal{D}}(y_i - \text{Mode}\left[p(y_i|\boldsymbol{x}_i,\boldsymbol{\theta})\right])^2\right] \tag{47}$$

Finally, for the (non-)linear classification setups, we define the accuracy metric as

$$(N-)LC_{Accuracy} = \mathbb{E}_{\mathcal{D}\sim\chi}\mathbb{E}_{\boldsymbol{\theta}\sim q_\varphi(\cdot|\mathcal{D})}\left[\frac{100}{N_\mathcal{D}} \times \sum_{i=1}^{N_\mathcal{D}}\delta(y_i, \text{Mode}\left[p(y_i|\boldsymbol{x}_i,\boldsymbol{\theta})\right])\right] \tag{48}$$

where $\delta(a,b) = 1$ if and only if $a = b$. Thus this metric captures the accuracy of the posterior predictive distribution. Another metric that we use to test the quality of the posterior is the symmetric KL divergence, defined as

$$\text{Symmetric } \mathbb{KL}(p(\boldsymbol{\theta}||\mathcal{D}), q_\varphi(\boldsymbol{\theta}|\mathcal{D})) = \frac{1}{2}\mathbb{KL}(p(\boldsymbol{\theta}||\mathcal{D})||q_\varphi(\boldsymbol{\theta}|\mathcal{D})) + \frac{1}{2}\mathbb{KL}(q_\varphi(\boldsymbol{\theta}|\mathcal{D})||p(\boldsymbol{\theta}||\mathcal{D})) \tag{49}$$

# E   Architecture Details

In this section, we outline the two candidate architectures that we consider for the backbone of our amortized variational inference model. We discuss the specifics of the architectures and the hyperparameters used for our experiments.

## E.1   Transformer

We use a transformer model (Vaswani et al., 2017) as a permutation invariant architecture by removing positional encodings from the setup and using multiple layers of the encoder model. We append the set of

| Objective | $q_\varphi$ | Model | Accuracy ($\uparrow$) | | | |
| | | | 1-layer | | 2-layers | |
| | | | 2D | 50D | 2D | 50D |
|---|---|---|---|---|---|---|
| Baseline | - | Random | $49.951_{\pm 0.287}$ | $49.904_{\pm 0.281}$ | $50.040_{\pm 0.467}$ | $50.044_{\pm 0.239}$ |
| | - | Optimization | $96.762_{\pm 0.034}$ | $76.139_{\pm 0.033}$ | $96.810_{\pm 0.009}$ | $78.225_{\pm 0.129}$ |
| | - | Langevin | $96.077_{\pm 0.027}$ | $70.142_{\pm 0.229}$ | $96.564_{\pm 0.113}$ | $71.328_{\pm 0.410}$ |
| | - | HMC | $91.734_{\pm 0.152}$ | $67.986_{\pm 0.372}$ | $91.336_{\pm 0.591}$ | $71.825_{\pm 0.507}$ |
| Fwd-KL | Gaussian | GRU | $59.551_{\pm 0.199}$ | $58.637_{\pm 0.250}$ | $60.247_{\pm 0.645}$ | $58.862_{\pm 0.065}$ |
| | | DeepSets | $49.946_{\pm 0.285}$ | $49.910_{\pm 0.282}$ | $50.032_{\pm 0.466}$ | $50.040_{\pm 0.242}$ |
| | | Transformer | $59.887_{\pm 0.235}$ | $58.826_{\pm 0.237}$ | $60.552_{\pm 0.398}$ | $58.953_{\pm 0.110}$ |
| Rev-KL | | GRU | $88.822_{\pm 0.471}$ | $68.368_{\pm 0.342}$ | $81.884_{\pm 1.450}$ | $67.264_{\pm 0.138}$ |
| | | DeepSets | $91.019_{\pm 0.454}$ | $61.732_{\pm 0.111}$ | $82.396_{\pm 0.471}$ | $67.320_{\pm 0.143}$ |
| | | Transformer | $89.988_{\pm 0.197}$ | $73.744_{\pm 0.319}$ | $83.399_{\pm 0.841}$ | $67.167_{\pm 0.028}$ |
| Fwd-KL | Flow | GRU | $61.179_{\pm 0.833}$ | $60.225_{\pm 0.115}$ | $60.400_{\pm 1.019}$ | $59.027_{\pm 0.212}$ |
| | | DeepSets | $49.568_{\pm 0.230}$ | $50.130_{\pm 0.101}$ | $50.356_{\pm 0.773}$ | $49.806_{\pm 0.331}$ |
| | | Transformer | $60.886_{\pm 0.252}$ | $60.253_{\pm 0.082}$ | $61.694_{\pm 0.314}$ | $60.426_{\pm 0.203}$ |
| Rev-KL | | GRU | $90.363_{\pm 0.709}$ | $66.197_{\pm 0.118}$ | $83.443_{\pm 0.619}$ | $69.053_{\pm 0.256}$ |
| | | DeepSets | $89.150_{\pm 0.338}$ | $62.939_{\pm 0.112}$ | $79.889_{\pm 0.567}$ | $69.015_{\pm 0.147}$ |
| | | Transformer | $91.065_{\pm 0.156}$ | $72.581_{\pm 0.117}$ | $83.533_{\pm 0.677}$ | $68.933_{\pm 0.120}$ |

Table 19: **Variable-Dimensional**. Results for estimating the parameters of nonlinear classification models with ReLU activation function and two classes, with the expected accuracy according to the posterior predictive as metric.

observations with a [CLS] token before passing it to the model and use its output embedding to predict the parameters of the variational distribution. Since no positional encodings or causal masking is used in the whole setup, the final embedding of the [CLS] token becomes invariant to permutations in the set of observations, thereby leading to permutation invariance in the parameters of $q_\varphi$.

We use 4 encoder layers with a 256 dimensional attention block and 1024 feed-forward dimensions, with 4 heads in each attention block for our Transformer models to make the number of parameters comparative to the one of the DeepSets model.

## E.2 DeepSets

Another framework that can process set-based input is Deep Sets (Zaheer et al., 2017). In our experiments, we used an embedding network that encodes the input into representation space, a mean aggregation operation, which ensures that the representation learned is invariant concerning the set ordering, and a regression network. The latter's output is either used to directly parameterize a diagonal Gaussian or as conditional input to a normalizing flow, representing a summary statistics of the set input.

For DeepSets, we use 4 layers each in the embedding network and the regression network, with a mean aggregation function, ReLU activation functions, and 627 hidden dimensions to make the number of parameters comparable to those in the Transformer model.

## E.3 RNN

For the recurrent neural network setup, we use the Gated Recurrent Unit (GRU). Similar to the above setups, we use a 4-layered GRU model with 256 hidden dimensions. While such an architecture is not permutation

| | | | Accuracy ($\uparrow$) | | | |
| | | | 1-layer | | 2-layers | |
| **Objective** | $q_\varphi$ | **Model** | 2D | 50D | 2D | 50D |
|---|---|---|---|---|---|---|
| Baseline | - | Random | $19.847_{\pm 0.352}$ | $20.052_{\pm 0.066}$ | $19.874_{\pm 0.222}$ | $20.028_{\pm 0.106}$ |
| | - | Optimization | $94.607_{\pm 0.011}$ | $56.091_{\pm 0.158}$ | $93.873_{\pm 0.028}$ | $60.253_{\pm 0.053}$ |
| | - | Langevin | $90.815_{\pm 0.341}$ | $46.072_{\pm 0.225}$ | $91.849_{\pm 0.088}$ | $49.808_{\pm 0.337}$ |
| | - | HMC | $81.145_{\pm 0.303}$ | $44.561_{\pm 0.309}$ | $79.559_{\pm 0.483}$ | $50.967_{\pm 0.436}$ |
| Fwd-KL | Gaussian | GRU | $30.960_{\pm 0.638}$ | $32.164_{\pm 0.151}$ | $31.017_{\pm 0.397}$ | $32.224_{\pm 0.108}$ |
| | | DeepSets | $19.846_{\pm 0.348}$ | $20.053_{\pm 0.063}$ | $19.871_{\pm 0.226}$ | $20.032_{\pm 0.104}$ |
| | | Transformer | $30.652_{\pm 0.401}$ | $32.208_{\pm 0.178}$ | $31.148_{\pm 0.429}$ | $32.342_{\pm 0.217}$ |
| Rev-KL | | GRU | $72.874_{\pm 0.113}$ | $37.987_{\pm 0.119}$ | $56.999_{\pm 0.599}$ | $29.971_{\pm 0.248}$ |
| | | DeepSets | $69.456_{\pm 0.370}$ | $36.712_{\pm 0.249}$ | $55.193_{\pm 0.538}$ | $36.417_{\pm 9.779}$ |
| | | Transformer | $73.531_{\pm 0.391}$ | $44.702_{\pm 0.165}$ | $57.724_{\pm 0.332}$ | $30.175_{\pm 0.177}$ |
| Fwd-KL | Flow | GRU | $33.232_{\pm 0.607}$ | $33.704_{\pm 0.026}$ | $31.937_{\pm 0.483}$ | $32.370_{\pm 0.284}$ |
| | | DeepSets | $19.898_{\pm 0.156}$ | $19.950_{\pm 0.256}$ | $20.062_{\pm 0.347}$ | $20.064_{\pm 0.207}$ |
| | | Transformer | $32.916_{\pm 0.194}$ | $33.766_{\pm 0.137}$ | $32.374_{\pm 0.301}$ | $33.846_{\pm 0.486}$ |
| Rev-KL | | GRU | $77.997_{\pm 0.663}$ | $38.715_{\pm 0.153}$ | $61.947_{\pm 0.294}$ | $51.962_{\pm 0.812}$ |
| | | DeepSets | $68.957_{\pm 0.551}$ | $37.123_{\pm 0.108}$ | $51.145_{\pm 14.201}$ | $42.707_{\pm 12.882}$ |
| | | Transformer | $77.867_{\pm 2.241}$ | $44.156_{\pm 0.485}$ | $57.410_{\pm 0.088}$ | $52.077_{\pm 0.077}$ |

Table 20: **Fixed-Dimensional**. Results for estimating the parameters of nonlinear classification models with ReLU activation function and five classes, with the expected accuracy according to the posterior predictive as metric.

invariant, by training on tasks that require such invariance could encourage learning of solution structure that respects this invariance.

### E.4 Normalizing Flows

Assuming a Gaussian posterior distribution as the approximate often leads to poor results as the true posterior distribution can be far from the Gaussian shape. To allow for more flexible posterior distributions, we use normalizing flows (Kingma & Dhariwal, 2018; Kobyzev et al., 2020; Papamakarios et al., 2021; Rezende & Mohamed, 2015) for approximating $q_\varphi(\boldsymbol{\theta}|\mathcal{D})$ conditioned on the output of the summary network $h_\psi$. Specifically, let $g_\nu : \boldsymbol{z} \mapsto \boldsymbol{\theta}$ be a diffeomorphism parameterized by a conditional invertible neural network (cINN) with network parameters $\nu$ such that $\boldsymbol{\theta} = g_\nu(\boldsymbol{z}; h_\psi(\mathcal{D}))$. With the change-of-variables formula it follows that $p(\boldsymbol{\theta}) = p(\boldsymbol{z}) \left| \det \frac{\partial}{\partial \boldsymbol{z}} g_\nu(\boldsymbol{z}; h_\psi(\mathcal{D})) \right|^{-1} = p(\boldsymbol{z}) |\det J_\nu(\boldsymbol{z}; h_\psi(\mathcal{D}))|^{-1}$, where $J_\nu$ is the Jacobian matrix of $g_\nu$. Further, integration by substitution gives us $d\boldsymbol{\theta} = |\det J_\nu(\boldsymbol{z}; h_\psi(\mathcal{D})| d\boldsymbol{z}$ to rewrite the objective from eq. 12 as:

$$\arg \min_{\varphi} \mathbb{KL}[q_\varphi(\boldsymbol{\theta}|\mathcal{D})||p(\boldsymbol{\theta}|\mathcal{D})] \tag{50}$$

$$= \arg \min_{\varphi} \mathbb{E}_{\mathcal{D} \sim \chi} \mathbb{E}_{\boldsymbol{\theta} \sim q_\varphi(\boldsymbol{\theta}|\mathcal{D})} [\log q_\varphi(\boldsymbol{\theta}|\mathcal{D}) - \log p(\boldsymbol{\theta}, \mathcal{D})] \tag{51}$$

$$= \arg \min_{\varphi = \{\psi, \nu\}} \mathbb{E}_{\mathcal{D} \sim \chi} \mathbb{E}_{\boldsymbol{z} \sim p(\boldsymbol{z})} \left[ \log \frac{q_\nu(\boldsymbol{z}|h_\psi(\mathcal{D}))}{|\det J_\nu(\boldsymbol{z}; h_\psi(\mathcal{D}))|} - \log p(g_\nu(\boldsymbol{z}; h_\psi(\mathcal{D})), \mathcal{D}) \right] \tag{52}$$

As shown in BayesFlow (Radev et al., 2020), the normalizing flow $g_\nu$ and the summary network $h_\psi$ can be trained simultaneously. The AllInOneBlock coupling block architecture of the FrEIA Python package (Ardizzone et al., 2018), which is very similar to the RNVP style coupling block (Dinh et al., 2017), is used

| Objective | $q_\varphi$ | Model | Accuracy (↑) | | | |
| | | | 1-layer | | 2-layers | |
| | | | 2D | 50D | 2D | 50D |
|---|---|---|---|---|---|---|
| Baseline | - | Random | $50.278_{\pm 0.337}$ | $50.028_{\pm 0.064}$ | $50.188_{\pm 0.479}$ | $49.982_{\pm 0.084}$ |
| | - | Optimization | $96.943_{\pm 0.012}$ | $68.086_{\pm 0.016}$ | $94.444_{\pm 0.024}$ | $63.950_{\pm 0.022}$ |
| | - | Langevin | $95.143_{\pm 0.094}$ | $61.694_{\pm 0.340}$ | $92.719_{\pm 0.016}$ | $57.447_{\pm 0.437}$ |
| | - | HMC | $92.489_{\pm 0.338}$ | $59.963_{\pm 0.202}$ | $87.548_{\pm 0.094}$ | $56.319_{\pm 0.882}$ |
| Fwd-KL | Gaussian | GRU | $50.274_{\pm 0.337}$ | $50.023_{\pm 0.060}$ | $50.187_{\pm 0.477}$ | $49.992_{\pm 0.072}$ |
| | | DeepSets | $50.271_{\pm 0.334}$ | $50.024_{\pm 0.061}$ | $50.188_{\pm 0.472}$ | $49.984_{\pm 0.077}$ |
| | | Transformer | $50.273_{\pm 0.336}$ | $50.031_{\pm 0.066}$ | $50.191_{\pm 0.471}$ | $49.994_{\pm 0.078}$ |
| Rev-KL | | GRU | $89.270_{\pm 0.272}$ | $50.014_{\pm 0.048}$ | $50.191_{\pm 0.463}$ | $49.995_{\pm 0.080}$ |
| | | DeepSets | $89.788_{\pm 0.213}$ | $50.018_{\pm 0.061}$ | $50.191_{\pm 0.478}$ | $49.977_{\pm 0.075}$ |
| | | Transformer | $89.366_{\pm 0.108}$ | $64.926_{\pm 0.260}$ | $50.182_{\pm 0.469}$ | $49.986_{\pm 0.071}$ |
| Fwd-KL | Flow | GRU | $49.651_{\pm 0.040}$ | $50.119_{\pm 0.068}$ | $49.987_{\pm 0.015}$ | $49.904_{\pm 0.018}$ |
| | | DeepSets | $49.639_{\pm 0.031}$ | $50.113_{\pm 0.065}$ | $49.988_{\pm 0.022}$ | $49.910_{\pm 0.043}$ |
| | | Transformer | $49.636_{\pm 0.040}$ | $50.115_{\pm 0.063}$ | $49.989_{\pm 0.017}$ | $49.909_{\pm 0.042}$ |
| Rev-KL | | GRU | $49.769_{\pm 0.141}$ | $50.082_{\pm 0.091}$ | $49.915_{\pm 0.070}$ | $50.004_{\pm 0.084}$ |
| | | DeepSets | $49.782_{\pm 0.073}$ | $50.080_{\pm 0.087}$ | $49.831_{\pm 0.152}$ | $49.994_{\pm 0.078}$ |
| | | Transformer | $63.233_{\pm 19.243}$ | $50.026_{\pm 0.047}$ | $49.869_{\pm 0.207}$ | $50.036_{\pm 0.056}$ |

Table 21: **Variable-Dimensional**. Results for estimating the parameters of nonlinear classification models with TanH activation function and two classes, with the expected accuracy according to the posterior predictive as metric.

as the basis for the cINN. AllInOneBlock combines the most common architectural components, such as ActNorm, permutation, and affine coupling operations.

For our experiments, 6 coupling blocks define the normalizing flow network, each with a 1 hidden-layered non-linear feed-forward subnetwork with ReLU non-linearity and 128 hidden dimensions.

# F Experimental Details

Unless specified, we obtain a stream of datasets for all our experiments by simply sampling from the assumed probabilistic model, where the number of observations $n$ is sampled uniformly in the range $[64, 128]$. For efficient mini-batching over datasets with different cardinalities, we sample datasets with maximum cardinality (128) and implement different cardinalities by masking out different numbers of observations for different datasets whenever required.

To evaluate both our proposed approach and the baselines, we compute an average of the predictive performances across 25 different posterior samples for each of the 100 fixed test datasets for all our experiments. That means for our proposed approach, we sample 25 different parameter vectors from the approximate posterior that we obtain. For MCMC, we rely on 25 MCMC samples, and for optimization, we train 25 different parameter vectors where the randomness comes from initialization. For the optimization baseline, we perform a quick hyperparameter search over the space $\{0.01, 003, 0.001, 0.0003, 0.0001, 0.00003\}$ to pick the best learning rate that works for all of the test datasets and then use it to train for 1000 iterations using the Adam optimizer (Kingma & Ba, 2014). For the MCMC baseline, we use the open-sourced implementation of Langevin-based MCMC sampling[2] where we leave a chunk of the starting samples as burn-in and then

---

[2]https://github.com/alisiahkoohi/Langevin-dynamics

| | | | Accuracy ($\uparrow$) | | | |
|---|---|---|---|---|---|---|
| | | | 1-layer | | 2-layers | |
| **Objective** | $q_\varphi$ | **Model** | 2D | 50D | 2D | 50D |
| Baseline | - | Random | $20.041_{\pm 0.136}$ | $20.002_{\pm 0.079}$ | $19.914_{\pm 0.111}$ | $19.954_{\pm 0.005}$ |
| | - | Optimization | $92.059_{\pm 0.012}$ | $40.722_{\pm 0.027}$ | $88.848_{\pm 0.005}$ | $34.136_{\pm 0.041}$ |
| | - | Langevin | $88.357_{\pm 0.309}$ | $30.941_{\pm 0.097}$ | $83.788_{\pm 0.170}$ | $26.538_{\pm 0.206}$ |
| | - | HMC | $79.161_{\pm 0.292}$ | $27.508_{\pm 0.379}$ | $74.987_{\pm 0.130}$ | $25.377_{\pm 0.246}$ |
| Fwd-KL | Gaussian | GRU | $20.264_{\pm 0.139}$ | $20.158_{\pm 0.056}$ | $20.215_{\pm 0.138}$ | $20.093_{\pm 0.028}$ |
| | | DeepSets | $20.042_{\pm 0.133}$ | $20.000_{\pm 0.088}$ | $19.916_{\pm 0.114}$ | $19.955_{\pm 0.007}$ |
| | | Transformer | $20.240_{\pm 0.126}$ | $20.153_{\pm 0.070}$ | $20.118_{\pm 0.131}$ | $20.089_{\pm 0.022}$ |
| Rev-KL | | GRU | $66.565_{\pm 7.725}$ | $20.011_{\pm 0.099}$ | $19.913_{\pm 0.125}$ | $19.954_{\pm 0.022}$ |
| | | DeepSets | $57.294_{\pm 0.362}$ | $20.011_{\pm 0.091}$ | $19.915_{\pm 0.115}$ | $19.959_{\pm 0.020}$ |
| | | Transformer | $72.865_{\pm 1.340}$ | $22.185_{\pm 1.872}$ | $19.911_{\pm 0.124}$ | $19.953_{\pm 0.014}$ |
| Fwd-KL | Flow | GRU | $19.963_{\pm 0.239}$ | $20.176_{\pm 0.056}$ | $19.952_{\pm 0.189}$ | $20.156_{\pm 0.079}$ |
| | | DeepSets | $19.757_{\pm 0.259}$ | $20.045_{\pm 0.071}$ | $19.692_{\pm 0.184}$ | $20.019_{\pm 0.069}$ |
| | | Transformer | $19.925_{\pm 0.262}$ | $20.185_{\pm 0.059}$ | $19.882_{\pm 0.175}$ | $20.159_{\pm 0.070}$ |
| Rev-KL | | GRU | $67.042_{\pm 2.230}$ | $20.065_{\pm 0.060}$ | $19.707_{\pm 0.245}$ | $19.989_{\pm 0.101}$ |
| | | DeepSets | $35.220_{\pm 10.870}$ | $20.000_{\pm 0.054}$ | $19.739_{\pm 0.216}$ | $19.966_{\pm 0.019}$ |
| | | Transformer | $72.798_{\pm 1.049}$ | $20.032_{\pm 0.040}$ | $19.752_{\pm 0.276}$ | $20.017_{\pm 0.037}$ |

Table 22: **Variable-Dimensional**. Results for estimating the parameters of nonlinear classification models with TanH activation function and five classes, with the expected accuracy according to the posterior predictive as metric.

start accepting samples after a regular interval (to not make them correlated). The details about the burn-in time and the regular interval for acceptance are provided in the corresponding experiments' sections below.

For our proposed approach of amortized inference, we do not consider explicit hyperparameter optimization and simply use a learning rate of 1e-4 with the Adam optimizer. For all experiments, we used linear scaling of the KL term in the training objectives as described in (Higgins et al., 2017), which we refer to as warmup. Furthermore, training details for each experiment can be found below.

### F.1 Fixed-Dim

In this section, we provide the experimental details relevant to reproducing the results of Section 4. All the models are trained with streaming data from the underlying probabilistic model, such that every iteration of training sees a new set of datasets. Training is done with a batch size of 128, representing the number of datasets seen during one optimization step. Evaluations are done with 25 samples and we ensure that the test datasets used for each probabilistic model are the same across all the compared methods, i.e., baselines, forward KL, and reverse KL. We train the amortized inference model and the forward KL baselines for the following different probabilistic models:

**Mean of Gaussian (GM):** We train the amortization models over $20,000$ iterations for both the 2-dimensional as well as the 100-dimensional setup. We use a linear warmup with 5000 iterations over which the weight of the KL term in our proposed approach scales linearly from 0 to 1. We use an identity covariance matrix for the data-generating process, but it can be easily extended to the case of correlated or diagonal covariance-based Gaussian distributions.

**Gaussian Mixture Model (GMM):** We train the mixture model setup for $200,000$ iterations with $50,000$ iterations of warmup. We mainly experiment with 2-dimensional and 5-dimensional mixture models,

| Objective | $q_\varphi$ | Model | $L_2$ Loss ($\downarrow$) | | | | | | $\leftarrow \chi_{real}$ |
|---|---|---|---|---|---|---|---|---|---|
| | | | Linear Model \| MLP-TanH Data | | | MLP-TanH Model \| Linear Data | | | $\leftarrow \chi_{sim}$ |
| | | | *LR* | *NLR* | *GP* | *LR* | *NLR* | *GP* | |
| Baseline | - | Random | - | $17.761_{\pm 0.074}$ | - | $17.847_{\pm 0.355}$ | - | - | |
| | - | Optimization | - | $1.213_{\pm 0.000}$ | - | $0.360_{\pm 0.001}$ | - | - | |
| | - | Langevin | - | $1.218_{\pm 0.002}$ | - | $0.288_{\pm 0.001}$ | - | - | |
| | - | HMC | - | $1.216_{\pm 0.002}$ | - | $0.275_{\pm 0.001}$ | - | - | |
| Fwd-KL | Gaussian | GRU | $2.415_{\pm 0.269}$ | - | - | - | $15.632_{\pm 0.283}$ | - | |
| | | DeepSets | $1.402_{\pm 0.017}$ | - | - | - | $16.046_{\pm 0.393}$ | - | |
| | | Transformer | $2.216_{\pm 0.097}$ | - | - | - | $15.454_{\pm 0.246}$ | - | |
| Rev-KL | | GRU | $1.766_{\pm 0.044}$ | $1.216_{\pm 0.001}$ | $4.566_{\pm 0.199}$ | $0.375_{\pm 0.001}$ | $0.386_{\pm 0.002}$ | $0.524_{\pm 0.019}$ | |
| | | DeepSets | $1.237_{\pm 0.006}$ | $1.216_{\pm 0.001}$ | $3963.694_{\pm 5602.411}$ | $0.365_{\pm 0.000}$ | $0.377_{\pm 0.003}$ | $0.385_{\pm 0.011}$ | |
| | | Transformer | $1.892_{\pm 0.113}$ | $1.226_{\pm 0.001}$ | $4.313_{\pm 0.707}$ | $0.367_{\pm 0.006}$ | $0.382_{\pm 0.003}$ | $0.458_{\pm 0.048}$ | |
| Fwd-KL | Flow | GRU | $2.180_{\pm 0.024}$ | - | - | - | $9.800_{\pm 0.473}$ | - | |
| | | DeepSets | $1.713_{\pm 0.244}$ | - | - | - | $15.253_{\pm 0.403}$ | - | |
| | | Transformer | $1.632_{\pm 0.070}$ | - | - | - | $7.949_{\pm 0.419}$ | - | |
| Rev-KL | | GRU | $1.830_{\pm 0.081}$ | $1.214_{\pm 0.001}$ | $5.690_{\pm 0.196}$ | $0.346_{\pm 0.004}$ | $0.349_{\pm 0.001}$ | $0.520_{\pm 0.015}$ | |
| | | DeepSets | $1.282_{\pm 0.036}$ | $1.218_{\pm 0.001}$ | $11.690_{\pm 10.602}$ | $0.339_{\pm 0.003}$ | $0.344_{\pm 0.002}$ | $0.397_{\pm 0.026}$ | |
| | | Transformer | $1.471_{\pm 0.016}$ | $1.226_{\pm 0.004}$ | $5.194_{\pm 0.320}$ | $0.346_{\pm 0.002}$ | $0.347_{\pm 0.001}$ | $0.480_{\pm 0.030}$ | |

Table 23: **Model Misspecification**. Results for model misspecification under different training data $\chi_{sim}$, when evaluated under MLP-TanH and Linear Data ($\chi_{real}$), with the underlying model as a linear and MLP-TanH model respectively.

with 2 and 5 mixture components for each setup. While we do use an identity covariance matrix for the data-generating process, again, it can be easily extended to other cases.

**Linear Regression (LR):** The amortization models for this setup are trained for $50,000$ iterations with $12,500$ iterations of warmup. The feature dimensions considered for this task are 1 and 100 dimensions, and the predictive variance $\sigma^2$ is assumed to be known and set as $0.25$.

**Nonlinear Regression (NLR):** We train the setup for $100,000$ iterations with $25,000$ iterations consisting of warmup. The feature dimensionalities considered are 1-dimensional and 25-dimensional, and training is done with a known predictive variance similar to the LR setup. For the probabilistic model, we consider both a 1-layered and a 2-layered multi-layer perceptron (MLP) network with 32 hidden units in each, and either a RELU or TANH activation function.

**Linear Classification (LC):** We experiment with 2-dimensional and 100-dimensional setups with training done for $50,000$ iterations, out of which $12,500$ are used for warmup. Further, we train for both binary classification as well as a 5-class classification setup.

**Nonlinear Classification (NLC):** We experiment with 2-dimensional and 25-dimensional setups with training done for $100,000$ iterations, out of which $2,5000$ are used for warmup. Further, we train for both binary classification as well as a 5-class classification setup. For the probabilistic model, we consider both a 1-layered and a 2-layered multi-layer perceptron (MLP) network with 32 hidden units in each, and either a RELU or TANH activation function.

## F.2 Variable-Dim

In this section, we provide the experimental details relevant to reproducing the results of Section 4. All the models are trained with streaming data from the underlying probabilistic model, such that every iteration of training sees a new set of datasets. Training is done with a batch size of 128, representing the number of datasets seen during one optimization step. Further, we ensure that the datasets sampled resemble a

| Objective | $q_\varphi$ | Model | $L_2$ Loss ($\downarrow$) | | | | | | $\leftarrow \chi_{real}$ |
|---|---|---|---|---|---|---|---|---|---|
| | | | Linear Model \| GP Data | | | MLP-TanH Model \| GP Data | | | $\leftarrow \chi_{sim}$ |
| | | | LR | NLR | GP | LR | NLR | GP | |
| Baseline | - | Random | - | - | $2.681_{\pm 0.089}$ | - | - | $16.236_{\pm 0.381}$ | |
| | - | Optimization | - | - | $0.263_{\pm 0.000}$ | - | - | $0.007_{\pm 0.000}$ | |
| | - | Langevin | - | - | $0.266_{\pm 0.001}$ | - | - | $0.022_{\pm 0.001}$ | |
| | - | HMC | - | - | $0.266_{\pm 0.000}$ | - | - | $0.090_{\pm 0.002}$ | |
| Fwd-KL | Gaussian | GRU | $0.268_{\pm 0.000}$ | - | - | - | $14.077_{\pm 0.368}$ | - | |
| | | DeepSets | $0.269_{\pm 0.001}$ | - | - | - | $14.756_{\pm 0.280}$ | - | |
| | | Transformer | $0.270_{\pm 0.001}$ | - | - | - | $14.733_{\pm 0.513}$ | - | |
| Rev-KL | | GRU | $0.268_{\pm 0.000}$ | $0.269_{\pm 0.000}$ | $0.266_{\pm 0.000}$ | $0.334_{\pm 0.005}$ | $0.157_{\pm 0.003}$ | $0.080_{\pm 0.003}$ | |
| | | DeepSets | $0.269_{\pm 0.000}$ | $0.269_{\pm 0.000}$ | $0.265_{\pm 0.000}$ | $0.331_{\pm 0.003}$ | $0.146_{\pm 0.002}$ | $0.063_{\pm 0.000}$ | |
| | | Transformer | $0.269_{\pm 0.000}$ | $0.269_{\pm 0.000}$ | $0.267_{\pm 0.000}$ | $0.310_{\pm 0.013}$ | $0.155_{\pm 0.006}$ | $0.066_{\pm 0.004}$ | |
| Fwd-KL | Flow | GRU | $0.268_{\pm 0.000}$ | - | - | - | $9.756_{\pm 0.192}$ | - | |
| | | DeepSets | $0.269_{\pm 0.001}$ | - | - | - | $14.345_{\pm 0.628}$ | - | |
| | | Transformer | $0.269_{\pm 0.000}$ | - | - | - | $8.557_{\pm 0.561}$ | - | |
| Rev-KL | | GRU | $0.268_{\pm 0.000}$ | $0.270_{\pm 0.001}$ | $0.266_{\pm 0.000}$ | $0.289_{\pm 0.011}$ | $0.120_{\pm 0.004}$ | $0.059_{\pm 0.003}$ | |
| | | DeepSets | $0.269_{\pm 0.000}$ | $0.269_{\pm 0.001}$ | $0.266_{\pm 0.000}$ | $0.270_{\pm 0.008}$ | $0.115_{\pm 0.002}$ | $0.059_{\pm 0.002}$ | |
| | | Transformer | $0.269_{\pm 0.001}$ | $0.270_{\pm 0.000}$ | $0.267_{\pm 0.000}$ | $0.293_{\pm 0.008}$ | $0.120_{\pm 0.005}$ | $0.055_{\pm 0.002}$ | |

Table 24: **Model Misspecification**. Results for model misspecification under different training data $\chi_{sim}$, when evaluated under GP Data ($\chi_{real}$), with the underlying model as a linear and MLP-TanH model respectively.

uniform distribution over the feature dimensions, ranging from 1-dimensional to the maximal dimensional setup. Evaluations are done with 25 samples and we ensure that the test datasets used for each probabilistic model are the same across all the compared methods, i.e., baselines, forward KL, and reverse KL. We train the amortized inference model and the forward KL baselines for the following different probabilistic models:

**Mean of Gaussian (GM):** We train the amortization models over $50,000$ iterations using a linear warmup with $12,5000$ iterations over which the weight of the KL term in our proposed approach scales linearly from 0 to 1. We use an identity covariance matrix for the data-generating process, but it can be easily extended to the case of correlated or diagonal covariance-based Gaussian distributions. In this setup, we consider a maximum of 100 feature dimensions.

**Gaussian Mixture Model (GMM):** We train the mixture model setup for $500,000$ iterations with $125,000$ iterations of warmup. We set the maximal feature dimensions as 5 and experiment with 2 and 5 mixture components. While we do use an identity covariance matrix for the data-generating process, again, it can be easily extended to other cases.

**Linear Regression (LR):** The amortization models for this setup are trained for $100,000$ iterations with $25,000$ iterations of warmup. The maximal feature dimension considered for this task is 100-dimensional, and the predictive variance $\sigma^2$ is assumed to be known and set as 0.25.

**Nonlinear Regression (NLR):** We train the setup for $250,000$ iterations with $62,500$ iterations consisting of warmup. The maximal feature dimension considered is 100-dimensional, and training is done with a known predictive variance similar to the LR setup. For the probabilistic model, we consider both a 1-layered and a 2-layered multi-layer perceptron (MLP) network with 32 hidden units in each, and either a RELU or TANH activation function.

**Linear Classification (LC):** We experiment with a maximal 100-dimensional setup with training done for $100,000$ iterations, out of which $25,000$ are used for warmup. Further, we train for both binary classification as well as a 5-class classification setup.

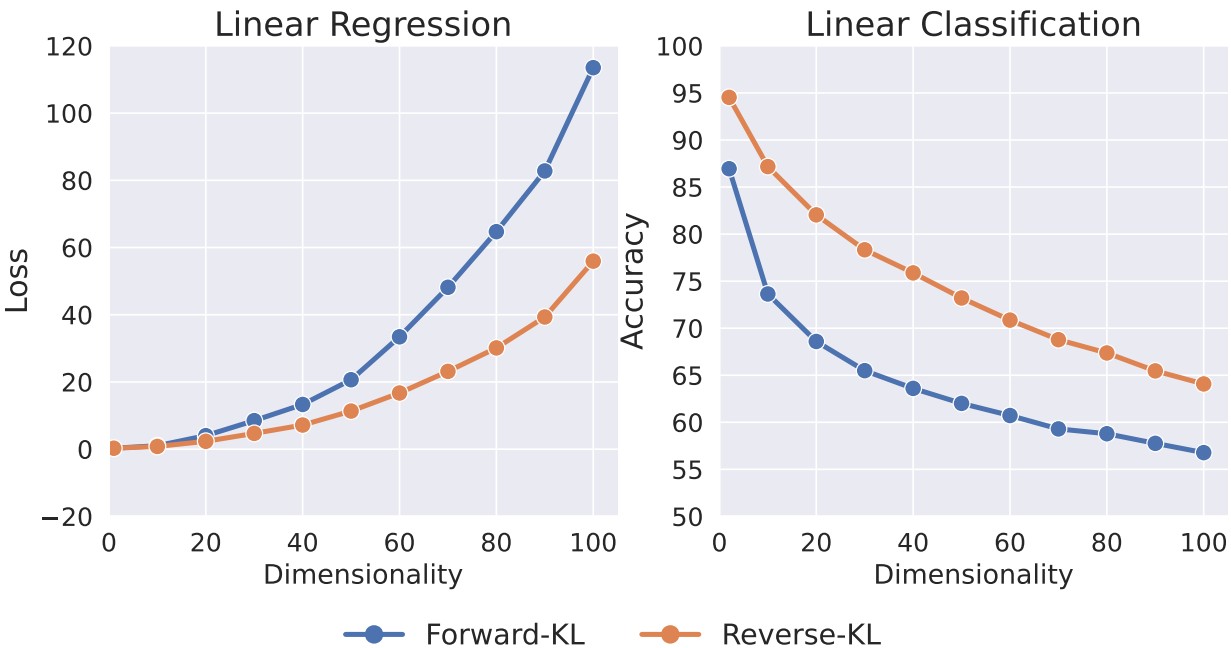

Figure 5: **Trends of Performance over different Dimensions in Variable Dimensionality Setup:** We see that our proposed reverse KL methodology outperforms the forward KL one.

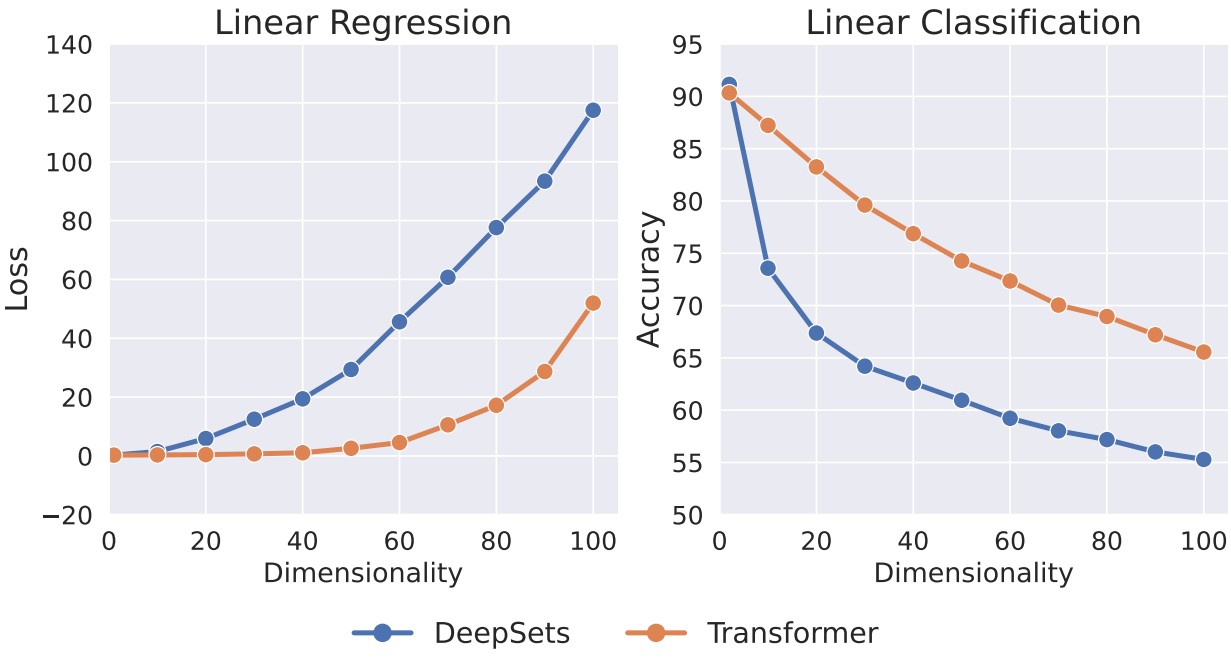

Figure 6: **Trends of Performance over different Dimensions in Variable Dimensionality Setup:** We see that transformer models generalize better to different dimensional inputs than DeepSets.

**Nonlinear Classification (NLC):** We experiment with a maximal 100-dimensional setup with training done for $250,000$ iterations, out of which $62,500$ are used for warmup. Further, we train for both binary classification as well as a 5-class classification setup. For the probabilistic model, we consider both a 1-layered and a 2-layered multi-layer perceptron (MLP) network with 32 hidden units in each, and either a RELU or TANH activation function.

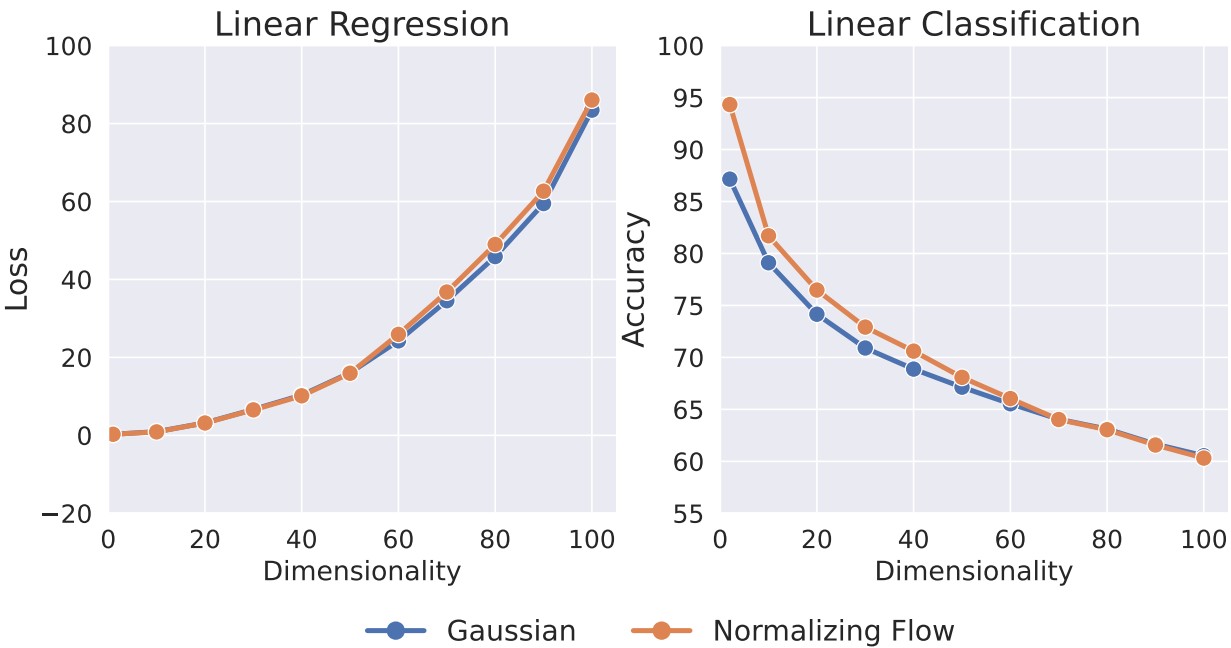

Figure 7: **Trends of Performance over different Dimensions in Variable Dimensionality Setup:** We see that normalizing flows leads to similar performances than Gaussian based variational approximation.

### F.3 Model Misspecification

In this section, we provide the experimental details relevant to reproducing the results of Section 4. All models during this experiment are trained with streaming data from the currently used dataset-generating function $\chi$, such that every iteration of training sees a new batch of datasets. Training is done with a batch size of 128, representing the number of datasets seen during one optimization step. Evaluation for all models is done with 10 samples from each dataset-generator used in the respective experimental subsection and we ensure that the test datasets are the same across all compared methods, i.e., baselines, forward KL, and reverse KL.

**Linear Regression Model:** The linear regression amortization models are trained following the training setting for linear regression fixed dimensionality, that is, $50,000$ training iterations with $12,500$ iterations of warmup. The feature dimension considered for this task is 1-dimension. The model is trained separately on datasets from three different generators $\chi$: linear regression, nonlinear regression, and Gaussian processes, and evaluated after training on test datasets from all of them. For training with datasets from the linear regression probabilistic model, the predictive variance $\sigma^2$ is assumed to be known and set as 0.25. The same variance is used for generating datasets from the nonlinear regression dataset generator with 1 layer, 32 hidden units, and TANH activation function. Lastly, datasets from the Gaussian process-based generator are sampled similarly, using the GPytorch library Gardner et al. (2018), where datasets are sampled of varying cardinality, ranging from 64 to 128. We use a zero-mean Gaussian Process (GP) with a unit lengthscale radial-basis function (RBF) kernel serving as the covariance matrix. Further, we use a very small noise of $\sigma^2 = 1\mathrm{e}^{-6}$ in the likelihood term of the GP. Forward KL training in this experiment can only be done when the amortization model and the dataset-generating function are the same: when we train on datasets from the linear regression-based $\chi$. Table 23 provides a detailed overview of the results.

**Nonlinear Regression Models:** The nonlinear regression amortization models are trained following the training setting for nonlinear regression fixed dimensionality, that is, $100,000$ training iterations with $25,000$ iterations of warmup. Here, we consider two single-layer perceptions with 32 hidden units with a TANH activation function. The feature dimensionality considered is 1 dimension. We consider the same dataset-generating functions as in the misspecification experiment for a linear regression model above. However, the activation function used in the nonlinear regression dataset generator matches the activation function of

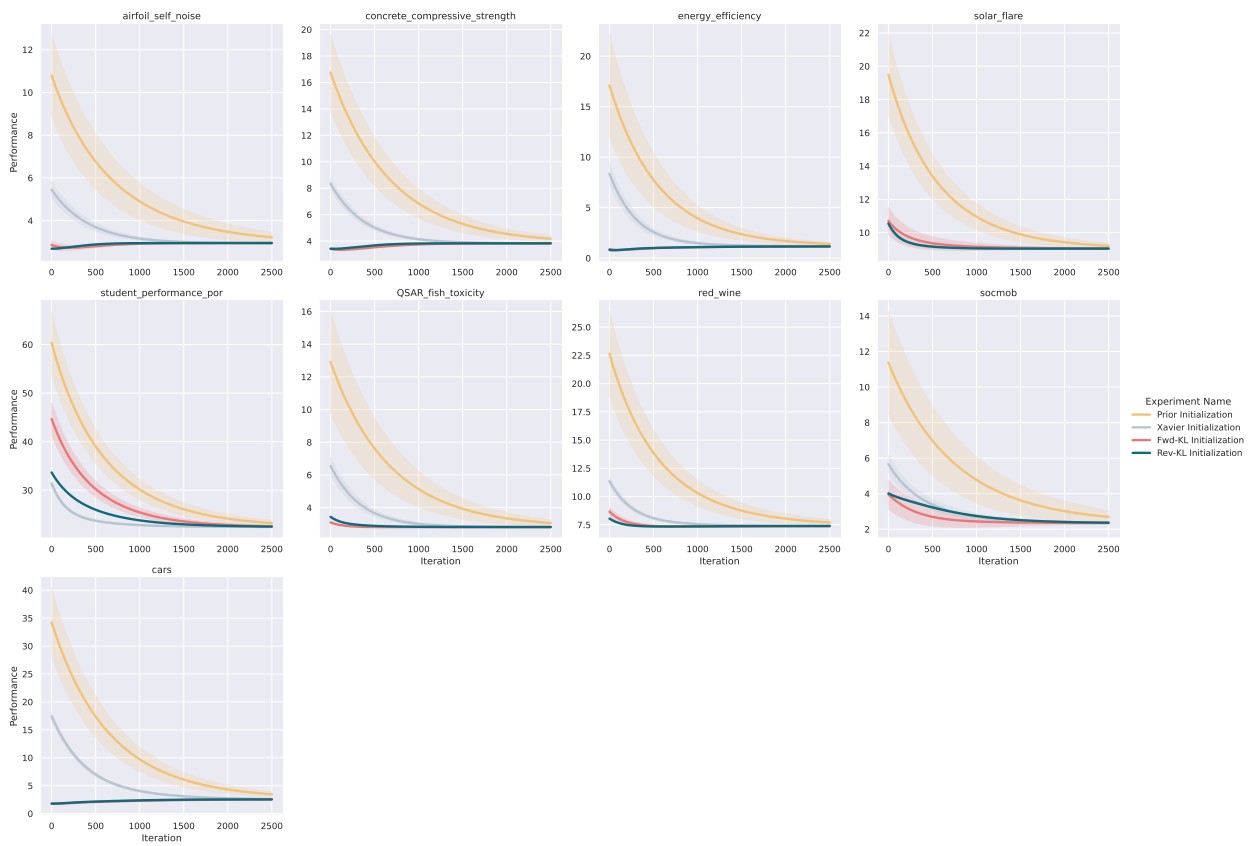

Figure 8: **Tabular Experiments | Linear Regression with Diagonal Gaussian**: For every regression dataset from the OpenML platform considered, we initialize the parameters of a linear regression-based probabilistic model with the amortized inference models which were trained with a diagonal Gaussian assumption. The parameters are then further trained with maximum-a-posteriori (MAP) estimate with gradient descent. Reverse and Forward KL denote initialization with the correspondingly trained amortized model. Prior refers to a MAP-based optimization baseline initialized from the prior $\mathcal{N}(0, I)$, whereas Xavier refers to initialization from the Xavier initialization scheme.

the currently trained amortization model. In this case, forward KL training is possible in the two instances when trained on datasets from the corresponding nonlinear regression probabilistic model. A more detailed overview of the results can be found in Table 23 and 24.

## F.4 Tabular Experiments

For the tabular experiments, we train the amortized inference models for (non-)linear regression (NLR/LR) as well as (non-)linear classification (NLC/LC) with $\boldsymbol{x} \sim \mathcal{N}(\mathbf{0}, \mathbf{I})$ as opposed to $\boldsymbol{x} \sim \mathcal{U}(-\mathbf{1}, \mathbf{1})$ in the dataset generating process $\chi$, with the rest of the settings the same as MAXIMUM-DIM experiments. For the nonlinear setups, we only consider the RELU case as it has seen predominant success in deep learning. Further, we only consider a 1-hidden layer neural network with 32 hidden dimensions in the probabilistic model.

After having trained the amortized inference models, both for forward and reverse KL setups, we evaluate them on real-world tabular datasets. We first collect a subset of tabular datasets from the OpenML platform as outlined in Appendix G. Then, for each dataset, we perform a 5-fold cross-validation evaluation where the dataset is chunked into 5 bins, of which, at any time, 4 are used for training and one for evaluation. This procedure is repeated five times so that every chunk is used for evaluation once.

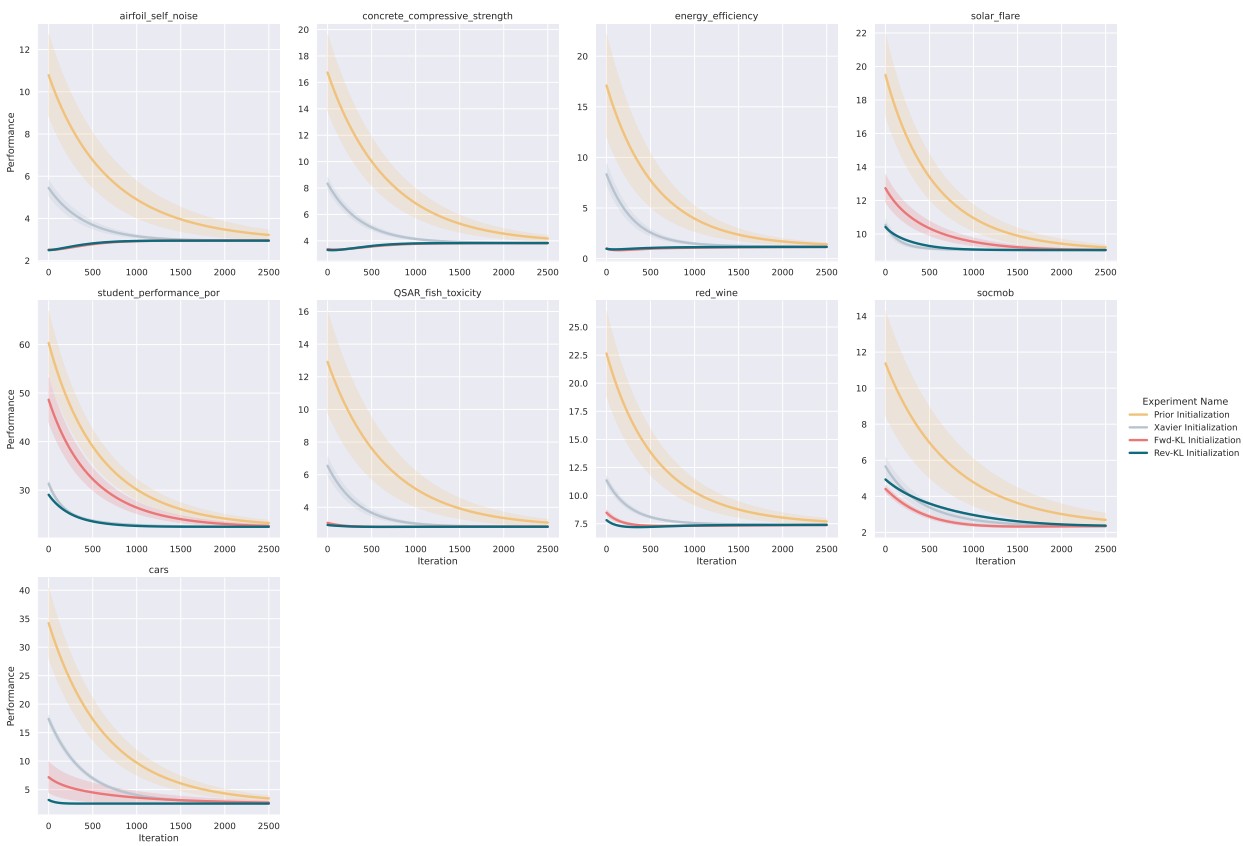

Figure 9: **Tabular Experiments | Linear Regression with Normalizing Flow**: For every regression dataset from the OpenML platform considered, we initialize the parameters of a linear regression-based probabilistic model with the amortized inference models which were trained with a normalizing flow-based model. The parameters are then further trained with maximum-a-posteriori (MAP) estimate with gradient descent. Reverse and Forward KL denote initialization with the correspondingly trained amortized model. Prior refers to a MAP-based optimization baseline initialized from the prior $\mathcal{N}(0, I)$, whereas Xavier refers to initialization from the Xavier initialization scheme.

For each dataset, we normalize the observations and the targets so that they have zero mean and unit standard deviation. For the classification setups, we only normalize the inputs as the targets are categorical. For both forward KL and reverse KL amortization models, we initialize the probabilistic model from samples from the amortized model and then further finetune it via dataset-specific maximum a posteriori optimization. We repeat this setup over 25 different samples from the inference model. In contrast, for the optimization baseline, we initialize the probabilistic models' parameters from $\mathcal{N}(0, I)$, which is the prior that we consider, and then train 25 such models with maximum a posteriori objective using Adam optimizer.

While we see that the amortization models, particularly the reverse KL model, lead to much better initialization and convergence, it is important to note that the benefits vanish if we initialize using the Xavier-init initialization scheme. However, we believe that this is not a fair comparison as it means that we are considering a different prior now, while the amortized models were trained with $\mathcal{N}(0, I)$ prior. We defer the readers to the section below for additional discussion and experimental results.

## G OpenML Datasets

For the tabular regression problems, we consider the suite of tasks outlined in *OpenML-CTR23 - A curated tabular regression benchmarking suite* (Fischer et al., 2023), from which we further filter out datasets that

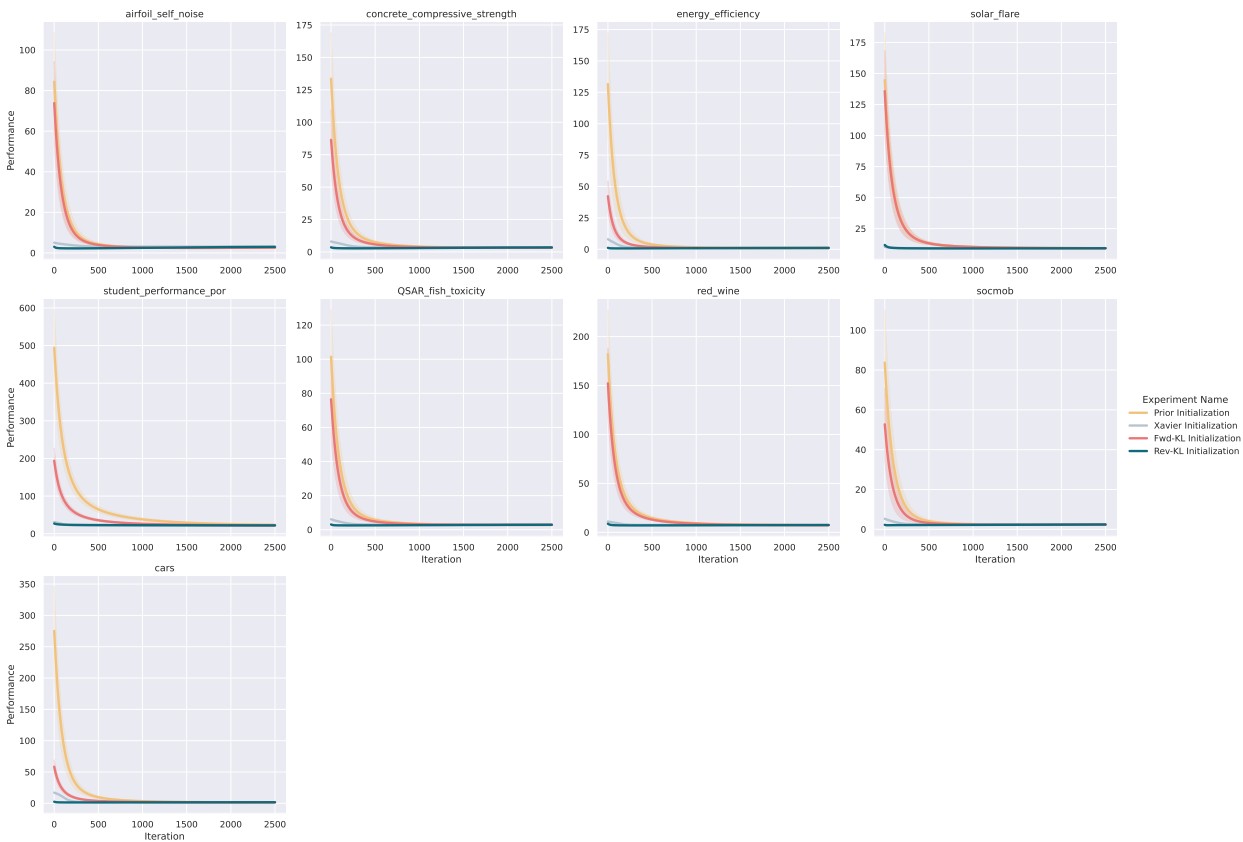

Figure 10: **Tabular Experiments | Nonlinear Regression with Diagonal Gaussian**: For every regression dataset from the OpenML platform considered, we initialize the parameters of a nonlinear regression-based probabilistic model with the amortized inference models which were trained with a diagonal Gaussian assumption. The parameters are then further trained with maximum-a-posteriori (MAP) estimate with gradient descent. Reverse and Forward KL denote initialization with the correspondingly trained amortized model. Prior refers to a MAP-based optimization baseline initialized from the prior $\mathcal{N}(0, I)$, whereas Xavier refers to initialization from the Xavier initialization scheme.

have more than 2000 examples and 100 features. We also remove datasets with missing information and NaNs. Similarly, we consider the *OpenML-CC18 Curated Classification benchmark* (Bischl et al., 2019) suite of tasks for classification and perform a similar filtering algorithm. We remove datasets with missing information and NaNs, as well as datasets with more than 2000 examples and 100 features. In addition, we also exclude datasets that are not made for binary classification. At the end of this filtering mechanism, we end up with 9 regression and 13 classification problems, and our dataset filtration pipeline is heavily inspired by Hollmann et al. (2022). We provide the datasets considered for both regression and classification below:

**Regression**: AIRFOIL_SELF_NOISE, CONCRETE_COMPRESSIVE_STRENGTH, ENERGY_EFFICIENCY, SOLAR_FLARE, STUDENT_PERFORMANCE_POR, QSAR_FISH_TOXICITY, RED_WINE, SOCMOB and CARS.

**Classification**: CREDIT-G, DIABETES, TIC-TAC-TOE, PC4, PC3, KC2, PC1, BANKNOTE-AUTHENTICATION, BLOOD-TRANSFUSION-SERVICE-CENTER, ILPD, QSAR-BIODEG, WDBC and CLIMATE-MODEL-SIMULATION-CRASHES.

# H   Additional Experiments

In this section, we outline the additional experiments we conducted in obtaining Bayesian posteriors for the different probabilistic models for different hyperparameters and their downstream uses. We provide a comprehensive account of the results in the relevant sections below.

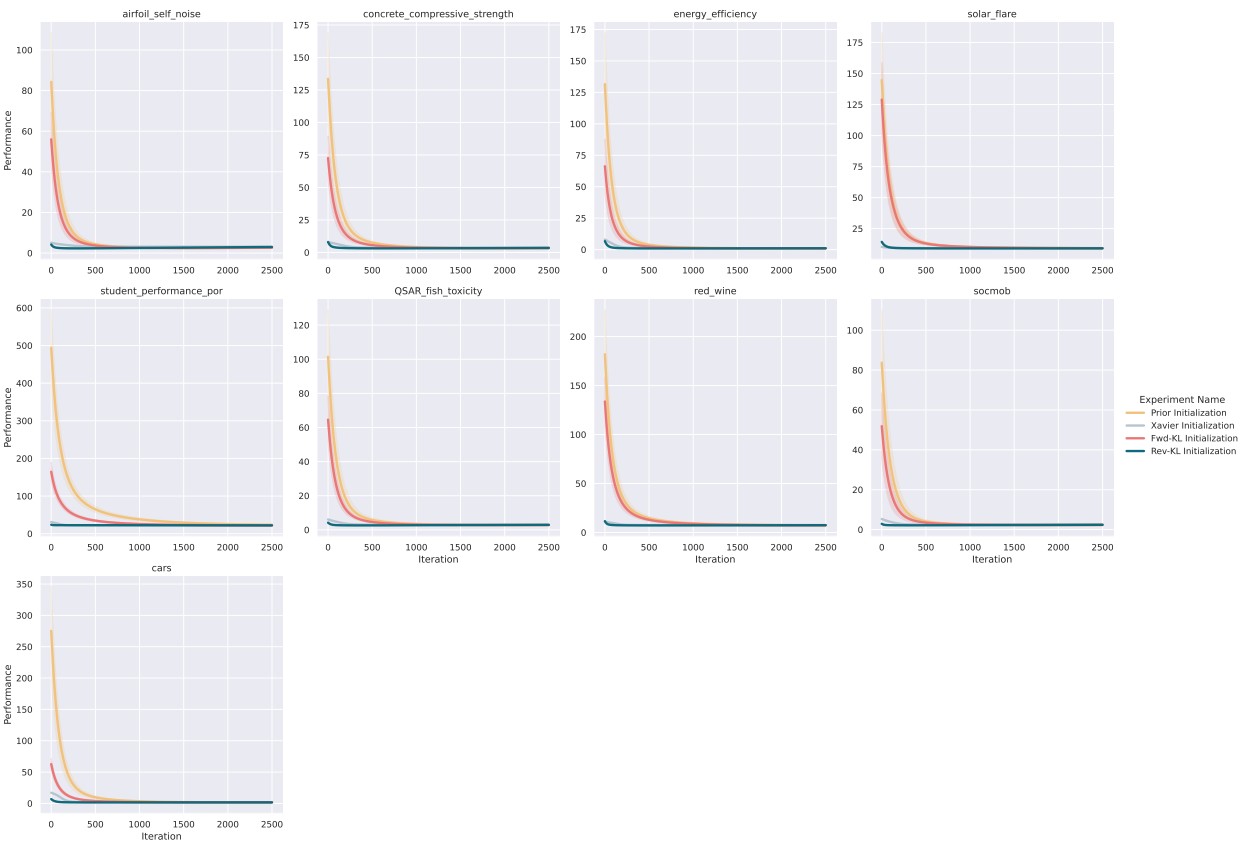

Figure 11: **Tabular Experiments | Nonlinear Regression with Normalizing Flow**: For every regression dataset from the OpenML platform considered, we initialize the parameters of a nonlinear regression-based probabilistic model with the amortized inference models which were trained with a normalizing flow-based model. The parameters are then further trained with maximum-a-posteriori (MAP) estimate with gradient descent. Reverse and Forward KL denote initialization with the correspondingly trained amortized model. Prior refers to a MAP-based optimization baseline initialized from the prior $\mathcal{N}(0, I)$, whereas Xavier refers to initialization from the Xavier initialization scheme.

### H.1    Fixed-Dim

While we highlighted the results with the Gaussian mixture model and classification settings with only 2 clusters/classes, we also conducted experiments with an increased number of clusters and classes, making the problem even more challenging. Tables 7-14 shows that both forward and reverse KL methods perform reasonably, with forward KL struggling more in challenging scenarios.

Next, we also consider harder tasks based on the Bayesian Neural Network (BNN) paradigm, where we consider nonlinear regression and classification setups with different activation functions: TANH and RELU for a 1-layered and 2-layered BNN. We provide the results of our experiments in Tables 7-14. The results indicate that forward KL approaches struggle a lot in such scenarios, often achieving performance comparable to random chance. On the contrary, we see that reverse KL-based amortization leads to performances often similar to dataset-specific optimization, thereby showing the superiority of our proposed method.

### H.2    Variable-Dim

Our experiments on variable dimensional datasets can be evaluated for arbitrary feature cardinality, of which we show a few examples in Section 4. In this section, we provide results for additional dimensionality setups. In particular, we refer the readers to Tables 15-22, which contain experimental results w.r.t different dimensionalities (e.g. 50D setup), as well as different number of clusters and classes, respectively, for the GMM

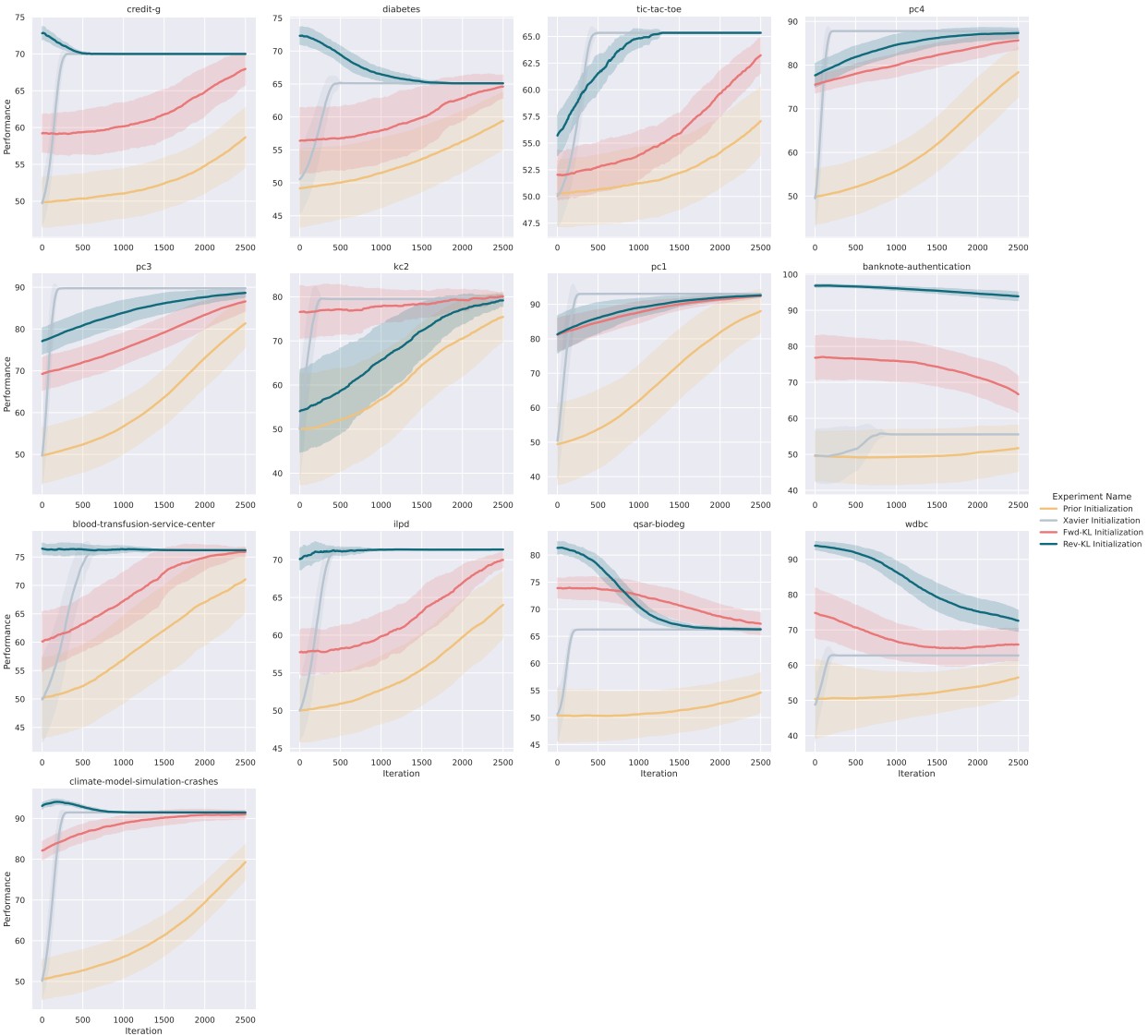

Figure 12: **Tabular Experiments | Linear Classification with Diagonal Gaussian**: For every classification dataset from the OpenML platform considered, we initialize the parameters of a linear classification-based probabilistic model with the amortized inference models which were trained with a diagonal Gaussian assumption. The parameters are then further trained with maximum-a-posteriori (MAP) estimate with gradient descent. Reverse and Forward KL denote initialization with the correspondingly trained amortized model. Prior refers to a MAP-based optimization baseline initialized from the prior $\mathcal{N}(0, I)$, whereas Xavier refers to initialization from the Xavier initialization scheme.

and LC setup. Throughout, we see that amortization leads to reasonable performance, and in particular, we see forward KL-based amortization starting to struggle in high-dimensional setups.

Again, to make the setup more challenging, we consider the Bayesian Neural Network (BNN) setup where we consider nonlinear regression and classification with different activation functions: TANH and RELU for a 1-layered and 2-layered BNN, but which can now be tested for an arbitrary number of input features. Our experiments are highlighted in Tables 15-22, for 1- and 2-layered BNN, among others. In such complex multi-modal and complicated setups, forward KL often performs comparable to random chance and thus does not lead to any good approximation of the true posterior distribution. On the other hand, our proposed method indeed leads to good predictive performance, often comparable to dataset-specific optimization routines.

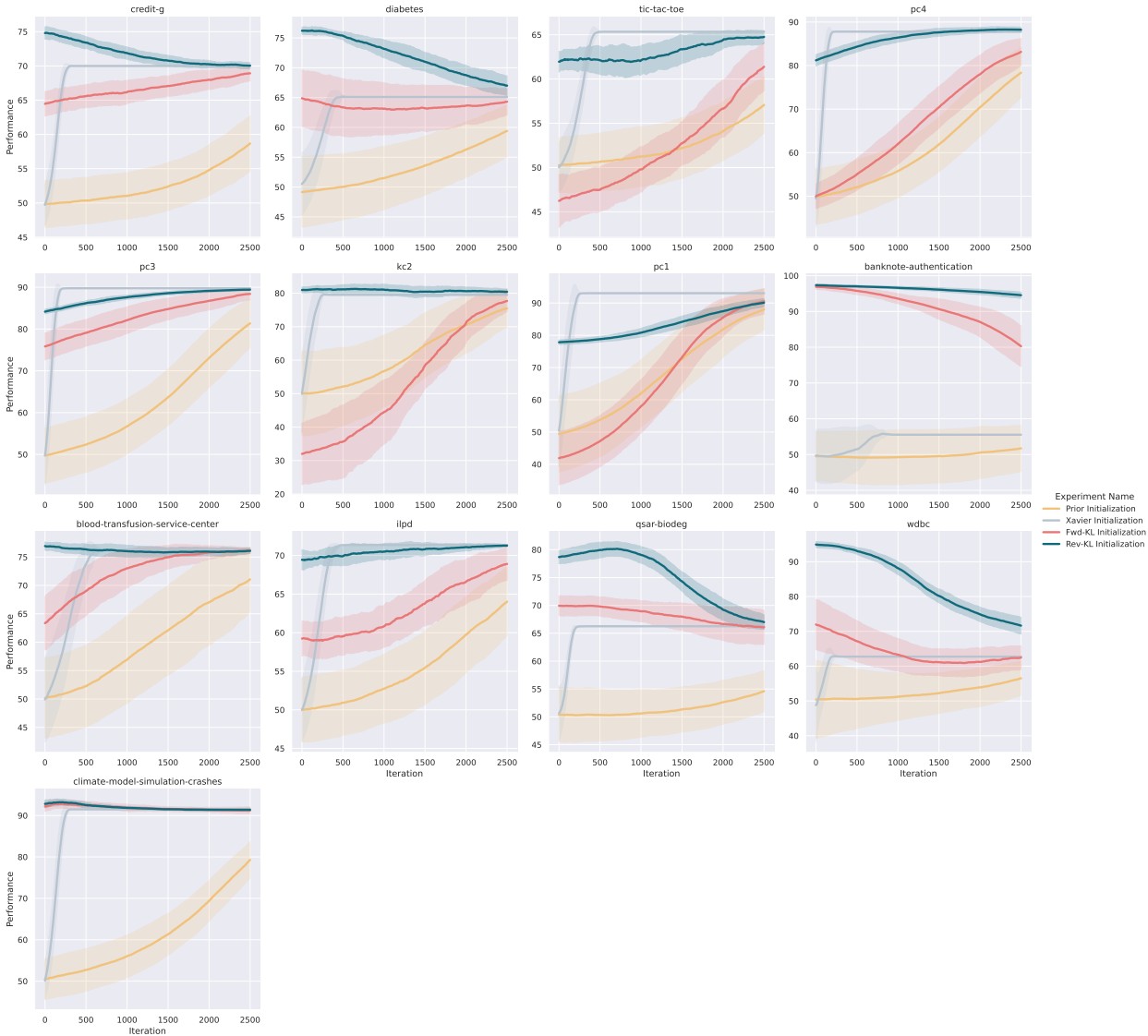

Figure 13: **Tabular Experiments | Linear Classification with Normalizing Flow**: For every classification dataset from the OpenML platform considered, we initialize the parameters of a linear classification-based probabilistic model with the amortized inference models which were trained with a normalizing flow-based model. The parameters are then further trained with maximum-a-posteriori (MAP) estimate with gradient descent. Reverse and Forward KL denote initialization with the correspondingly trained amortized model. Prior refers to a MAP-based optimization baseline initialized from the prior $\mathcal{N}(0, I)$, whereas Xavier refers to initialization from the Xavier initialization scheme.

## H.3 Model Misspecification

As a representative of the results on model misspecification (Section 4), we highlighted training and evaluation of the amortization models with Transformer backbone on a subset of in-distribution and OoD data-generating functions (Table 3) to show superiority in generalization of reverse KL trained system vs. forward KL based ones on OoD data but also to highlight that training a misspecified amortization model on OoD datasets directly with our approach results in even better posterior predictive performance.

In addition to those experiments, we also conducted a broader range of experiments utilizing DeepSets as the backbone, various OoD data-generating functions for training and evaluation of the reverse KL system, and an additional nonlinear regression model with RELU activation function. For a comprehensive description

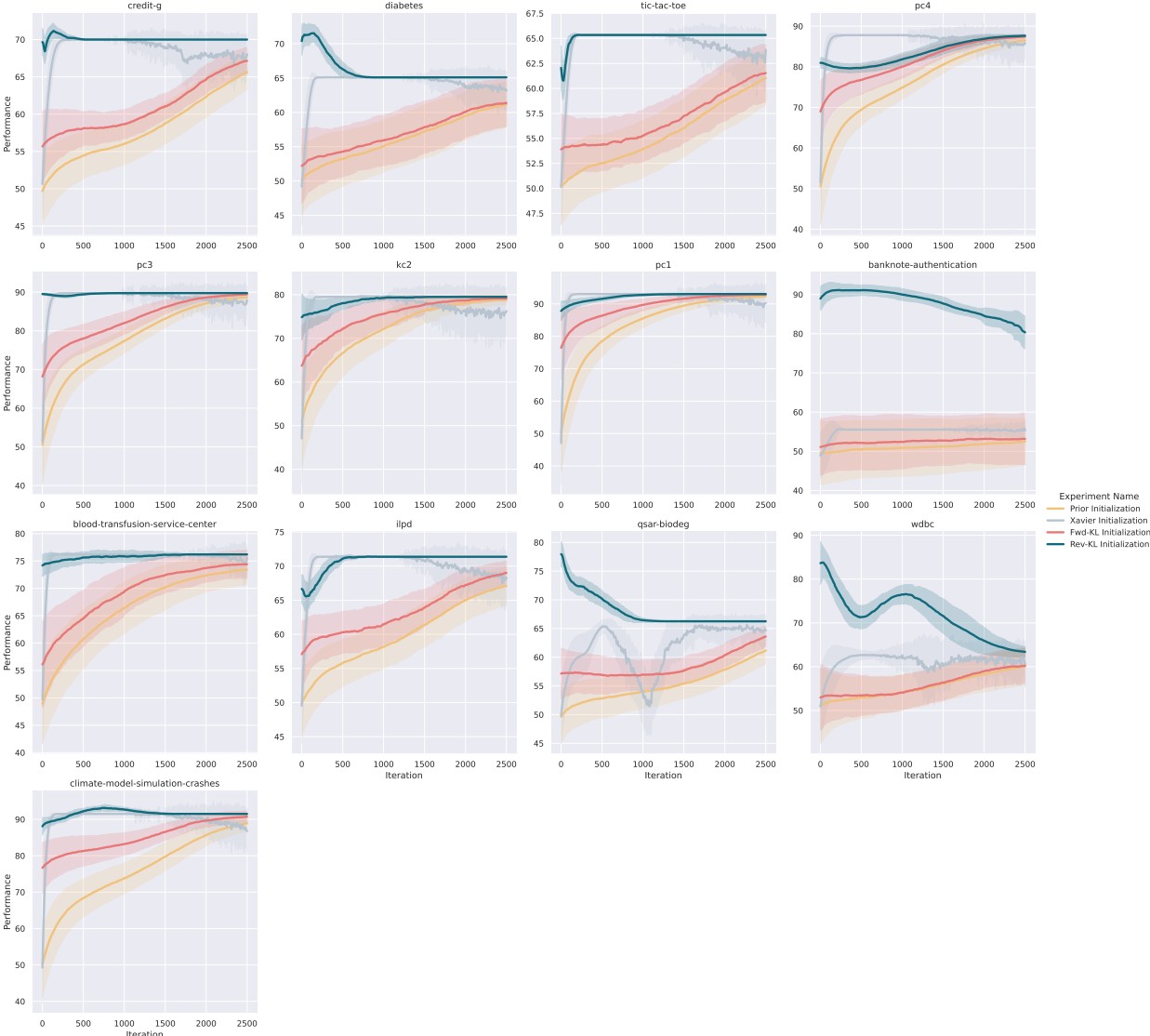

Figure 14: **Tabular Experiments | Nonlinear Classification with Diagonal Gaussian**: For every classification dataset from the OpenML platform considered, we initialize the parameters of a nonlinear classification-based probabilistic model with the amortized inference models which were trained with a diagonal Gaussian assumption. The parameters are then further trained with maximum-a-posteriori (MAP) estimate with gradient descent. Reverse and Forward KL denote initialization with the correspondingly trained amortized model. Prior refers to a MAP-based optimization baseline initialized from the prior $\mathcal{N}(0, I)$, whereas Xavier refers to initialization from the Xavier initialization scheme.

of these experiments and the complete setup, please refer to Section F.3. We considered two probabilistic models, including a linear regression model and a nonlinear regression models utilizing the TANH activation function. The detailed results for each model can be found in Tables 23 and 24.

In all experiments, reverse KL outperforms forward KL trained amortization models in in-distribution performance and excels in posterior prediction on OoD datasets. Although the significant difference in posterior prediction performance of forward vs. reverse KL in cases where the underlying model is nonlinear was already mentioned in previous experiments, here, reverse KL-trained models also excel in evaluations of posterior prediction for the linear regression model. Although only by a margin, in the case of approximating the posterior of the simpler linear regression model, a diagonal Gaussian-shaped posterior shows the best posterior prediction results when evaluated on OoD datasets from the nonlinear regression dataset generating

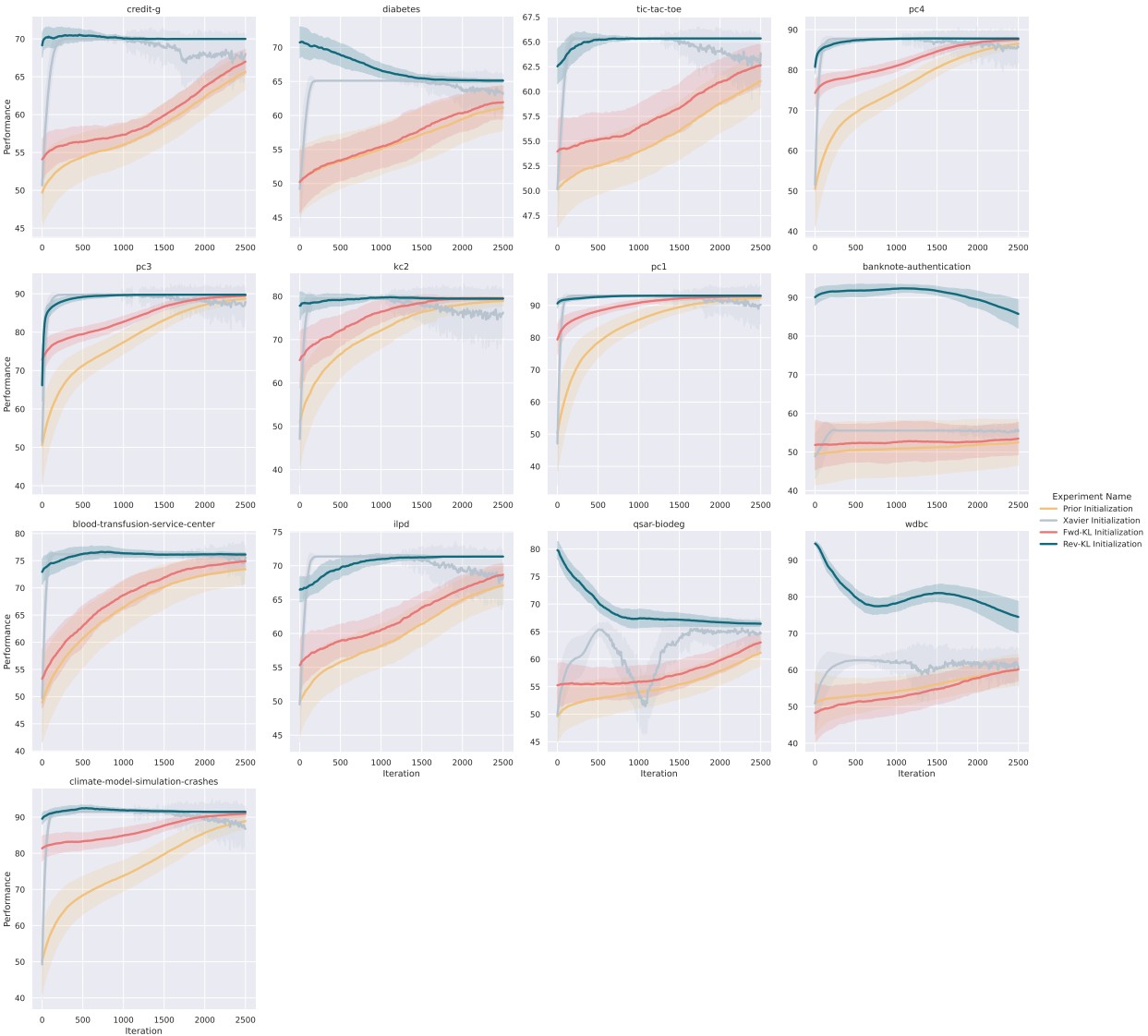

Figure 15: **Tabular Experiments | Nonlinear Classification with Normalizing Flow**: For every classification dataset from the OpenML platform considered, we initialize the parameters of a linear classification-based probabilistic model with the amortized inference models which were trained with a normalizing flow-based model. The parameters are then further trained with maximum-a-posteriori (MAP) estimate with gradient descent. Reverse and Forward KL denote initialization with the correspondingly trained amortized model. Prior refers to a MAP-based optimization baseline initialized from the prior $\mathcal{N}(0, I)$, whereas Xavier refers to initialization from the Xavier initialization scheme.

function. In almost all other experiments, the posterior prediction performance could be enhanced when we used the normalizing flow based posterior. A definitive conclusion cannot be drawn regarding the superiority of one backbone over the other, i.e. between DeepSets or Transformer. However, amortization models with DeepSets as the backbone tend towards better generalization regarding OoD datasets.

## H.4 Tabular Experiments

As a case of extreme OoD generalization, we test our amortized models trained to handle variable feature dimensions on the suite of regression and classification problems that we filtered out from the OpenML platform, as outlined in Appendix G. We consider both linear and nonlinear probabilistic models to tackle the regression

and binary classification setups, which lead to predicting the parameters of a linear regression/classification model and a small nonlinear neural network based on RELU activation function. Further, we also perform the analysis with a diagonal Gaussian assumption and a normalizing flow-based amortization model trained with both a forward and reverse KL objective. We provide the results on the regression problems in (a) linear model with diagonal Gaussian assumption (Figure 8), (b) linear model with normalizing flow (Figure 9), (c) nonlinear model with diagonal Gaussian assumption (Figure 10), and (d) nonlinear model with normalizing flow (Figure 11). The results of the classification problems are shown in (a) linear model with diagonal Gaussian assumption (Figure 12), (b) linear model with normalizing flow (Figure 13), (c) nonlinear model with diagonal Gaussian assumption (Figure 14), and (d) nonlinear model with normalizing flow (Figure 15). Our experiments indicate that initializing with amortized models leads to better performance and training than models trained via maximum a-posteriori approach and initialized with the prior, i.e., $\mathcal{N}(0, I)$.

We do provide an additional baseline of initializing with XAVIER-INIT initialization, which often leads to faster convergence; however, as we consider the prior to be a unit normal, this is an unfair baseline as we assume the weights to be initialized from a different prior. We leave the work of computing Bayesian posteriors with different priors and testing an amortized Bayesian model with XAVIER-INIT prior for the future.

