# OpenReview forum: "Amortizing Bayesian Posterior Inference in Tractable Likelihood Models"
_TMLR — Rejected by TMLR_

### Review · Reviewer_Mnwc · 2024-07-03

**Summary Of Contributions:**

This paper introduced a novel approach to alleviate the computational burden associated with Bayesian inference by amortizing posterior parameter inference. The main contributions can be summarized as follows:

(1) Amortized Bayesian Posterior Inference Framework: this paper proposed a framework to use neural networks to map datasets directly to posterior distributions, possessing the ability to generalize to new datasets without training.

(2) Reverse-KL based Variational Inference: this paper employed the reverse-KL divergence to measure the gap between the approximate and the true posterior in the variational inference problem. Compared to the Simulation-Based Inference (SBI) which uses the forward-KL divergence to measure the gap, the reverse-KL based approach can efficiently handle model misspecification.

(3) Extensive Benchmarking: experiments across various probabilistic models and real-world scenarios are conducted to showcase the effectiveness of the proposed approach in variable feature dimensions settings and model misspecification settings.

**Audience:**

Yes

**Broader Impact Concerns:**

The proposed approach for amortizing Bayesian posterior inference has significant potential to improve the efficiency and scalability of Bayesian analysis in various scientific and practical applications. I didn't see any negative impact of the paper.

**Claims And Evidence:**

Yes

**Requested Changes:**

(1) In the section 'Handling Model Misspecification', some experimental results are presented for the proposed methods. For the same $\chi_{real}$, if we use different $\chi_{sim}$ (not the same as $\chi_{real}$), how does the results look like? I think experiments like this will enhance our understanding of the robustness to model misspecification.

(2) Related to approximating multi-modal posteriors, it is mentioned that forward-KL captures multi-modality better while reverse-KL gives better approximation. Is this true for general multi-modal distributions (not necessary GMMs) with lots of modes? I believe a more though explanation would be helpful.

(3) Typo: page 3 paragraph 'Amortization', $q_{\varphi}(z_i|x_i):=\mathcal{N}(\cdot; \mu_{\varphi}(x_i),\Sigma(x_i))$ to $q_{\varphi}(z_i|x_i):=\mathcal{N}(\cdot; \mu_{\varphi}(x_i),\Sigma_{\varphi}(x_i))$

**Strengths And Weaknesses:**

Strengths

(1) The proposed framework of amortizing Bayesian posterior inference is computationally efficient in various scenarios, compared to baselines including MLE via gradient-based optimization, Langevin-based MCMC sampling and SBI-based forward-KL baseline.

(2) The proposed approach is highly adaptive due to its robustness to model misspecification and its ability to generalize to variable feature dimensions.

(3) Extensive experiments are conducted to provide strong evidence of the proposed method's improved performance compared to multiple baselines.

Weaknesses

(1) The proposed method that is based on reverse-KL is getting worse with increased dimensions. The sensitivity of the proposed method to the dimensionality is only compared to SBI based on forward-KL (in Figure 5). It is not clear how does it compared to other baselines, such as Langevin-based MCMC sampling.

(2) The proposed method seems to be outperformed by the optimization and MCMC baselines in estimating the means of GMMs. In particular, the GMM experiments in the paper are mostly a small number of clusters (2 and 5). It leaves doubts on the ability of the proposed methods on approximating multi-modal posteriors with a large number of modes.

---

> ### Author Response · Authors · 2024-07-18
> **Author Response**
>
> We thank the reviewer for their useful feedback and comments, and hope that our response addresses the concerns raised.
>
> **Sensitivity to Dimensionality**: We agree that the proposed method gets worse with increasing dimensions but this is expected as the dimensionality increases, the difficulty of the problem increases as well, which would hold for the Langevin-based MCMC approach as well. For example, Table 2 highlights the performance of all the approaches on the two extremes (2-dimensional and 100-dimensional) setups, where we can see a comparison with Langevin-based MCMC and see a significant drop in performance, as well as better performance of Transformer based reverse-KL setup.
>
> **Outperformance by Optimization and MCMC in GMMs**: While optimization based approaches do lead to better performance than the proposed approach, we do see that the proposed approach does significantly better than MCMC baselines in Tables 1 and 2. While we agree that the GMM tests are done with a fairly few number of clusters, we can still see that for the nonlinear regression and classification tasks the posterior is still highly multimodal owing to permutation symmetries in the weight matrix. In this setup, we still see that the proposed approach does considerably well.
>
> **Model Misspecification**: We refer the reviewer to Tables 14-16 that present these results. For example, in Table 14 we look at a linear model where the first row indicates $\chi_{sim}$ and the second row indicates $\chi_{real}$. So if we look at the NLR columns depicted by the second row, it would give us a comparative performance on $\chi_{real} = NLR$ for different $\chi_{sim} = \{LR, NLR, GP\}$. We have updated the draft to make it more clearer in the tables in the Appendix, and we generally see that the performance is best when $\chi_{real} = \chi_{sim}$ as expected.
>
> **Forward KL and Multimodality**: It is indeed quite true that the Forward KL approach tries to capture multiple modes as opposed to prioritizing only a few modes. This can be seen through the optimization objective, and is also mentioned in the Chapter 10 of [1]. From the optimization objective, we can see that the Forward KL objective is
>
> $$\int p(x) \log \frac{p(x)}{q(x)}$$
>
> Which implies that if the approximate density $q(x)$ puts a low density $\epsilon \to 0$ in a certain volume where the true density $p(x) \neq 0$, then the above term would end up tending to $\infty$. Thus, the optimization requires that volumes with non-negligible density need to be covered under the approximate density $q(x)$. On the other hand, through a similar walkthrough one can see that in Reverse KL approach, $\int q(x) \log \frac{q(x)}{p(x)}$, all one requires is that if $p(x)$ has negligible density over some volume, then $q(x)$ should also have negligible density there. However, the necessity to cover all regions where $p(x)$ has a density with $q(x)$ is not enforced that firmly because $q(x)$ can have zero density in areas where $p(x)$ is nonzero since the integrand would not contribute to the optimization objective in this case. Thus, the reverse KL approach can lead to capturing fewer modes as long as low density regions are not provided sufficient density under the approximation.
>
> **Typo**: We thank the reviewer for spotting this typo and have made the corresponding changes.
>
> [1] Bishop, Christopher M., and Nasser M. Nasrabadi. Pattern recognition and machine learning. Vol. 4. No. 4. New York: springer, 2006.

---

> > ### Author Response · Authors · 2024-10-15
> > **Request for Discussion**
> >
> > We thank the reviewer for their time and valuable feedback; and would like to ask if our rebuttal addresses their concerns? We would be happy to provide further clarifications if the reviewer has more questions.

---

### Review · Reviewer_oVsr · 2024-08-15

**Summary Of Contributions:**

Authors empirically compare several methods for learning amortized Bayesian posteriors.
They focus on contrasting the forward- and reverse-KL objective, and comparing two permutation-invariant neural network architectures - DeepSets and Transformers.

**Audience:**

Yes

**Claims And Evidence:**

No

**Requested Changes:**

See Weaknesses.

**Strengths And Weaknesses:**

Disclosure: I have reviewed an earlier version of this manuscript for a workshop venue before. In the current version, the writing has been improved and some pieces of my previous feedback have been incorporated, while some haven’t.

Strengths:
S1. I find the paper quite well-written and the content is well presented. The appendices seem thorough but I haven’t reviewed them carefully.
S2. The evaluation of various permutation-invariant architectures for the purpose of amortized inference is a worthwhile research contribution.


Weaknesses:
W1. Confidence intervals.
Given the magnitude of differences between different methods, some assessment of the variance of the results over multiple runs is needed. Then, in the results highlight all of the rows for which the confidence intervals overlap with the best value.

W2. Metrics.
In my opinion, the metrics used currently in Tables 1-4 (L2-loss and accuracy) do not comprehensively assess the quality of the approximate posterior yielded by the method, and as such are not sufficient to understand the performance of approximate Bayesian inference achieved (which is the goal of the procedure).

W2a. If I understand the metrics used correctly (please correct me if I’m wrong), a VI algorithm which uses a unimodal Gaussian distribution, estimates the mean well, but underestimates its variance w.r.t. the true posterior, would achieve a better performance than the true posterior through the lens of the L2-loss metric - am I right?

W2b. The goal of (approximate) Bayesian inference is posterior distribution estimation (and the ground truth posterior depends on the prior), not point-estimate estimation which is assessed by currently used metrics.
In my opinion, to assess the quality of approximate posterior q(\theta | D), we should compare q(.) to the true posterior p(\theta | D), e.g. using KL[p(\theta | D) || q(\theta | D)], Jensen-Shannon divergence (JSD), or a Classifier Two Sample Test, for more discussion see Section 2.2 of https://arxiv.org/abs/2101.04653.
Authors use JSD for Experiments in Sec 5.1, Table 5, and there the difference in the results between the forward-KL and reverse-KL seems somewhat marginal (see W1).

W2c. In my opinion, authors can evaluate the JSD for most of the problems they consider by running MCMC “for a sufficiently long time” (which is definitely very doable for problems in Tables 1-4) and treating the KDE of those samples as a proxy for the ground truth posterior.

W3. Mode-seeking vs mode-averaging behavior (and under- vs overestimating variance).
I feel like this aspect is undernarrated/played-down by the authors, while crucial for Bayesian inference (particularly for multimodal posteriors).
For example, if one of the modes of the true posterior is missed by the approximate posterior (or in fact, even if the variance is underestimated/too light-tailed), importance sampling estimators using such approximate posterior as a proposal may have infinite variance [Art Owen, 2013, Example 9.1].
In those cases, the overdispersion of the  approximate posterior due to the mode-averaging behavior of the forward-KL objective might be a feature.

Overall, currently the narrative of the paper seems to me to suggest the takeaway of “the reverse-KL objective works better than the forward-KL”, while that’s simply not true - it might indeed be easier to learn in some settings and doesn’t require “observed \theta” in the data, but it also behaves in a way that is not desired in some usecases.
I would like authors to narrate that appropriately/with more nuance.

W4. Please include some simple non-permutation-invariant NN architecture baseline (e.g. an RNN) to illustrate how much better the permutation-invariant architectures are.

W5. “[SBI] has limited applicability in more general estimation problems like the distribution over weights of a Bayesian Neural Networks.” - I think it’s a misnomer to call every case of amortizing the posterior using forward-KL “SBI” - some approaches, e.g. [Paige & Wood, 2016], assume known likelihood and still use forward-KL for specific reasons.


Comments:
C1. Typo: “does worsen with increasing dimensionality”.
C2. Regarding section “Handling Model Misspecification”.
C2a. What is the quantity displayed in Table 3?
C2b. If the probabilistic model is misspecified w.r.t. the true data-generating-process, what is the desired behavior of approximate Bayesian inference? Should it still yield an approximate posterior for a given dataset D as faithful as possible to the “true posterior under that misspecified model”, or something else? How does that relate to the answer for the question in C2a?

---

> ### Author Response · Authors · 2024-10-07
> **Author Response**
>
> We thank the reviewer for their useful feedback and comments, and hope that our response addresses the concerns raised.
>
> **W1**: We thank the reviewer for bringing this point up. We omit the standard deviation in the table just to make them more readable but following the reviewer’s concerns, we highlight the standard deviation of our results in Table 6 (corresponding to Table 1) as an example. Is this more in line with what the reviewer wanted? We will be making similar edits for the other tables as well.
>
> **W2**: We would actually like to push back on this point. We definitely agree with the reviewer that loss and accuracy metrics aren’t the ideal candidates for measuring the quality of approximation of the posterior distribution. However, for the complex class of distributions that we consider, better metrics are not available because the posterior is intractable and thus one cannot report KL-divergence based metrics for tasks other than linear regression and estimating the mean of a gaussian distribution, both of which are unimodal problems and thus much easier to solve than the other tasks presented in the paper. Bayesian methods have been claimed to lead to better generalization properties since they allow modeling of multiple hypotheses (with their respective probabilities) when making predictions. Thus, we believe that while these metrics do not quantify the divergence with the true posterior directly, they do test for parts of Bayesian inference that we often care about.
>
> **W2a**: This is not entirely true. For example, VI algorithms can lead to mode seeking behaviour and if this algorithm gets stuck in a lower probability mode, it could still lead to worse downstream performance than the true posterior, even through the lens of loss and accuracy metrics.
>
> **W2b**: We completely agree with the reviewer here that our goal would be to measure some notion of divergence or two-sample test with the true posterior. However, the unavailability of the true posterior makes such metrics unsuitable for our tasks. Additionally, the differences seem marginal in Table 5 exactly because it is a unimodal and much simpler problem compared to nonlinear regression and classification settings which are much more high dimensional with many more modes.
>
> **W2c**: One could replace the true posterior with samples from an MCMC algorithm, but this has its own set of problems. Firstly, for our MCMC results, we do run an MCMC algorithm for ~10k steps to get those results, which show worse performance than proposed and optimization baselines especially for high dimensional problems. Even if we were to run MCMC for longer, there is no clear idea on what “sufficiently long time” is and we will end up obtaining metrics against something that would be somewhat arbitrary. While we were setting up the experiments, we did however consider what the reviewer proposed and performed some initial two-sample classifier tests but we were not able to obtain any signal. In particular, for high dimensional problems the classifier would be able to learn with 100% accuracy and it is not clear whether this is a shortcoming in MCMC converging or the proposed approach.
>
> Additionally, we would like to stress to the reviewer that running MCMC “for sufficiently long time” is not feasible for the experiments that we do. Each point in all of the tables is obtained by averaging performance over 100 different datasets. Table 1 in itself is 11 experiments, with 100 datasets each with limited visibility into whether MCMC is even capturing the true posterior or not once we have run it. The problems in evaluating our proposed approach is the same problem that exists for MCMC, and thus we wanted to make sure our metrics are not confounded with shortcomings of MCMC. To this end, we test performance of different posterior approximation methods by testing for downstream generalization following the line of reasoning that Bayesian averaging or uncovering the full posterior should lead to more robust and better downstream performance.
>
> **W3**: We appreciate the author for bringing this point forward. Our takeaway is not that reverse KL is a better solution than forward KL and we do maintain that, as the reviewer pointed out, the overdispersion of forward KL can be a feature and it does prevent mode collapse which could be an important feature for some applications. Our claim in this work is to limit ourselves to parameter estimation in small-scaled models and understand that for the purpose of good and robust downstream performance, how do different amortized samplers perform. We will update our draft to provide this clarification and highlight a more nuanced comparison between the two. Overall, we don’t think there has been a lot of work in explicitly obtaining posteriors over parameters in an amortized (conditioned on datasets) manner and we present two valid ways of doing so, with a comparative analysis between the two.
>
> (1/2)

---

> > ### Author Response · Authors · 2024-10-07
> > **Author Response**
> >
> > **W4**: We hesitate to perform a comparison with non permutation-invariant architectures because such methods should inherently not be used in this setting, since the true posterior is permutation-invariant to the order of observations. Additionally, as mentioned, we randomly sample the number of observations and other architectures like MLPs cannot handle data with varying sizes.
> >
> > **W5**: We understand and agree with the reviewer’s point. The reason why we want to club methods like [Paige & Wood, 2016] into SBI as well is because we can see the graphical model as a simulator, and then training in this setup again reduces to standard simulation based inference. While we do have access to the likelihood in this setup, the paradigm is still to sample some parameters from the prior, use the simulator to get observations (this simulator can be neural network, a HMM, etc.) and then fit an inverse network to map the observations to the parameter space. That being said, we have updated the draft to more accurately call such approaches as SBI-NPE instead.
> >
> > **C2a**: The metric used in Table 3 is L2 loss.
> >
> > **C2b**: Even if the model is misspecified, there is a notion of true posterior for that particular model under those sets of observations. For example, even with nonlinear data one could perform Bayesian Linear Regression and there will be a notion of the true posterior for this model and data. The purpose of this table is to highlight how the two differently trained models perform when we do have model mis-specification in a manner that we have access to true data but not to the “simulator” that generates this true data. Just like in standard point estimation where the goal is to find the closest approximation to the true solution while being restricted to some hypothesis class; here the posterior for the mis-specified model should be a close approximation to the true posterior for the exact mis-specified model under the given set of observations.
> >
> > (2/2)

---

> ### Comment · Reviewer_oVsr · 2024-10-07
>
> Thank you for a thorough response, and the revised version of the manuscript!
>
> (ordered most to least important:)
>
> W2. With regard to your response to W2, W2c and your discussion with Reviewer VHDx (esp. the “Comparisons to NPs“ part):
> I think I see (at least) two routes for me to change my assessment of the “Evidence” criterium for this paper:
> a) predictive performance route: keep using predictive performance metrics like accuracy and L2-loss, follow the recommendations of Reviewer VHDx (e.g. compare to NPs), drop the claims of the goal of the method being approximating the posterior (because you’re not measuring directly how well it does on that goal).
> b) posterior approximation route: if it’s not possible to use the metrics of https://github.com/sbi-benchmark/sbibm (mentioned by both Reviewer VHDx and me) on the problems you’ve chosen due to them not being amenable to reliable MCMC inference, evaluate the approaches you’re comparing on the problems proposed by sbi-benchmark (since the authors of the benchmark seem to have figured out how to evaluate those metrics on the problems they propose to evaluate NPE methods on), and then narrate the results based on those results.
>
> I’m sorry, I don’t think I’ll accept the argument (apologies for strawmanning a little here:) “we assess the quality of the posterior approximation via predictive performance metrics (because improved predictive performance might be one of the claimed consequences of better posterior approximation, and we often care about the predictive performance metrics)”, I don’t think it’s a sound logical leap, especially if related work (sbi-benchmark) offers a sound evaluation protocol for the quality of the posterior without making any logical leaps.
>
>
> W4. imo the evaluation of the permutation-invariant architectures is one of the interesting aspects of this work, so it’s a shame you don’t want to drive that point home by comparing to an appropriate baseline in this setting.
>
> > such methods should inherently not be used in this setting
>
> We all know that non-invariant methods are used in invariant settings all the time. My guess is that the largest difference we’re talking about here is the sample-efficiency of learning and asymptotic performance for a fixed parameter count - would you agree?
>
> > we randomly sample the number of observations and other architectures like MLPs cannot handle data with varying sizes.
>
> That’s why I suggested RNNs, e.g., see Fig 3 in https://arxiv.org/abs/1610.09900
>
>
> W1. Thank you!
> W1a. In Table 6, the variance should have at least 1 significant digit.
> W1b. “Then, in the results highlight all of the rows for which the confidence intervals overlap with the best value.” - e.g., I’d like to see the “Rev-KL Flow Transformer NLC 25D” result highlighted because 75.8+0.8>=76.5. I’d like to see the same pattern of highlighting for all the other cases of such overlap (both in Table 6 as well as Table 1 in the main text, and all the others).
>
>
> W5. Acknowledged, thanks! I’m a bit happier with the updated SBI-NPE nomenclature!
>
> W2a. I've just realized I previously misunderstood what L2-loss you're computing in my original review - I should’ve read App D.
>
> I thought you're trying to assess the quality of the approximate posterior by comparing it to "the true generating value of the latent" $\theta^*$ as per
> $\mathbb{E}_{\theta \sim q(\theta | \mathcal{D})} \left[\text{L2}(\theta^*, \theta)\right]$.
> Now I understand it's also a predictive performance metric, just like accuracy. (Rereading the draft, you do make it pretty clear, so it’s only my fault)
>
> Hence, I think the answer to my original question is "no".
> Apologies if my question was confusing, it was due to a misunderstanding.

---

> > ### Author Response · Authors · 2024-10-15
> > **Author Response**
> >
> > We thank the reviewer for their prompt response and valuable comments, which we address below.
> >
> > **W2**: We would like to stress again that our goal is posterior inference, and not predictive performance; which is why option a, does not work. We would first like to take this opportunity to highlight that SBI-bench constructs most of the metrics through samples from a reference posterior, which is obtained via multiple steps of approximation. In particular, first it uses an MCMC sampler to obtain $10k$ samples from the unnormalized density. Then secondly, they use a normalizing flow on these MCMC samples to approximate this density which is another approximation. Finally, they use rejection sampling using a mixture of the normalizing flow and prior, for which they obtain the constant $M$ (refer Appendix B.1) through another approximation scheme. Thus, while such a methodology may work for very small scale problems, it would not work for high-dimensional problems especially at scale. For example, they consider only single observations and evaluate for 10 "datasets" (in quotes because they use only single observations per dataset). In comparison, for each of our tasks we use $100$ datasets to evaluate, with the number of observations ranging from $64$ to $128$. Furthermore, their biggest inference problems are at max $20$ dimensional (generally single digit) with an observation space of $10$ dimensions, while we consider inference problems with more than $1000$ parameters. At this scale, conducting MCMC followed by learning a normalizing flow and then finding the right constant for rejection sampling is unfortunately unfeasible; but these problems are still of massive interest for the community! Given the levels of approximation in SBI-Bench and their low-dimensional setup, we hope we have at least convinced the reviewer why predictive performance instead (which is often what we care about in machine learning; especially in terms of generalization) can be a good proxy.
> >
> > At the same time, we do understand the reviewer's concerns as well; and to alleviate them we are setting up an additional Hamiltonian Monte Carlo (HMC) baseline. The goal is to provide some semblance of a two-sample classifier test with the HMC samples as reference samples for the posterior.
> >
> > Finally, we would like to stress how important this problem is. Current methods largely look at Bayesian inference for model parameters for fixed datasets only, and our experiments into amortized inference allows more scalable learning by training conditional models for such approximate inference.
> >
> > **W4**: We appreciate the reviewer's point; and are currently in the middle of running experiments with a GRU baseline to alleviate their concerns. Given the large scale of our study, we haven't been able to finish the GRU experiments but we will update the reviewer once we have the results.
> >
> > We will also further update the draft with variance estimates, and highlight based on confidence interval overlap. As we said, it might make the tables messy so we might move some of the low dimensional experiments to the Appendix.
> >
> > Is there any other concerns or questions that the reviewer has? We would be more than happy to provide clarifications.

---

> > > ### Comment · Reviewer_oVsr · 2024-10-15
> > >
> > > > At the same time, we do understand the reviewer’s concerns as well; and to alleviate them we are setting up an additional Hamiltonian Monte Carlo (HMC) baseline. The goal is to provide some semblance of a two-sample classifier test with the HMC samples as reference samples for the posterior.
> > >
> > > Thanks a lot, really appreciate this, please ping me when you have an updated draft!
> > >
> > >
> > >
> > > >W4: We appreciate the reviewer’s point; and are currently in the middle of running experiments with a GRU baseline to alleviate their concerns. Given the large scale of our study, we haven’t been able to finish the GRU experiments but we will update the reviewer once we have the results.
> > > We will also further update the draft with variance estimates, and highlight based on confidence interval overlap. As we said, it might make the tables messy so we might move some of the low dimensional experiments to the Appendix.
> > > Is there any other concerns or questions that the reviewer has? We would be more than happy to provide clarifications.
> > >
> > > All of this sounds good, thanks!
> > >
> > > I think the only other concern I have is regarding the “Model misspecification” scenario, I will revisit this later.
> > >
> > >
> > >
> > > > we hope we have at least convinced the reviewer why predictive performance instead (which is often what we care about in machine learning; especially in terms of generalization) can be a good proxy.
> > >
> > > The way I hear what you're saying: the posterior approximation problems you're targeting are so difficult that even in a non-amortized setting (where we can spend large amounts of GPU compute with parallel MCMC https://github.com/google-research/google-research/tree/master/bnn_hmc), we still find it difficult to produce an approximation trustworthy enough for us to use it as a reference for credibly evaluating the single-forward-pass amortized approximation. Honestly, if we cannot approximate the solution to the problem reliably in a non-amortized way, that gives me next to no confidence in the ability of the amortized method to approximate the solution.
> > >
> > > If the non-amortized case of these problems is truly so hard, then I'd like to see convincing evaluation of the amortized posterior approximation on simpler problems, where we can generate sufficiently reliable non-amortized reference approximation or we can obtain closed-form/analytical reference solution.
> > >
> > > Even with the additional context provided by the authors: I think that predictive performance is a valuable goal in its own right, but I still do not consider it a convincing proxy metric for the quality of the posterior approximation - the fact that evaluating the quality of the posterior approximation is difficult doesn't imply that predictive performance is a good proxy for it.

---

> ### Author Response · Authors · 2024-10-15
> **Author Response**
>
> > The way I hear what you're saying: the posterior approximation problems you're targeting are so difficult that even in a non-amortized setting (where we can spend large amounts of GPU compute with parallel MCMC https://github.com/google-research/google-research/tree/master/bnn_hmc), we still find it difficult to produce an approximation trustworthy enough for us to use it as a reference for credibly evaluating the single-forward-pass amortized approximation. Honestly, if we cannot approximate the solution to the problem reliably in a non-amortized way, that gives me next to no confidence in the ability of the amortized method to approximate the solution.
>
> This is true for the work that the reviewer has linked as well. For example, HMC itself comes with its set of hyperparameters and the work linked majorly only considers whether HMC found the right posterior by looking at predictive performances (eg. accuracy and log likelihood); please correct us if not. What we are trying to say is that even in non-amortized setting, even with parallel MCMC, we are not aware of works that can guarantee that we get reliable posterior estimates; i.e. we only have asymptotic guarantees but we know in practice MCMC approaches and langevin dynamics remain quite dependent on the choice of hyperparameters.
>
> > If the non-amortized case of these problems is truly so hard, then I'd like to see convincing evaluation of the amortized posterior approximation on simpler problems, where we can generate sufficiently reliable non-amortized reference approximation or we can obtain closed-form/analytical reference solution.
>
> We completely agree that if the problems are so hard that it is difficult to get reference posterior samples to effectively evaluate posterior estimation, we should at least provide tractable cases and show that our amortized approach performs well on those such that we are being grounded in posterior estimation. To this end, we do already show for estimating the mean of the Gaussian as well as Bayesian linear regression, we do uncover the right posterior relatively well; these are cases where an analytic form is available. We further also show visualizations in Figures 1,2.
>
> > Even with the additional context provided by the authors: I think that predictive performance is a valuable goal in its own right, but I still do not consider it a convincing proxy metric for the quality of the posterior approximation - the fact that evaluating the quality of the posterior approximation is difficult doesn't imply that predictive performance is a good proxy for it.
>
> We feel that this requires additional discussion; in particular why do we even want to do posterior estimation well? Our outlook here is that a big use-case is being able to do better prediction and generalization. We would really appreciate hearing more about the reviewer's thoughts on this matter to be able to better resolve the disagreement here.

---

> ### Comment · Reviewer_oVsr · 2024-10-17
>
> >This is true for the work that the reviewer has linked as well. For example, HMC itself comes with its set of hyperparameters and the work linked majorly only considers whether HMC found the right posterior by looking at predictive performances (eg. accuracy and log likelihood); please correct us if not. What we are trying to say is that even in non-amortized setting, even with parallel MCMC, we are not aware of works that can guarantee that we get reliable posterior estimates; i.e. we only have asymptotic guarantees but we know in practice MCMC approaches and langevin dynamics remain quite dependent on the choice of hyperparameters.
>
> I mostly agree with everything quoted above, apart from the (not very significant) fact that the mentioned work considers also \hat{R} in Sec 5.1.
>
> However, I want to stress - I’m not asking you to test your amortized approach on models as complex as BNNs - much simpler models would suffice.
>
> >To this end, we do already show for estimating the mean of the Gaussian as well as Bayesian linear regression, we do uncover the right posterior relatively well; these are cases where an analytic form is available. We further also show visualizations in Figures 1,2.
>
> The most pressing issue I have with the paper in its current shape is that your quantitative results (Figs 1, 2, and 4ab don’t count as quantitative) are not highlighting the issue with the mode-seeking behavior of the approximate posteriors learned with Rev-KL (in contrast to Fwd-KL).
>
> To amend this, I’d like to ask you to evaluate the JSD (or even better just KL[p||q]) for the Gaussian Mixture Model in Table 5.
> I would imagine the qualitative phenomena of Figure 4ab might lead to an increase of the term KL[p||q] due to the missed mode of the posterior.
>
> > why do we even want to do posterior estimation well?
>
> From the Bayesian modelling perspective:
> 1. Posterior reflects our updated beliefs about the values of the parameters in the model - in many cases outside of predictive modelling, those variables have a real-world/physical meaning, and that motivates the accuracy of the posterior itself being a goal in its own right.
> 2. Maybe even more importantly, those posteriors are used for decision-making, e.g., choosing action $a$ to minimize the expected loss via $\argmin_a \E_{p(\theta | \mathcal{D})}[\mathcal{L}(\theta, a)]$. If we use an approximate posterior as a proposal distribution for importance sampling, see my original comment W3. Bayesian statistics are deeply rooted in decision theory as per [Robert, 2007, The Bayesian Choice].
> 3. More generally, many tasks can be cast as $\E_{p(\theta|\mathcal{D})}[f(\theta)]$ for some functional $f(\theta)$. The previous point was one example. Posterior predictive is another: $\E_{p(\theta|y)}[p(x^* | \theta)]$. If we consider any $f(x)$, once again, if we want to use an approximate posterior as a proposal distribution for importance sampling, see my original comment W3.
>
> My understanding (somewhat superficial, admittedly) is that for BNNs, due to symmetries of the posterior caused by the parameter permutation-invariance, you can achieve a decent improvement in predictive performance only approximating a single mode of the multi-modal symmetric posterior, so the weakness of the Rev-KL I’m complaining about doesn’t come through when using only predictive metrics.
>
> PS I just saw two recent interesting works related to the topic of multimodality of BNN posteriors: https://arxiv.org/abs/2310.08287, https://arxiv.org/abs/2408.05496

---

> > ### Comment · Reviewer_oVsr · 2024-10-31
> >
> > Thank you for adding the GRU baseline!
> >
> > To be honest, I find the ordering of the GRU/DeepSets/Transformer results a bit surprising in that I was hoping for a conclusion that GRU as a non-permutation-invariant always performs worst, but the results do convincingly show that the Transformer architecture performs competitively across all problem settings (most robustly out of all the architectures considered).

---

> > > ### Author Response · Authors · 2024-11-03
> > > **Author Response**
> > >
> > > We thank the reviewer for their response and useful comments, and for helping us improve our draft! We also apologize for the late response, but we are happy to say that we have amended the draft further to include additional experiments.
> > >
> > > > To amend this, I’d like to ask you to evaluate the JSD (or even better just KL[p||q]) for the Gaussian Mixture Model in Table 5. I would imagine the qualitative phenomena of Figure 4ab might lead to an increase of the term KL[p||q] due to the missed mode of the posterior.
> > >
> > > We would like to point the reviewer to Tables 5 and 6 as well as Figure 4, which evaluate the quality of posterior approximation in various ways. Table 5 shows the Jensen-Shannon Divergence for a subset of tasks for which the true posterior is available in closed form, and Figure 4 visualizes the same, showing that Forward KL does model the multi-modality better than Reverse KL. This is now also illustrated in quantitative metrics in Table 6, which computes the $W_2$ wasserstein metric between samples from 100 chains of Langevin based MCMC and those from the approximate posterior. We hope that this resolves the reviewer's concerns regarding posterior estimation?
> > >
> > > > Posterior reflects our updated beliefs about the values of the parameters in the model - in many cases outside of predictive modelling, those variables have a real-world/physical meaning, and that motivates the accuracy of the posterior itself being a goal in its own right.
> > >
> > > We definitely agree with the reviewer on this, but given the kind of probabilistic models that we consider (especially BNNs), we felt that in this space it is more reasonable to test them for their generalization capabilities in line with the claim that Bayesian estimates should generalize well by including multiple hypotheses in their prediction. While we agree with the following points that the reviewer makes, we do not know of applications of Bayesian inference in decision-making **when considering such high-dimensional objects**.
> > >
> > > > My understanding (somewhat superficial, admittedly) is that for BNNs, due to symmetries of the posterior caused by the parameter permutation-invariance, you can achieve a decent improvement in predictive performance only approximating a single mode of the multi-modal symmetric posterior, so the weakness of the Rev-KL I’m complaining about doesn’t come through when using only predictive metrics.
> > >
> > > While we agree with this, we do feel that some kinds of multi-modality is okay to not be covered. For example, the identifiability problem in clustering presents one such multi-modality, where even though the true posterior should cover all orderings of clusters but doing so is really not useful as all the modes do lead to the same equivalent solution (in terms of assignment of points to which cluster).
> > >
> > > > To be honest, I find the ordering of the GRU/DeepSets/Transformer results a bit surprising in that I was hoping for a conclusion that GRU as a non-permutation-invariant always performs worst, but the results do convincingly show that the Transformer architecture performs competitively across all problem settings (most robustly out of all the architectures considered).
> > >
> > > We have also updated the draft to discuss this in Section 5. We understand that it might be confusing as to why GRU outperforms DeepSets, and we believe this is because DeepSets relies on a particular aggregation operation (eg. sum or mean) which could be good or bad depending on the task. Transformers do not rely on such limitation and are thus perform robustly well. Similarly, GRUs also do not specify an aggregation operation and can thus learn more flexible functions that DeepSets cannot; and the lack of invariance inductive bias in GRUs can be overcome from the training data. Since we train on simulations on the fly, the abundance of data can lead to GRUs essentially learning a soft form of permutation invariance even though it is not explicitly baked in.
> > >
> > > We hope that our revised draft where we have made the writing clearer, added standard deviations for our experiments, added the GRU baseline and evaluated directly the quality of the posterior through an MCMC proxy to the oracle, have resolved the reviewer's concerns.

---

> > > > ### Author Response · Authors · 2024-11-12
> > > > **Author Response**
> > > >
> > > > Thanks to the reviewer, we have now incorporated the changes in the Appendix as well, including but not limited to additional experiments with the GRU baseline and standard deviation for all our experiments.

---

### Review · Reviewer_VHDx · 2024-09-20

**Summary Of Contributions:**

The paper proposes a framework for approximate posterior inference amortized over different datasets and over different feature dimensionality.

Experiments in both real world and simulated settings are used to study the approach in comparison to SBI.

**Audience:**

No

**Broader Impact Concerns:**

No concerns.

**Claims And Evidence:**

Yes

**Requested Changes:**

- If you want to use SBI as a baseline explain that it is typically used in a different setting, then introduce NPE and say that you are bringing ideas from NPE, namely amortizing over datasets to the VI setting.
- Compare to a NP baseline, especially since the evaluation metrics you use measure loss and accuracy, or later simply provide an initialization for MAP.
- use some kind of evaluation metric that assesses the quality of the entire posterior not just point estimates. (for ideas see sbi benchmark)
- In terms of describing the method in section 3, it would have been good to explain the set up: which data is given, which parts of the model are assumed to be known etc. before diving into details.
- Both in the background section and in the methods section the prior is somehow swept under the rug. If you are given a dataset and are supposed to predict the posterior distribution, that posterior (e.g. how sharp it is) will depend on the prior. You are just calling it random but it would have been good to discuss explicitly.
- Improve the explanation of the model misspecification setting. Since the modeling set-up including the prior (see two points above) could be described more clearly, the exact details of the model misspecification setting are difficult to understand, both for your approach and for SBI/NPE.
- Consider reframing your approach either as a generalization of neural processes and then also describe the method section that way, or by deriving your approach from a meta learning formulation of posterior inference.

**Strengths And Weaknesses:**

Strengths:

- It is a nice idea to use permutation invariant architectures to be able to handle datasets with varying numbers of observations
- It is a nice idea to amortize also over varying number of feature dimensions
- I like that the authors also study their approach under model misspecification

Weaknesses:

- Given that the authors focus on *Tractable Likelihood Models" (see title), SBI does not seem like the right comparison baseline which is typically designed and use when a tractable likelihood is not available
- Instead, baselines to consider include neural processes (NP) or formulating inference with different datasets as a meta learning problem and then using a meta learning based approach.
- The authors mention that their approach is an extension to NPs that provides a full Bayesian treatment to all parameters, but the metrics chosen for experimental evaluation don't really evaluate the quality of the full posterior but are instead L2 loss and accuracy (table 1) and in the experiments with real data, the result of their approach (please clarify if you use a sample from the posterior or the mean) is used to initialize a MAP approximation. In those evaluation settings, one might as well use NPs. Also, since these evaluation settings don't address the quality of the full posterior, it is clear that they put the mode-seeking behavior of the backward KL in an advantage over the mode covering behavior of the backward KL objective.
- SBI description: When introducing SBI in their background section, the authors should start by saying that SBI is typically used in the likelihood free setting and explain why amortization has emerged there. SBI is a wide class of approaches. After introducing it, the authors might want to mention neural posterior estimation (NPE). This is the actual baseline they are comparing against. Then throughout the paper they can call it NPE

---

> ### Author Response · Authors · 2024-10-07
> **Author Response**
>
> We thank the reviewer for their useful feedback and comments, and hope that our response addresses the concerns raised.
>
> **Goal and Motivation**: We believe that the reviewer may have misunderstood the main motivation and goal of our work. Our goal is accurate posterior estimation in an amortized fashion for arbitrary probabilistic models and the reason we use downstream prediction loss and accuracy is because the other metrics for this task are either intractable or can be problematic (eg. using MCMC to approximate true posterior and then compute metrics). With this in mind, we think that SBI, or forward KL minimization setups, while not leveraging the likelihood information, are still relevant methods for performing posterior estimation. It is also for this exact reason that we do not look at meta learning or neural processes since they are often obtaining posteriors over arbitrary latent variables and not parameters of a pre-defined probabilistic model. If there are any other approaches that the reviewer recommends that are better suited for this problem, we would be happy to discuss and include it.
>
> **Real World Experiments and Point Estimates**: We refer the reviewer to Figure 3. In this figure, the left-most point in each sub-figures corresponds to epoch 0, which means no training has been done at this point. The loss and accuracy for these models are obtained by taking multiple samples from the posterior estimate from forward and reverse KL approaches, and is essentially the following quantity:
>
> $\mathbb{E}\_{\theta \sim q(\theta | \mathcal{D})} \left[\text{Metric}[y, f_\theta(x)]\right]$
>
> Correspondingly, during training we perform MAP based fine tuning on each of these multiple samples of $\theta$ obtained above.
>
> **Quality of full posterior**: We completely agree with the reviewer that it would be ideal to have some measure of divergence between the approximate and true posterior, but in this regime of intractable posteriors we can only rely on proxies. In this work, we are considering predictive generalization as a proxy for good posterior estimation, or at least in the sense that is usable and desirable for a lot of machine learning problems.
>
> **SBI vs NPE**: We agree with the reviewer, and have made changes to make it clear that the goal of this work is posterior estimation and we compare against a specific method of SBI, i.e. NPE. We choose to call it SBI-NPE instead of solely NPE because reverse KL based methods for approximating the posterior also fall under the umbrella of neural posterior estimation (i.e. we are still estimating the posterior distribution using neural models).
>
> **SBI used in a different setting**: We refer the reviewer to Section 2, that highlights that SBI is typically used in settings without tractable likelihood functions.
>
> **Comparisons to NPs**: As we discussed above, our goal is to efficiently approximate the posterior distribution over the parameters of parametric probabilistic models. In contrast, NPs often deal with predictive modeling for regression and classification tasks, which is an altogether different goal from our work. We agree that we do look at loss and accuracy metrics, but that is solely to be used as proxies to the quality of posterior estimation. That is, we often care about good posterior estimation to be able to make more robust downstream predictions.
>
> **Prior**: The reviewer brings up an important point and we have updated our draft to discuss the prior more precisely. We typically consider the prior as $\mathcal{N}(0, I)$, and for all our tasks, we do sample parameters from this prior in generating data (except Gaussian Processes and real-world datasets). Thus, generally our choice of prior is either correctly aligned with the data generating process, or uninformative as is often the case for many real world problems.
>
> **Model misspecification**: We will additionally update the draft further to clarify our explanation for this. Our setting is simply that we have a ground truth data generating distribution $p_{true}(\cdot | \theta)$ and an assumption of the world $p_{model}(\cdot | \theta')$. Given a set of observations $\mathcal{D}$ from the former, we want to approximate the posterior of the latter under such observations. This can be done easily for reverse KL which does not need to rely on data generated from the latter. However, for training with forward KL one needs to synthesize data corresponding to the latter (not the true data generating system). Hence, we study both reverse and forward KL setups with data generated from the model, as well as reverse KL with data generated from the former. This is in line with typical ML problems, where true generating distribution is unknown and we solely have an assumption over it by defining a modeling space (latter).

---

> > ### Author Response · Authors · 2024-10-15
> > **Request for Discussion**
> >
> > We thank the reviewer for their time and valuable feedback; and would like to ask if our rebuttal addresses their concerns? We would be happy to provide further clarifications if the reviewer has more questions.

---

> > > ### Comment · Reviewer_VHDx · 2024-10-27
> > > **Evaluation of Posterior**
> > >
> > > I still disagree with the following sentence:
> > > "As a result, simulation-based inference (SBI) styled neural posterior estimation approaches (SBI-NPE) (Radev et al., 2020; Geffner et al., 2023; Cranmer et al., 2020) which rely on training with datasets and their corresponding known parameters may struggle to generalize to data of practical use."
> > > See Cranmer et al. 2020 for examples of SBI of practical use. Instead say something like "As a result, simulation-based inference (SBI) styled neural posterior estimation approaches (SBI-NPE) (Radev et al., 2020; Geffner et al., 2023; Cranmer et al., 2020) which rely on training with datasets and their corresponding known parameters are not applicable when the underlying parameters are unknown which occurs in many practical settings."
> > >
> > > Multiple reviewers have raised the concern that the posteriors are only evaluated in terms of predictive accuracy. See e.g. the SBI benchmark paper for inspiration on how to evaluate a posterior empirically (Luecke et al. 2021). You might not need to do an MCMC based baseline for all the parameters you are inferring (amortizing over) but could pick a specific parameter from one of the regression or classification datasets and overlay how samples from the baseline compare to your posterior estimate of the parameter e.g. via a calibration plot or a plot similar to what you show in Figure 2.

---

> > > > ### Author Response · Authors · 2024-10-30
> > > > **Author Response**
> > > >
> > > > We thank the reviewer for their comments, and address them below:
> > > >
> > > > > I still disagree with the following sentence... Instead say something
> > > >
> > > > We have updated the sentence in the revised version. Additionally, we would like to clarify that the problem does not only exist in settings where the underlying parameters are unknown; but also when the posterior is high-dimensional and multi-modal, as is evidenced by our experiments in Tables 1 and 2.
> > > >
> > > > > You might not need to do an MCMC based baseline for all the parameters you are inferring (amortizing over) but could pick a specific parameter from one of the regression or classification datasets and overlay how samples from the baseline compare to your posterior estimate of the parameter e.g. via a calibration plot or a plot similar to what you show in Figure 2.
> > > >
> > > > Would this be different in spirit from Figure 4 (c) and (d) where we do overlay the kernel density estimate for linear regression with the ground-truth posterior present as well (as opposed to MCMC since we have access to the true posterior here). Is the reviewer looking for something else in particular?
> > > >
> > > > We would also like to bring to reviewer's attention that in the latest revision of our work, we have included standard deviations for all our results in the main draft. We have also included a non permutation invariant architecture (GRU) as baseline to our experiments. We are in the process of doing the same for the appendix results as well.

---

### Decision · Action_Editor_hygV · 2024-11-19

**Recommendation:** Reject

**Comment:**

I want to begin by apologizing to the authors for the delayed review process — I appreciate their patience. However, in spite of these delays, I believe the reviewers have seriously considered the manuscript and have provided productive commentary to the authors. The clarity of the claims and the evidence provided is the primary topic I would recommend for the authors to reconsider in a holistic way. The reviewers are looking for a framing of the results primarily in terms of information theoretic metrics to reach the conclusion that the experiments back the proposed claims. I would also recommend that the authors carefully revisit the nature and order of the figures and tables to maximize clarity (e.g., is Table 3 showing L2 losses?I don't believe Table 3 is ever fully defined).

**Audience:**

TMLR's audience would be interested in these topics.

**Claims And Evidence:**

The reviewers and authors have engaged in a thorough post-review conversation, with multiple edits made to the manuscript and its appendix. A primary concern revolves around how the reviewers interpret the claims in the manuscript (reverse KL based amortized VI is a good way to perform amortized inference) relative to the evidence provided. The initial framing of the evidence under predictive accuracy remains a large part of the narrative, and the additional studies that the authors have conducted have not led to consensus among the reviewers. The current version of the PDF has significant edits — it would benefit from a holistic review (e.g., Figures 1 and 2 are seemingly very similar; their caption does not clearly explain them; both are referenced 2-3 pages after inserted in the manuscript, etc.)

**Resubmission Of Major Revision:**

The authors may consider submitting a major revision at a later time.